# VARIATIONAL DIFFUSION POSTERIOR SAMPLING WITH MIDPOINT GUIDANCE

**Badr Moufad**[*,1]  **Yazid Janati**[*,1]  **Lisa Bedin**[*,1]
**Alain Durmus**[1]  **Randal Douc**[2]  **Eric Moulines**[1]  **Jimmy Olsson**[3]
[1]Ecole polytechnique  [2]Télécom SudParis  [3]KTH Royal Institute of Technology

## ABSTRACT

Diffusion models have recently shown considerable potential in solving Bayesian inverse problems when used as priors. However, sampling from the resulting denoising posterior distributions remains a challenge as it involves intractable terms. To tackle this issue, state-of-the-art approaches formulate the problem as that of sampling from a surrogate diffusion model targeting the posterior and decompose its scores into two terms: the prior score and an intractable guidance term. While the former is replaced by the pre-trained score of the considered diffusion model, the guidance term has to be estimated. In this paper, we propose a novel approach that utilises a decomposition of the transitions which, in contrast to previous methods, allows a trade-off between the complexity of the intractable guidance term and that of the prior transitions. We validate the proposed approach through extensive experiments on linear and nonlinear inverse problems, including challenging cases with latent diffusion models as priors. We then demonstrate its applicability to various modalities and its promising impact on public health by tackling cardiovascular disease diagnosis through the reconstruction of incomplete electrocardiograms. The code is publicly available at https://github.com/yazidjanati/mgps.

## 1 INTRODUCTION

Inverse problems aim to reconstruct signals from incomplete and noisy observations and are prevalent across various fields. In signal and image processing, common examples include signal deconvolution, image restoration, and tomographic image reconstruction (Stuart, 2010; Idier, 2013). Other applications extend to protein backbone motif scaffolding (Watson et al., 2023) and urban mobility modeling (Jiang et al., 2023). Due to the presence of noise and the inherent complexity of the measurement process, these problems are typically ill-posed, meaning they have an infinite number of possible solutions. While many of these solutions may fit the observed data, only a few align with the true underlying signal. Bayesian inverse problems provide a principled framework for addressing the challenges of signal reconstruction by incorporating prior knowledge, enabling more plausible and meaningful solutions. The *a priori* knowledge about the signal $\mathbf{x}$ to be reconstructed is captured in the prior distribution $q(\mathbf{x})$, while the information about the observation $\mathbf{y}$ is encoded in the likelihood function $p(\mathbf{y}|\mathbf{x})$. Given these components, solving the inverse problem boils down to sampling from the posterior distribution $\pi(\mathbf{x}) \propto p(\mathbf{y}|\mathbf{x})q(\mathbf{x})$, which integrates both prior knowledge and observational data.

The choice of the prior distribution is crucial for achieving accurate reconstructions. If the goal is to reconstruct high-resolution data, the prior must be able to model data with similar fine detail and complexity. Recently, denoising diffusion models (DDMs) (Sohl-Dickstein et al., 2015; Song & Ermon, 2019; Ho et al., 2020; Song et al., 2021b) have emerged as a powerful approach in this context. These models can generate highly realistic and detailed reconstructions and are becoming increasingly popular as priors. Using a DDM, a sample $X_0$ being approximately distributed according to a given data distribution $q$ of interest can be generated by iteratively denoising an initial sample

---

[*] authors contributed equally.
Correspondence: {firstname.secondname@polytechnique.edu}

$X_n$ from a standard Gaussian distribution. The denoising process $(X_k)_{k=n}^0$ consists of the iterative refinement of $X_k$ for $k \in [\![1, n]\!]$ based on a parametric approximation of the score $\nabla \log q_k$, where $q_k(\mathbf{x}_k) = \int q(\mathbf{x}_0) q_{k|0}(\mathbf{x}_k|\mathbf{x}_0) \, \mathrm{d}\mathbf{x}_0$, with $q_{k|0}(\cdot|\mathbf{x}_0)$ being a Gaussian noising transition.

Denoising Diffusion Models form a highly powerful class of prior distributions, but their use introduces significant challenges in posterior sampling. Specifically, since a DDM prior does not admit an explicit and tractable density, conventional Markov chain Monte Carlo (MCMC) methods, such as the Metropolis–Hastings algorithm and its variants (Besag, 1994; Neal, 2011), cannot be applied in general. Furthermore, gradient-based MCMC methods often prove inefficient, as they tend to get trapped in local modes of the posterior.

The problem of sampling from the posterior in Bayesian inverse problems with DDM priors has recently been addressed in several papers (Kadkhodaie & Simoncelli, 2020; Song et al., 2021b; Kawar et al., 2021; 2022; Lugmayr et al., 2022; Ho et al., 2022; Chung et al., 2023; Song et al., 2023a). These methods typically modify the denoising process to account for the observation $\mathbf{y}$. One principled approach for accurate posterior sampling involves skewing the denoising process at each stage $k$ using the score $\nabla \log \pi_k$, where $\pi_k$ is defined analogously to $q_k$, with $q$ replaced by the posterior $\pi$. This guides the denoising process in a manner that is consistent with the observation. Notably, the posterior score $\nabla \log \pi_k$ decomposes into two components: the prior score and an additional intractable term commonly referred to as *guidance*. Previous works leverage *pre-trained* DDMs for the prior score, while providing various *training-free* approximations for the guidance term (Ho et al., 2022; Chung et al., 2023; Song et al., 2023a; Finzi et al., 2023). This framework enables solving a wide range of inverse problems for a given prior. However, despite the notable success of these methods, the approximations often introduce significant errors, leading to inaccurate reconstructions when the posterior distribution is highly multimodal, the measurement process is strongly nonlinear, or the data is heavily contaminated by noise.

**Our contribution.** We begin with the observation that the posterior denoising step, which transforms a sample $X_{k+1}$ into a sample $X_k$, does not necessarily require the conditional scores $\nabla \log \pi_{k+1}$ or $\nabla \log \pi_k$. Instead, we demonstrate that this step can be decomposed into two intermediate phases: first, denoising $X_{k+1}$ into an intermediate *midpoint* state $X_{\ell_k}$, where $\ell_k < k$, and then noising back to obtain $X_k$, unconditionally on the observation $\mathbf{y}$. This decomposition introduces an additional degree of freedom, as it only requires the estimation of the guidance term at the intermediate step $\ell_k$ rather than step $k+1$. Building on this insight, we introduce MIDPOINT GUIDANCE POSTERIOR SAMPLING (MGPS), a novel diffusion posterior sampling scheme that explicitly leverages this approach. Our algorithm develops a principled approximation of the denoising transition by utilizing a Gaussian variational approximation, combined with the guidance approximation proposed by Chung et al. (2023) at the intermediate steps $\ell_k$. The strong empirical performance of our method is demonstrated through an extensive set of experiments. In particular, we validate our approach on a Gaussian mixture toy example, as well as various linear and nonlinear image reconstruction tasks–inpainting, super-resolution, phase retrieval, deblurring, JPEG dequantization, high-dynamic range–using both pixel-space and latent diffusion models (LDM). Finally, we demonstrate the versatility of our approach by applying it to cardiovascular diagnosis using reconstructed electrocardiograms, showing that MGPS achieves significant improvements over competing methods.

## 2 POSTERIOR SAMPLING WITH DDM PRIOR

**Problem setup.** In this paper, we focus on the approximate sampling from a density of the form

$$\pi(\mathbf{x}) := p(\mathbf{y}|\mathbf{x}) q(\mathbf{x}) / \mathcal{Z} , \tag{2.1}$$

where $p(\mathbf{y}|\cdot) : \mathbb{R}^d \to \mathbb{R}$ is a *non-negative* and *differentiable* likelihood that *can be evaluated pointwise*, $q$ is a prior distribution, and $\mathcal{Z} := \int p(\mathbf{y}|\mathbf{x}) q(\mathbf{x}) \, \mathrm{d}\mathbf{x}$ is the normalizing constant. We are interested in the case where a DDM $p_0^\theta$ for the prior has been pre-trained based on i.i.d. observations from $q$. As a by-product, we also have parametric approximations $(\mathbf{s}_k^\theta)_{k=1}^n$ of the *scores* $(\nabla \log q_k)_{k=1}^n$, where the marginals $(q_k)_{k=1}^n$ are defined as $q_k(\mathbf{x}_k) := \int q(\mathbf{x}_0) q_{k|0}(\mathbf{x}_k|\mathbf{x}_0) \, \mathrm{d}\mathbf{x}_0$ with $q_{k|0}(\mathbf{x}_k|\mathbf{x}_0) := \mathcal{N}(\mathbf{x}_k; \sqrt{\alpha_k} \mathbf{x}_0, v_k \mathbf{I}_d)$ and $v_k := 1 - \alpha_k$. Typically, the sequence $(\alpha_k)_{k=0}^n$ is a decreasing sequence with $\alpha_0 = 1$ and $\alpha_n$ approximately equals zero for $n$ large enough.

These score approximations enable the generation of new samples from $q$ according to one of the DDM sampling schemes (Song & Ermon, 2019; Ho et al., 2020; Song et al., 2021a;b; Karras

et al., 2022). All these approaches boil down to simulating a Markov chain $(X_k)_{k=n}^0$ backwards in time starting from a standard Gaussian $p_n := \mathrm{N}(0_d, \mathbf{I}_d)$ and following the Gaussian transitions $p_{k|k+1}^\theta(\mathbf{x}_k|\mathbf{x}_{k+1}) = \mathcal{N}(\mathbf{x}_k; \mathbf{m}_{k|k+1}^\theta(\mathbf{x}_{k+1}), v_{k|k+1}\mathbf{I}_d)$, where the mean functions $\mathbf{m}_{k|k+1}^\theta$ can be derived from the learned DDM, and $v_{k|k+1} > 0$ are fixed and pre-defined variances. The backward transitions are designed in such a way that the marginal law of $X_k$ approximates $q_k$; see Appendix A.1 for more details on the form of the backward transitions.

In our problem setup, we assume we only have access to the approximated scores of the prior and no observation from neither the posterior $\pi$ nor the prior $q$. This setup encompasses of solving an inverse problem without training a conditional model from scratch based on a paired signal/observation dataset; a requirement typically imposed by conditional DDM frameworks (Song et al., 2021b; Batzolis et al., 2021; Tashiro et al., 2021; Saharia et al., 2022). On the other hand, we require access to $p(\mathbf{y}|\cdot)$. This setup includes many applications in Bayesian inverse problems, *e.g.*, image restoration, motif scaffolding in protein design (Trippe et al., 2023; Wu et al., 2023), and trajectory control in traffic-simulation frameworks (Jiang et al., 2023).

**Conditional score.** Since we have access to a DDM model for $q$, a natural approach to sampling from $\pi$ is to define a DDM approximation based on the pre-trained score approximations for $q_k$. Following the DDM approach, the basic idea here is to sample sequentially from the marginals $\pi_k(\mathbf{x}_k) := \int \pi(\mathbf{x}_0)q_{k|0}(\mathbf{x}_k|\mathbf{x}_0)\,\mathrm{d}\mathbf{x}_0$ by relying on approximations of the conditional scores $(\nabla \log \pi_k)_{k=1}^n$. The latter can be expressed in terms of the unconditional scores by using

$$\pi_k(\mathbf{x}_k) \propto \int p(\mathbf{y}|\mathbf{x}_0)q(\mathbf{x}_0)q_{k|0}(\mathbf{x}_k|\mathbf{x}_0)\,\mathrm{d}\mathbf{x}_0\,, \tag{2.2}$$

where after defining the backward transition kernel $q_{0|k}(\mathbf{x}_0|\mathbf{x}_k) := q(\mathbf{x}_0)q_{k|0}(\mathbf{x}_k|\mathbf{x}_0)/q_k(\mathbf{x}_k)$ yields

$$\pi_k(\mathbf{x}_k) \propto p_k(\mathbf{y}|\mathbf{x}_k)q_k(\mathbf{x}_k)\,, \quad \text{where} \quad p_k(\mathbf{y}|\mathbf{x}_k) := \int p(\mathbf{y}|\mathbf{x}_0)q_{0|k}(\mathbf{x}_0|\mathbf{x}_k)\,\mathrm{d}\mathbf{x}_0\,. \tag{2.3}$$

It then follows that

$$\nabla \log \pi_k(\mathbf{x}_k) = \nabla \log q_k(\mathbf{x}_k) + \nabla \log p_k(\mathbf{y}|\mathbf{x}_k)\,.$$

While we have a parametric approximation of the first term on the r.h.s., the second term is the bottleneck as it involves the integration of $p(\mathbf{y}|\cdot)$ against the conditional distribution $q_{0|k}(\cdot|\mathbf{x}_k)$.

**Existing approaches.** To circumvent this computational bottleneck, previous works have proposed approximations that involve replacing $q_{0|k}(\cdot|\mathbf{x}_k)$ in (2.3) with either a Dirac delta mass (Chung et al., 2023) or a Gaussian approximation (Ho et al., 2022; Song et al., 2023a)

$$q_{0|k}(\mathbf{x}_0|\mathbf{x}_k) \approx \mathcal{N}(\mathbf{x}_0; \mathbf{m}_{0|k}^\theta(\mathbf{x}_k), v_{0|k}\mathbf{I}_d)\,, \tag{2.4}$$

where $\mathbf{m}_{0|k}^\theta$ is a parametric approximation of $\mathbf{m}_{0|k}(\mathbf{x}_k) := \int \mathbf{x}_0\, q_{0|k}(\mathbf{x}_0|\mathbf{x}_k)\,\mathrm{d}\mathbf{x}_0$ and $v_{0|k}$ is a tuning parameter. Using Tweedie's formula (Robbins, 1956), it holds that $\mathbf{m}_{0|k}(\mathbf{x}_k) = (-\mathbf{x}_k + \sqrt{\alpha_k}\nabla \log q_k(\mathbf{x}_k))/v_k$, which suggests the approximation $\mathbf{m}_{0|k}^\theta(\mathbf{x}_k) := (-\mathbf{x}_k + \sqrt{\alpha_k}\mathbf{s}_k^\theta(\mathbf{x}_k))/v_k$. As for the variance parameter $v_{0|k}$, Ho et al. (2022) suggest setting $v_{0|k} = v_k/\alpha_k$, while Song et al. (2023a) use $v_{0|k} = v_k$. When $p(\mathbf{y}|\cdot)$ is the likelihood of a linear model, *i.e.*, $p(\mathbf{y}|\mathbf{x}) := \mathcal{N}(\mathbf{y}; \mathbf{A}\mathbf{x}, \sigma_{\mathbf{y}}^2\mathbf{I}_{d_{\mathbf{y}}})$, for $\mathbf{A} \in \mathbb{R}^{d_{\mathbf{y}} \times d}$ and $\sigma_{\mathbf{y}} > 0$, it can be exactly integrated against the Gaussian approximation (2.4), yielding $p_k(\mathbf{y}|\mathbf{x}_k) \approx \mathcal{N}(\mathbf{y}; \mathbf{A}\mathbf{m}_{0|k}^\theta(\mathbf{x}_k), \sigma_{\mathbf{y}}^2\mathbf{I}_{d_{\mathbf{y}}} + v_{0|k}\mathbf{A}\mathbf{A}^\intercal)$. Chung et al. (2023), on the other hand, use the pointwise approximation, yielding $p_k(\mathbf{y}|\mathbf{x}_k) \approx p(\mathbf{y}|\mathbf{m}_{0|k}^\theta(\mathbf{x}_k))$. We denote by $\tilde{p}_k(\mathbf{y}|\cdot)$ the resulting approximation stemming from any of these methods. Approximate samples from the posterior distribution are then drawn by simulating a backward Markov chain $(X_k)_{k=n}^0$, where $X_n \sim p_n$ and then, given $X_{k+1}$, $X_k$ is obtained via the update

$$\mathbf{X}_k := \tilde{X}_k + w_{k+1}(X_{k+1})\nabla \log \tilde{p}_{k+1}(\mathbf{y}|X_{k+1})\,, \quad \text{where} \quad \tilde{X}_k \sim p_{k|k+1}^\theta(\cdot|X_{k+1})\,, \tag{2.5}$$

and $w_{k+1}$ is a method-dependent weighting function. For instance, Chung et al. (2023) use $w_{k+1}(X_{k+1}) = \zeta\sigma_{\mathbf{y}}^2/\|\mathbf{y} - \mathbf{A}\mathbf{m}_{0|k+1}^\theta(X_{k+1})\|$, with $\zeta \in [0, 1]$. We note that the update (2.5) in general involves a vector-Jacobian product, *e.g.*, considering the approximation suggested by Chung et al. (2023), $\nabla_{\mathbf{x}_k} \log \tilde{p}_k(\mathbf{y}|\mathbf{x}_k) = \nabla_{\mathbf{x}_k}\mathbf{m}_{0|k}^\theta(\mathbf{x}_k)^\intercal\nabla_{\mathbf{x}_0} \log p(\mathbf{y}|\mathbf{x}_0)|_{\mathbf{x}_0 = \mathbf{m}_{0|k}^\theta(\mathbf{x}_k)}$, which incurs an additional computational cost compared to an unconditional diffusion step.

## 3  THE MGPS ALGORITHM

In this section we propose a novel scheme for the approximate inference of $\pi$. We start by presenting a midpoint decomposition of the backward transition that allows us to trade adherence to the prior backward dynamics for improved guidance approximation.

We preface our description of the MGPS algorithm with some additional notations. First, consider the DDPM forward process with transitions given by $q_{k|j}(\mathbf{x}_k|\mathbf{x}_j) := \mathcal{N}(\mathbf{x}_k; \sqrt{\alpha_k/\alpha_j}\mathbf{x}_j, (1-\alpha_k/\alpha_j)\mathbf{I}_d)$ for all $(j,k) \in [\![0,n]\!]^2$ such that $j < k$. We also define the joint law of the forward process

$$\pi_{0:n}(\mathbf{x}_{0:n}) := \pi(\mathbf{x}_0)\prod_{k=0}^{n-1} q_{k+1|k}(\mathbf{x}_{k+1}|\mathbf{x}_k)\,, \tag{3.1}$$

when initialized with the posterior (2.1) of interest. Note that with these definitions, $\pi_k$ (given by (2.3)) is the marginal of $\pi_{0:n}(\mathbf{x}_{0:n})$ w.r.t. $\mathbf{x}_k$. The time-reversed decomposition of (3.1) writes

$$\pi_{0:n}(\mathbf{x}_{0:n}) := \pi_n(\mathbf{x}_n)\prod_{k=0}^{n-1} \pi_{k|k+1}(\mathbf{x}_k|\mathbf{x}_{k+1})\,, \tag{3.2}$$

where, more generally, for all $(i,j) \in [\![0,n]\!]^2$ such that $i < j$,

$$\pi_{i|j}(\mathbf{x}_i|\mathbf{x}_j) := \pi_i(\mathbf{x}_i)q_{j|i}(\mathbf{x}_j|\mathbf{x}_i)\big/\pi_j(\mathbf{x}_j) \propto p_i(\mathbf{y}|\mathbf{x}_i)q_{i|j}(\mathbf{x}_i|\mathbf{x}_j)\,, \tag{3.3}$$

where the marginals are given by (2.2), is the conditional density of $X_i$ given $X_j = \mathbf{x}_j$ and $\mathbf{y}$.

Using the decomposition (3.2), an exact draw $X_0$ from (2.1) is obtained by first drawing $X_n$ from $\pi_n$ and then simulating recursively $X_k$ from the transitions $\pi_{k|k+1}(\cdot|X_{k+1})$ for all $k$. However, according to (3.3), such transitions involve the intractable likelihood $p_k(\mathbf{y}|\mathbf{x}_k)$ which is hard to approximate accurately when $k$ is large.

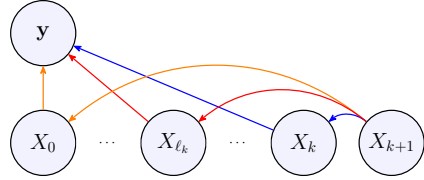

**Midpoint decomposition.**  The transition (3.3) has two components: the conditional density $p_i(\mathbf{y}|\mathbf{x}_i)$ and the prior transition density $q_{i|j}(\mathbf{x}_i|\mathbf{x}_j)$. The approximation $p_i(\mathbf{y}|\mathbf{x}_i) \approx p(\mathbf{y}|\mathbf{m}_{0|i}^\theta(\mathbf{x}_i))$ of Chung et al. (2023) is accurate when $i$ is small, whereas the Gaussian approximation of $q_{i|j}(\mathbf{x}_i|\mathbf{x}_j)$ proposed by Ho et al. (2020) is accurate when the difference $|i-j|$ is small. Thus, in order to combine the best of both worlds, we introduce a decomposition that transfers the problem of sampling from

Figure 1: For each color, the different solid arrows indicate different conditional densities that need to be approximated for a given choice of the midpoint $\ell_k$. The longer the arrow, the more difficult it is to approximate the corresponding conditional density. By placing the $\ell_k$ midway between zero and $k$, the shortest arrows are obtained.

$\pi_{k|k+1}$ to that of sampling from $\pi_{\ell_k|k+1}$, *i.e.*, drawing $X_{\ell_k}$ given $X_{k+1}$ and $\mathbf{y}$, where $\ell_k < k$. The parameter $\ell_k$ balances the approximation errors of the two conditional densities in (3.3); see Figure 1. In order to make this construction, define, for each $\ell \in [\![0,k]\!]$, the bridge kernel $q_{k|\ell,k+1}(\mathbf{x}_k|\mathbf{x}_\ell,\mathbf{x}_{k+1}) \propto q_{k|\ell}(\mathbf{x}_k|\mathbf{x}_\ell)q_{k+1|k}(\mathbf{x}_{k+1}|\mathbf{x}_k)$, which is Gaussian; see Appendix A.1. By convention, $q_{k|k,k+1}(\cdot|\mathbf{x}_k,\mathbf{x}_{k+1}) = \delta_{\mathbf{x}_k}$.

**Lemma 3.1.** *For all $k \in [\![1,n-1]\!]$ and $\ell_k \in [\![0,k]\!]$ it holds that*

$$\pi_{k|k+1}(\mathbf{x}_k|\mathbf{x}_{k+1}) = \int q_{k|\ell_k,k+1}(\mathbf{x}_k|\mathbf{x}_{\ell_k},\mathbf{x}_{k+1})\pi_{\ell_k|k+1}(\mathbf{x}_{\ell_k}|\mathbf{x}_{k+1})\,\mathrm{d}\mathbf{x}_{\ell_k}\,. \tag{3.4}$$

The proof is found in Appendix A.3. Lead by the decomposition provided by Lemma 3.1, we define

$$\hat{\pi}_{k|k+1}^\ell(\mathbf{x}_k|\mathbf{x}_{k+1}) := \int q_{k|\ell_k,k+1}(\mathbf{x}_k|\mathbf{x}_{\ell_k},\mathbf{x}_{k+1})\hat{\pi}_{\ell_k|k+1}^\theta(\mathbf{x}_{\ell_k}|\mathbf{x}_{k+1})\,\mathrm{d}\mathbf{x}_{\ell_k}\,, \tag{3.5}$$

where $\hat{\pi}_{\ell_k|k+1}^\theta(\mathbf{x}_{\ell_k}|\mathbf{x}_{k+1}) \propto \hat{p}_{\ell_k}^\theta(\mathbf{y}|\mathbf{x}_{\ell_k})p_{\ell_k|k+1}^\theta(\mathbf{x}_{\ell_k}|\mathbf{x}_{k+1})$ with $\hat{p}_{\ell_k}^\theta(\mathbf{y}|\mathbf{x}_{\ell_k}) := p(\mathbf{y}|\mathbf{m}_{0|\ell_k}^\theta(\mathbf{x}_{\ell_k}))$ and $p_{\ell_k|k+1}^\theta(\mathbf{x}_{\ell_k}|\mathbf{x}_{k+1}) := q_{\ell_k|0,k+1}(\mathbf{x}_{\ell_k}|\mathbf{m}_{0|k+1}^\theta(\mathbf{x}_{k+1}),\mathbf{x}_{k+1})$. Finally, for any sequence $\ell := (\ell_k)_{k=1}^n$ satisfying $\ell_k \leq k$, we define our surrogate model for $\pi_{0:n}$ and the posterior (2.1) as

$$\hat{\pi}_{0:n}^\ell(\mathbf{x}_{0:n}) := \hat{\pi}_n(\mathbf{x}_n)\prod_{k=0}^{n-1}\hat{\pi}_{k|k+1}^\ell(\mathbf{x}_k|\mathbf{x}_{k+1})\,, \quad \hat{\pi}_0^\ell(\mathbf{x}_0) := \int \hat{\pi}_{0:n}^\ell(\mathbf{x}_{0:n})\,\mathrm{d}\mathbf{x}_{1:n}\,, \tag{3.6}$$

where $\hat\pi_n(\mathbf{x}_n) = \mathcal{N}(\mathbf{x}_n; 0_d, \mathbf{I}_d)$ and $\hat\pi_{0|1}^\ell(\cdot|\mathbf{x}_1) := \delta_{\mathbf{m}_{0|1}^\theta(\mathbf{x}_1)}$ (similarly to Ho et al., 2020; Song et al., 2021a). Since the bridge kernel $q_{k|\ell_k,k+1}$ is Gaussian, it can be easily sampled. Hence, we only need to focus on the approximate sampling from $\pi_{\ell_k|k+1}$. Moreover, since the decomposition (3.4) is valid for any $\ell_k$, it can be chosen to better balance the approximation errors, as discussed above.

**Choice of the sequence $(\ell_k)_{k=1}^{n-1}$.** The accuracy of the surrogate model (3.6) ultimately depends on the design of the sequence $(\ell_k)_{k=1}^{n-1}$. In fact, for a given $k$, note that a decrease in $\ell_k$ ensures that the approximation $\hat p_{\ell_k}^\theta(\mathbf{y}|\mathbf{x}_{\ell_k})$ of $p_{\ell_k}(\mathbf{y}|\mathbf{x}_{\ell_k})$ becomes more accurate. For instance, setting $\ell_k = 0$ eliminates any error of the approximate like-
lihood (since $p_0(\mathbf{y}|\cdot) = p(\mathbf{y}|\cdot)$); how-
ever, decreasing $\ell_k$ can cause the distribution
$\pi_{0|k+1}(\cdot|\mathbf{x}_{k+1})$ to become strongly multimodal,
especially in the early stages of the diffusion pro-
cess, making the DDPM Gaussian approxima-
tion less accurate. Conversely, setting $\ell_k$ closer
to $k$, similar to the DPS approach of Chung et al.
(2023), improves the accuracy of the approxima-
tion of $q_{\ell_k|k+1}$, but impairs the approximation
$\hat p_{\ell_k}^\theta(\mathbf{y}|\mathbf{x}_{\ell_k})$ of $p_{\ell_k}(\mathbf{y}|\mathbf{x}_{\ell_k})$. The choice of $\ell_k$ is
therefore subject to a particular trade-off, which

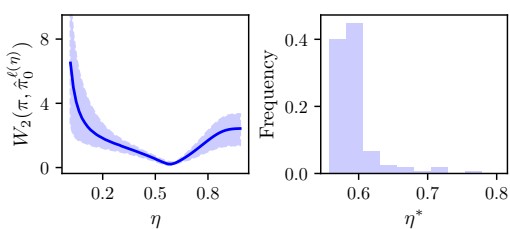

Figure 2: Left: average $W_2$ with 10%–90% quantile range. Right: distribution of the minimizing $\eta^*$.

we illustrate using the following Gaussian toy example. When considering the parameterization $\ell_k^\eta = \lfloor \eta k \rfloor$ with $\eta \in [0,1]$, we show that the Wasserstein-2 distance between $\pi$ and $\hat\pi_0^{\ell(\eta)}$ reaches its minimum for $\eta$ around $0.5$, which confirms that the additional flexibility obtained by introducing the midpoint can provide a surrogate model (3.6) with reduced approximation error.

**Example 3.2.** *Consider a linear inverse problem $Y = \mathbf{A}X + \sigma_\mathbf{y} Z$ with Gaussian noise and prior $q = \mathcal{N}(\mathbf{m}, \boldsymbol{\Sigma})$, where $\mathbf{m} \in \mathbb{R}^d$ and $\boldsymbol{\Sigma} \in \mathbb{R}^{d\times d}$ is positive definite. In this setting, we have access to the exact denoiser $\mathbf{m}_{0|k}$ in a closed form, which allows us to take $\mathbf{m}_{0|k}^\theta = \mathbf{m}_{0|k}$. Moreover, since the transition (3.5) admits an explicit expression in this case, we can quantify the approximation error of (3.6) w.r.t. $\pi$ for any sequence $(\ell_k)_{k=1}^{n-1}$. Note that in this case the true backward transitions $q_{\ell_k|k+1}$ are Gaussian and have the same mean as the Gaussian DDPM transitions, but differ in their covariance. All necessary derivations and computational details can be found in Appendix B. We consider sequences $\ell(\eta) := (\ell_k^\eta)_{k=1}^n$ with $\ell_k^\eta = \lfloor \eta k \rfloor$, $\eta \in [0,1]$, and compute the Wasserstein-2 distance $W_2(\pi, \hat\pi_0^{\ell(\eta)})$ as a function of $\eta$ on the basis of $500$ samples $(\mathbf{A}, \mathbf{m}, \boldsymbol{\Sigma})$ for $d = 100$. The left panel of Figure 2 displays the average error $W_2(\pi, \hat\pi_0^{\ell(\eta)})$ as a function of $\eta$, while the right panel displays the associated distribution of $\eta^* := argmin_{\eta\in[0,1]} W_2(\pi, \hat\pi_0^{\ell(\eta)})$. Figure 2 illustrates that the smallest approximation error for the surrogate model (3.6) is reached at intermediate values close to $\eta = 0.5$ rather than $\eta = 1$, which corresponds to a DPS-like approximation.*

**Variational approximation.** To sample from $\hat\pi_{0:n}^\ell$ in (3.6), we focus on simulating approximately and recursively (over $k$) the Markov chain with transition densities (3.5). Ideally, for $k = n$, we simply sample from $\hat\pi_n^\theta$; then, recursively, assuming that we have access to an approximate sample $X_{k+1}$ from $\hat\pi_{k+1}^\ell$, the next state $X_k$ is drawn from $\hat\pi_{k|k+1}^\ell(\cdot|X_{k+1})$. However, as $\hat\pi_{k|k+1}^\ell(\cdot|X_{k+1})$ is intractable, we propose using the Gaussian approximation that we specify next.

In the following, let $k \in [\![1, n]\!]$ and $\ell_k \in [\![0, k]\!]$ be fixed. For $\boldsymbol{\varphi} = (\hat{\boldsymbol{\mu}}_{\ell_k}, \hat{\boldsymbol{\rho}}_{\ell_k}) \in (\mathbb{R}^d)^2$, consider the Gaussian variational distribution

$$\lambda_{k|k+1}^{\boldsymbol{\varphi}}(\mathbf{x}_k|\mathbf{x}_{k+1}) := \int q_{k|\ell_k,k+1}(\mathbf{x}_k|\mathbf{x}_{\ell_k}, \mathbf{x}_{k+1}) \lambda_{\ell_k|k+1}^{\boldsymbol{\varphi}}(\mathbf{x}_{\ell_k}) \, \mathrm{d}\mathbf{x}_{\ell_k} , \tag{3.7}$$

$$\lambda_{\ell_k|k+1}^{\boldsymbol{\varphi}}(\mathbf{x}_{\ell_k}) := \mathcal{N}(\mathbf{x}_{\ell_k}; \hat{\boldsymbol{\mu}}_{\ell_k}, \mathrm{diag}(e^{2\hat{\boldsymbol{\rho}}_{\ell_k}})) . \tag{3.8}$$

In definition (3.8) the exponential function is applied element-wise to the vector $\hat{\boldsymbol{\rho}}_{\ell_k}$ and $\mathrm{diag}(e^{\hat{\boldsymbol{\rho}}_{\ell_k}})$ is the diagonal matrix with diagonal entries $e^{\hat{\boldsymbol{\rho}}_{\ell_k}}$. Based on the family $\{\lambda_{k|k+1}^{\boldsymbol{\varphi}} : \boldsymbol{\varphi} \in (\mathbb{R}^d)^2\}$ and an approximate sample $X_{k+1}$ from $\hat\pi_{k+1}^\ell$, we then seek the best fitting parameter $\boldsymbol{\varphi}$ that minimizes the upper bound on the backward KL divergence obtained using the data-processing inequality

$$\mathsf{KL}(\lambda_{k|k+1}^{\boldsymbol{\varphi}}(\cdot|X_{k+1}) \,\|\, \hat\pi_{k|k+1}^\ell(\cdot|X_{k+1})) \le \mathsf{KL}(\lambda_{\ell_k|k+1}^{\boldsymbol{\varphi}} \,\|\, \hat\pi_{\ell_k|k+1}^\theta(\cdot|X_{k+1})) =: \mathcal{L}_k(\boldsymbol{\varphi}; X_{k+1}) .$$

Here the gradient of the upper bound is given by

$$\nabla_{\boldsymbol{\varphi}} \mathcal{L}_k(\boldsymbol{\varphi}; X_{k+1}) = -\mathbb{E}\big[\nabla_{\boldsymbol{\varphi}} \log \hat p_{\ell_k}^\theta(\mathbf{y}|\hat{\boldsymbol{\mu}}_{\ell_k} + \mathrm{diag}(e^{\hat{\boldsymbol{\rho}}_{\ell_k}})Z)\big] + \nabla_{\boldsymbol{\varphi}} \mathsf{KL}(\lambda_{\ell_k|k+1}^{\boldsymbol{\varphi}} \,\|\, p_{\ell_k|k+1}^\theta(\cdot|X_{k+1})) ,$$

---

**Algorithm 1** MIDPOINT GUIDANCE POSTERIOR SAMPLING

1: **Input:** $(\ell_k)_{k=1}^n$ with $\ell_n = n$ and $\ell_1 = 1$; number $M$ of gradient steps.
2: $X_n \sim \mathcal{N}(0_d, \mathbf{I}_d)$, $\hat{X}_n \leftarrow X_n$
3: **for** $k = n - 1$ **to** 1 **do**
4: $\quad \hat{\boldsymbol{\mu}}_{\ell_k} \leftarrow \frac{\sqrt{\alpha_{\ell_k}}(1 - \alpha_{\ell_{k+1}}/\alpha_{\ell_k})}{1 - \alpha_{\ell_{k+1}}} \mathbf{m}_{0|\ell_{k+1}}^\theta(\hat{X}_{\ell_{k+1}}) + \frac{\sqrt{\alpha_{\ell_{k+1}}/\alpha_{\ell_k}}(1 - \alpha_{\ell_k})}{1 - \alpha_{\ell_{k+1}}} X_{k+1}$
5: $\quad \hat{\boldsymbol{\rho}}_{\ell_k} \leftarrow \frac{1}{2} \log \frac{(1 - \alpha_{\ell_{k+1}}/\alpha_{\ell_k})(1 - \alpha_{\ell_k})}{1 - \alpha_{\ell_{k+1}}}$
6: $\quad$ **for** $j = 1$ **to** $M$ **do**
7: $\quad\quad Z \sim \mathcal{N}(0_d, \mathbf{I}_d)$
8: $\quad\quad (\hat{\boldsymbol{\mu}}_{\ell_k}, \hat{\boldsymbol{\rho}}_{\ell_k}) \leftarrow \mathsf{OptimizerStep}(\nabla_{\boldsymbol{\varphi}} \widetilde{\mathcal{L}}_k(\cdot, Z; X_{k+1}); \hat{\boldsymbol{\mu}}_{\ell_k}, \hat{\boldsymbol{\rho}}_{\ell_k})$
9: $\quad$ **end for**
10: $\quad Z_{\ell_k}, Z_k \stackrel{\text{i.i.d.}}{\sim} \mathcal{N}(0_d, \mathbf{I}_d)$
11: $\quad \hat{X}_{\ell_k} \leftarrow \hat{\boldsymbol{\mu}}_{\ell_k} + \mathrm{diag}(e^{\hat{\boldsymbol{\rho}}_{\ell_k}}) Z_{\ell_k}$
12: $\quad X_k \sim q_{k|\ell_k, k+1}(\cdot | \hat{X}_{\ell_k}, X_{k+1})$ (See (A.5))
13: **end for**
14: $X_0 \leftarrow \mathbf{m}_{0|1}^\theta(X_1)$

---

where $Z \sim \mathrm{N}(0_d, \mathbf{I}_d)$ is independent of $X_{k+1}$. The first term is dealt with using the reparameterization trick (Kingma & Welling, 2014) and can be approximated using a Monte Carlo estimator based on a single sample $Z_{k+1}$; we denote this estimate by $\nabla_{\boldsymbol{\varphi}} \widetilde{\mathcal{L}}_k(\boldsymbol{\varphi}, Z_{k+1}; X_{k+1})$. The second term is the gradient of the KL divergence between two Gaussian distributions and can thus be computed in a closed form. Similarly to the previous approaches of Ho et al. (2022); Chung et al. (2023); Song et al. (2023a), our gradient estimator involves a vector-Jacobian product of the denoising network.

We now provide a summary of the MGPS algorithm, whose pseudocode is given in Algorithm 1. Given $(\ell_k)_{k=1}^{n-1}$, MGPS proceeds by simulating a Markov chain $(X_k)_{k=n}^0$ starting from $X_n \sim \mathcal{N}(0_d, \mathbf{I}_d)$. Recursively, given the state $X_{k+1}$, the state $X_k$ is obtained by

1. minimizing $\mathcal{L}_k(\cdot; X_{k+1})$ by performing $M$ stochastic gradient steps using the gradient estimator $\nabla_{\boldsymbol{\varphi}} \widetilde{\mathcal{L}}_k(\cdot; X_{k+1})$, yielding a parameter $\boldsymbol{\varphi}_k^*(X_{k+1})$,

2. sampling $\hat{X}_{\ell_k} \sim \lambda_{\ell_k|k+1}^{\boldsymbol{\varphi}_k^*}$, where we drop the dependence on $X_{k+1}$ in $\boldsymbol{\varphi}_k^*$, and then $X_k \sim q_{k|\ell_k, k+1}(\cdot | \hat{X}_{\ell_k}, X_{k+1})$.

**Remark 3.3.** *While conditionally on $\boldsymbol{\varphi}_k^*$ we have that $\hat{X}_{\ell_k} \sim \lambda_{\ell_k|k+1}^{\boldsymbol{\varphi}_k^*}$, the actual law of $\hat{X}_{\ell_k}$ given $X_{k+1}$ is not a Gaussian distribution as we need to marginalize over that of $\boldsymbol{\varphi}_k^*$ due to the randomness in the stochastic gradients.*

A well-chosen initialization for the Gaussian variational approximation parameters is crucial for achieving accurate fitting with few optimization steps, ensuring the overall runtime of MGPS remains competitive with existing methods. Given $\hat{X}_{\ell_{k+1}}$ sampled from $\lambda_{\ell_{k+1}|k+2}^{\boldsymbol{\varphi}_{k+1}^*}$ at the previous step, we choose the initial parameter $\boldsymbol{\varphi}_k$ so that

$$\lambda_{\ell_k|k+1}^{\boldsymbol{\varphi}_k}(\mathbf{x}_k) = q_{\ell_k|0, k+1}(\mathbf{x}_k | \mathbf{m}_{0|\ell_{k+1}}^\theta(\hat{X}_{\ell_{k+1}}), X_{k+1}). \quad (3.9)$$

This initialization is motivated by noting that $q_{\ell_k|0, k+1}(\cdot | \mathbf{m}_{0|k+1}^\pi(X_{k+1}), X_{k+1})$, the DDPM approximation of $\pi_{\ell_k|k+1}(\cdot | X_{k+1})$, where $\mathbf{m}_{0|k+1}^\pi$ is a supposed denoiser for the posterior, is a reasonable candidate for the initialization. As it is intractable, we replace it with $\mathbf{m}_{0|\ell_{k+1}}^\theta(\hat{X}_{\ell_{k+1}})$, our best current guess. Finally, in Appendix C.2 we devise a warm-start approach that improves the initialization during the early steps of the algorithm. This has proved advantageous in challenging problems.

**Related works.** We now discuss existing works that have similarities with our algorithm, MGPS. In essence, our method tries to reduce the approximation error incurred by DPS-like approximations. This has also been the focus of many works in the literature, which we review below.

When $p(\mathbf{y}|\cdot)$ is a likelihood associated to a linear inverse problem with Gaussian noise, Finzi et al. (2023); Stevens et al. (2023); Boys et al. (2023) leverage the fact that the Gaussian projection

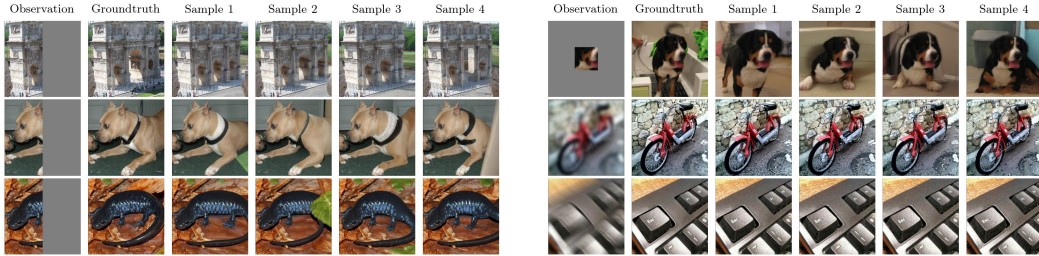

Figure 3: MGPS sample images for half mask (left), expand task, Gaussian blur and motion blur (right) on the `ImageNet` dataset.

of $q_{0|k}(\cdot|\mathbf{x}_k)$ minimizing the forward KL divergence can be approximated using the pre-trained denoisers; see Meng et al. (2021, Theorem 1). In the considered Gaussian setting, the likelihood $p(\mathbf{y}|\cdot)$ can be exactly integrated against the estimated Gaussian projection. However, this involves a matrix inversion that may be prohibitively expensive. Boys et al. (2023) circumvent the latter by using a diagonal approximation which involves the computation of $d$ vector-Jacobian products. For general likelihoods $p(\mathbf{y}|\cdot)$, Song et al. (2023b) use the Gaussian approximations of Ho et al. (2022); Song et al. (2023a) to estimate $p_k(\mathbf{y}|\cdot)$ for non-linear likelihoods $p(\mathbf{y}|\cdot)$ using a vanilla Monte Carlo estimate. Zhang et al. (2023) also consider the surrogate transitions (3.5) with $\ell_k = k$, and, given an approximate sample $X_{k+1}$ from $\hat{\pi}_{k+1}^{\ell}$, estimate the next sample $X_k$ maximizing $\mathbf{x}_k \mapsto \hat{\pi}_{k|k+1}^{\theta}(\mathbf{x}_k|X_{k+1})$ using gradient ascent. They also consider improved estimates of the likelihood $p_k(\mathbf{y}|\cdot)$ running a few diffusion steps. However, it is shown (Zhang et al., 2023, last subtable in Table 2) that this brings minor improvements at the expense of a sharp increase in computational cost, which is mainly due to backpropagation over the denoisers. See also Appendix C.3 for a discussion on how our method relates to the DPS (Chung et al., 2023) in the case where $\ell_k = k$.

Finally, a second line of work considers the distribution path $(\hat{\pi}_k)_{k=n}^0$, where $\hat{\pi}_k(\mathbf{x}_k) \propto p(\mathbf{y}|\mathbf{m}_{0|k}(\mathbf{x}_k))q_k(\mathbf{x}_k)$ for $k \in [\![0, n-1]\!]$ and $\hat{\pi}_n = \mathcal{N}(0_d, \mathbf{I}_d)$. This path bridges the Gaussian distribution and the posterior of interest. Furthermore, if one is able to accurately sample from $\hat{\pi}_{k+1}$, then these samples can be used to initialize a sampler targeting the next distribution $\hat{\pi}_k$. As these distributions are expected to be close, the sampling from $\hat{\pi}_k$ can also be expected to be accurate. Repeating this process yields approximate samples from $\pi$ *regardless* of the approximation error in the likelihoods. This approach is pursued by Rozet & Louppe (2023), who combines the update (2.5) with a Langevin dynamics targeting $\hat{\pi}_k$. Wu et al. (2023) use instead sequential Monte Carlo to recursively build empirical approximations of each $\hat{\pi}_k$ by evolving a sample of $N$ particles.

## 4 EXPERIMENTS

We now evaluate our algorithm on three different problems and compare it with several competitors. We begin in benchmarking our method on toy Gaussian-mixture targets and image experiments with both pixel space and latent diffusion models. We review the latter in Appendix A.2. Finally, we perform inpainting experiments on ECG data. For experiments based on pixel-space diffusion models, MGPS is benchmarked against state-of-the art methods in the literature: DPS (Chung et al., 2023), PGDM (Song et al., 2023a), DDNM (Wang et al., 2023), DIFFPIR (Zhu et al., 2023), and REDDIFF (Mardani et al., 2024). Regarding experiments using latent diffusion models, we compare against PSLD (Rout et al., 2024) and RESAMPLE (Song et al., 2024). Full details on the hyperparameters can be found in Appendix D.1. Our code for reproducing all the experiments is publicly available.[1]

**Gaussian mixture.** We first evaluate the accuracy of our method and the impact of the hyperparameters on a toy linear inverse problem with Gaussian mixture (GM) as $q$ and a Gaussian likelihood $p(\mathbf{y}|\mathbf{x}) := \mathcal{N}(\mathbf{y}; \mathbf{A}\mathbf{x}, \sigma_{\mathbf{y}}^2 \mathbf{I}_d)$, where $\mathbf{A} \in \mathbb{R}^{d_{\mathbf{y}} \times d}$ and $\sigma_{\mathbf{y}} = 0.05$. We repeat the experiment of Cardoso et al. (2023, App B.3), where the prior is a GM with 25 well-separated strata. For this specific example, both the posterior and the denoiser $\mathbf{m}_{0|k}$ are available in a closed form. In particular, the posterior can be shown to be a GM. The full details are provided in Appendix D.3. We consider the settings $(20, 1)$ and $(200, 1)$ for the dimensions $(d, d_{\mathbf{y}})$. For each method, we reconstruct 1000

---
[1]Code available at `https://github.com/yazidjanati/mgps`

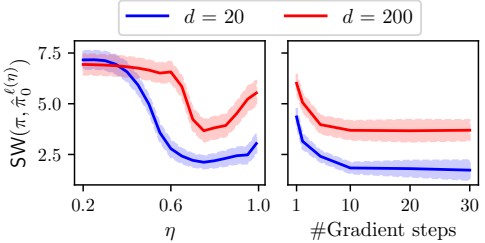

Figure 4: Left: SW as a function of $\eta$ with $\ell_k = \lfloor \eta k \rfloor$. Right: SW as a function of the number of gradient steps, for a specific choice of $(\ell_k)_k$.

Table 1: 95 % confidence interval for the SW on the GM experiment.

|  | $d = 20, d_{\mathbf{y}} = 1$ | $d = 200, d_{\mathbf{y}} = 1$ |
|---|---|---|
| MGPS | $\mathbf{2.11} \pm 0.30$ | $\mathbf{3.66} \pm 0.53$ |
| DPS | $8.93 \pm 0.49$ | $9.15 \pm 0.44$ |
| PGDM | $\underline{2.44} \pm 0.36$ | $\underline{5.23} \pm 0.38$ |
| DDNM | $4.24 \pm 0.37$ | $7.10 \pm 0.50$ |
| DIFFPIR | $4.14 \pm 0.42$ | $8.43 \pm 0.92$ |
| REDDIFF | $6.70 \pm 0.45$ | $8.35 \pm 0.39$ |

samples and then compute the sliced Wasserstein (SW) distance to exact samples from the posterior distribution. We repeat this procedure 100 times with randomly drawn matrices $\mathbf{A}$, and consider the resulting averaged SW distance. In every replication, the observation is generated as $Y = \mathbf{A}X + \sigma_{\mathbf{y}}Z$ where $Z \sim \mathcal{N}(0_{d_{\mathbf{y}}}, \mathbf{I}_{d_{\mathbf{y}}})$ and $X \sim q$. The results are reported in Table 1. It can be observed that MGPS with $\eta = 0.75$ outperforms all baselines in both settings. Although we have tuned the parameters of DPS it still exhibits considerable instability and often diverges. To account for these instabilities we set the SW to 10 when it diverges.

*Ablations.*   Next, we perform an ablation study on the total number of gradient steps and the choice of the sequence $(\ell_k)_{k=1}^{n-1}$. The right-hand plot in Figure 4 shows the average SW distance as a function of the number of gradient steps per denoising step when using $\ell_k = \lfloor 3k/4 \rfloor$. For $d = 20$, we observe a monotonic decrease of the SW distance. For $d = 200$, the SW distance decreases and then stabilizes after 10 gradient steps, indicating that in this case, MGPS performs well without requiring many additional gradient steps. We also report the average SW as a function of $\eta \in [0, 1]$ in the left-hand plot of Figure 4, following the same sequence choices as in Example 3.2, *i.e.*, $\ell_k^{\eta} = \lfloor \eta k \rfloor$. In both settings, the best SW is achieved at $\eta = 0.75$. In dimension $d = 200$, the SW at $\eta = 0.75$ is nearly twice as good as at $\eta = 1$, which corresponds to a DPS-like approximation. This demonstrates that the introduced trade-off leads to non-negligeable performance gains.

**Images.**   We evaluate our algorithm on a wide range of linear and nonlinear inverse problems with noise level $\sigma_{\mathbf{y}} = 0.05$. As linear problems we consider: image inpainting with a box mask of shape $150 \times 150$ covering the center of the image and half mask covering its right-hand side; image Super Resolution (SR) with factors $\times 4$ and $\times 16$; image deblurring with Gaussian blur; motion blur with $61 \times 61$ kernel size. We use the same experimental setup as Chung et al. (2023, Section 4) for the last two tasks. Regarding the nonlinear inverse problems on which we benchmark our algorithm, we consider: JPEG dequantization with quality factor $2\%$; phase retrieval with oversampling factor 2; non-uniform deblurring emulated via the forward model of Tran et al. (2021); high dynamic range (HDR) following Mardani et al. (2024, Section 5.2). Since the JPEG operator is not differentiable, we instead use the differentiable JPEG framework from Shin & Song (2017). We note that only DPS and REDDIFF apply to nonlinear problems.

*Datasets and evaluation.*   We test our algorithm on the $256 \times 256$ versions of the FFHQ (Karras et al., 2019) and ImageNet (Deng et al., 2009) datasets using publicly available pre-trained DDMs. For FFHQ, we use the pixel-space DDM of Choi et al. (2021) and the LDM of Rombach et al. (2022) with the VQ4 autoencoder. For ImageNet, we use the model by Dhariwal & Nichol (2021). Due to computational constraints, we first evaluate MGPS and competitors on all tasks using 50 random images per dataset. We then focus on the 5 most challenging tasks, evaluating MGPS and the top 2 competitors on 1k images in Table 7. For all tasks, we report the LPIPS (Zhang et al., 2018) between the reference image and the reconstruction, averaged over 50 images. For the five tasks tested on 1k images, we also report the FID in Table 7. Although pixel-wise metrics like PSNR and SSIM are less informative for multimodal posterior distributions, they are available in Tables 7 to 10. For all the considered competitors, we use the hyperparameters proposed in their official implementations if available, otherwise we manually tune them to achieve the best reconstructions (implementation details in Appendix D.1). For phase retrieval, due to the task's inherent instability, we follow the approach of Chung et al. (2023) by selecting the best reconstruction out of four; further discussion on this can be found in Chung et al. (2023, Appendix C.6).

Table 2: Mean LPIPS for various linear and nonlinear imaging tasks on the `FFHQ` and `ImageNet` $256 \times 256$ datasets with $\sigma_{\mathbf{y}} = 0.05$. Lower metrics are better.

| | **FFHQ** | | | | | | **ImageNet** | | | | | |
|---|---|---|---|---|---|---|---|---|---|---|---|---|
| Task | MGPS | DPS | PGDM | DDNM | DiffPir | RedDiff | MGPS | DPS | PGDM | DDNM | DiffPir | RedDiff |
| SR (×4) | **0.09** | **0.09** | 0.33 | 0.14 | 0.13 | 0.36 | **0.30** | 0.41 | 0.78 | 0.34 | 0.36 | 0.56 |
| SR (×16) | 0.26 | **0.24** | 0.44 | 0.30 | 0.28 | 0.51 | 0.53 | **0.50** | 0.60 | 0.70 | 0.63 | 0.83 |
| Box inpainting | **0.10** | 0.19 | 0.17 | 0.12 | 0.18 | 0.19 | **0.22** | 0.34 | 0.29 | 0.28 | 0.28 | 0.36 |
| Half mask | **0.20** | 0.24 | 0.26 | 0.22 | 0.23 | 0.28 | **0.29** | 0.44 | 0.38 | 0.38 | 0.35 | 0.44 |
| Gaussian Deblur | 0.15 | 0.16 | 0.87 | 0.19 | **0.12** | 0.23 | 0.32 | 0.35 | 1.00 | 0.45 | **0.29** | 0.54 |
| Motion Deblur | **0.13** | 0.16 | – | – | – | 0.21 | **0.22** | 0.39 | – | – | – | 0.40 |
| JPEG (QF = 2) | **0.16** | 0.39 | 1.10 | – | – | 0.32 | **0.42** | 0.63 | 1.31 | – | – | 0.51 |
| Phase retrieval | **0.11** | 0.46 | – | – | – | 0.25 | **0.47** | 0.62 | – | – | – | 0.60 |
| Nonlinear deblur | **0.23** | 0.52 | – | – | – | 0.66 | **0.44** | 0.88 | – | – | – | 0.67 |
| High dynamic range | **0.07** | 0.49 | – | – | – | 0.20 | **0.10** | 0.85 | – | – | – | 0.21 |

*Results.* We report the results in Table 2 for `FFHQ` and `ImageNet` with DDM prior, and Table 3 for `FFHQ` with LDM prior. These reveal that MGPS consistently outperforms the competing algorithms across both linear and nonlinear problems. Notably, on nonlinear tasks, MGPS achieves up to a twofold reduction in LPIPS compared to the other methods with DDM prior. It effectively manages the additional nonlinearities introduced by using LDMs and surpasses the state-of-the-art, as demonstrated in Table 3. Reconstructions obtained with MGPS are displayed in Figure 3 and Figure 5. Further examples and comparisons with other competitors are provided in Appendix D.6. It is seen that MGPS provides high quality reconstructions even for the most challenging examples. For the DDM prior, the results are obtained using the warm-start approach, see Algorithm 3, and setting $\ell_k = \lfloor k/2 \rfloor$ on all the tasks based on the `FFHQ` dataset. As for the `ImageNet` dataset, we use the same configuration on all the tasks, except for Gaussian deblur and motion deblur. For these tasks we found that using $\ell_k = \lfloor k/2 \rfloor \mathbb{1}_{k \geq \lfloor n/2 \rfloor} + k \mathbb{1}_{k < \lfloor n/2 \rfloor}$ improves performance. We use a similar strategy for the LDM prior. In Appendix D.2.1, we measure the runtime and GPU memory requirements for each algorithm. The memory requirement of our algorithm is the same as that of DPS and PGDM, but the runtime of MGPS with $n = 300$ is slightly larger. With fewer diffusion steps, the runtime is halved compared to DPS and PGDM and comparable to that of DiffPir, DDNM and RedDiff. Still, MGPS remains competitive and consistently outperforms the baselines, particularly on nonlinear tasks. See Table 8 and 9 for detailed results using $n \in \{50, 100\}$ diffusion steps.

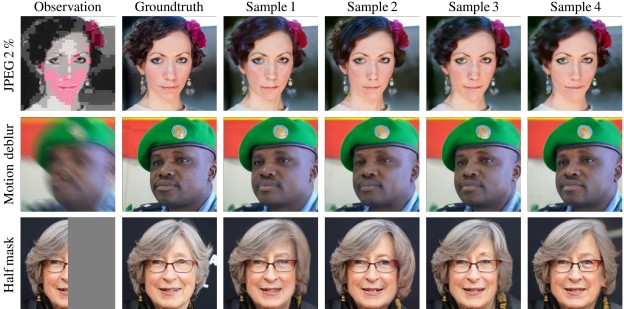

Figure 5: MGPS samples with LDM on `FFHQ` dataset.

Table 3: Mean LPIPS with LDM on `FFHQ`. Lower metrics are better.

| Task | MGPS | ReSample | PSLD |
|---|---|---|---|
| SR (×4) | **0.11** | 0.20 | 0.22 |
| SR (×16) | **0.30** | 0.36 | 0.35 |
| Box inpainting | **0.16** | 0.22 | 0.26 |
| Half mask | **0.25** | 0.30 | 0.31 |
| Gaussian Deblur | 0.16 | **0.15** | 0.35 |
| Motion Deblur | **0.18** | 0.19 | 0.41 |
| JPEG (QF = 2) | **0.20** | 0.26 | – |
| Phase retrieval | **0.34** | 0.41 | – |
| Nonlinear deblur | **0.26** | 0.30 | – |
| High dynamic range | **0.15** | **0.15** | – |

**ECG.** We now explore posterior sampling algorithms beyond image tasks to show their versatility and public health impact. Cardiovascular diseases cause one-third of global deaths, and better detection can improve management. Wearables like smartwatches can enhance diagnosis by capturing brief symptom episodes, particularly for conditions like paroxysmal atrial fibrillation (AF), which may go undetected during routine medical visits. However, they offer only a partial ECG view (Lead I instead of 12 leads) and a recent study showed the Apple Watch detected AF in only 34 of 90 episodes (Seshadri et al., 2020). To address this limitation, we propose using posterior sampling algorithms to reconstruct incomplete electrocardiograms (ECG). An ECG is an electrical recording of the heart's activity in which the signals generated by the heartbeats are recorded in order to diagnose various cardiac conditions such as cardiac arrhythmias and conduction abnormalities. Unlike static images, ECGs constitute complex time-series data consisting of 12 electrical signals acquired using 10 electrodes, 4 of which are attached to the limbs and record the 'limb leads', while the others record the 'precordial leads' around the heart. We study two conditional generation problems in ECGs. The first is a forecasting or missing-block (MB) reconstruction problem, where one half of the ECG is

reconstructed from the other; see fig. 8. This task evaluates the algorithm's ability to capture temporal information to predict a coherent signal. The second problem is an inpainting or missing-leads (ML) reconstruction, where the entire ECG is reconstructed from the lead I; see fig. 6. The question is whether we can capture the subtle information contained in lead I and reconstruct a coherent ECG with the same diagnosis as the real ECG. This task is challenging because lead I, being acquired with limb electrodes far from the heart, may contain very subtle features related to specific cardiac conditions. We train a state-space diffusion model (Goel et al., 2022) to generate ECGs using the 20k training ECGs from the PTB-XL dataset (Wagner et al., 2020), and benchmark the posterior sampling algorithm on the 2k test ECGs; see appendix D.5. We report the Mean Absolute Error (MAE) and the Root Mean Squared Error (RMSE) between ground-truth and reconstructions in Table 4. We demonstrate that a diffusion model trained to generate ECGs can serve as a prior to solve imputation tasks without additional training or fine-tuning, yielding superior results to a diffusion model trained conditionally on the MB task (Alcaraz & Strodthoff, 2022). The rationale for this result is discussed in Appendix D.5.2. We report in Table 5 the balanced accuracy of diagnosing Left Bundle Branch Block (LBBB), Right Bundle Branch Block (RBBB), Atrial Fibrillation (AF), and Sinus Bradycardia (SB) using the downstream classifier proposed in Strodthoff et al. (2020) (see appendix D.5.1) applied to both ground-truth and to reconstructed samples from lead I. See appendix D.5.2 for sample figures. MGPS outperforms all other posterior sampling algorithms with just 50 diffusion steps.

Table 4: MAE and RMSE for missing block task on the PTB-XL dataset.

| Metric | MGPS | DPS | PGDM | DDNM | DIFFPIR | REDDIFF | TRAINEDDIFF |
|---|---|---|---|---|---|---|---|
| MAE | $0.111 \pm 2e{-}3$ | $0.117 \pm 4e{-}3$ | $0.118 \pm 2e{-}3$ | $\mathbf{0.103 \pm 2e{-}3}$ | $0.115 \pm 2e{-}3$ | $0.171 \pm 3e{-}3$ | $0.116 \pm 2e{-}3$ |
| RMSE | $\mathbf{0.225 \pm 4e{-}3}$ | $0.232 \pm 4e{-}3$ | $0.233 \pm 4e{-}3$ | $\mathbf{0.224 \pm 4e{-}3}$ | $0.233 \pm 4e{-}3$ | $0.287 \pm 5e{-}3$ | $0.266 \pm 3e{-}3$ |

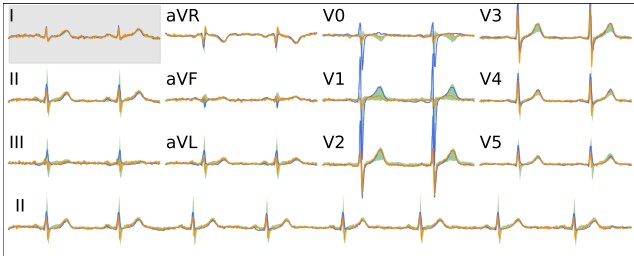

Figure 6: 10s ECG reconstruction from lead I. Ground-truth in blue, 10%–90% quantile range in green, random sample in orange.

Table 5: Balanced acc. downstream diagnosis from ECG reconstructed from lead I.

| Method | RBBB | LBBB | AF | SB |
|---|---|---|---|---|
| $\text{MGPS}_{50}$ | 0.81 | 0.92 | **0.94** | 0.66 |
| $\text{MGPS}_{300}$ | **0.90** | **0.93** | 0.92 | **0.66** |
| DPS | 0.54 | 0.84 | 0.79 | 0.50 |
| PGDM | 0.65 | 0.87 | 0.88 | 0.55 |
| DDNM | 0.71 | 0.83 | 0.86 | 0.59 |
| DIFFPIR | 0.57 | 0.80 | 0.77 | 0.53 |
| REDDIFF | 0.73 | 0.86 | 0.88 | 0.60 |
| Ground-truth | 0.99 | 0.98 | 0.94 | 0.70 |

## 5 CONCLUSION

We have introduced MGPS, a novel posterior sampling algorithm designed to solve general inverse problems using both diffusion models and latent diffusion models as a prior. Our approach is based on trading off the approximation error in the prior backward dynamics for a more accurate approximation of the guidance term. This strategy has proven to be effective in a variety of numerical experiments across various tasks. The results show that MGPS consistently performs competitively and often even superior to state-of-the-art methods.

**Limitations and future directions.** Although the proposed method is promising, it has certain limitations that invite further exploration. First, a detailed analysis of how the algorithm's performance depends on the choice of the intermediate time steps $(\ell_k)_{k=1}^{n-1}$ remains a challenging but critical subject for future research. Although our image and ECG experiments suggest that setting $\ell_k = \lfloor k/2 \rfloor$ leads to significant performance gains on most tasks, we have observed that using an adaptive sequence $\ell_k = \lfloor \eta_k k \rfloor$, where $\eta_k$ increases as $k$ decreases, further enhances results on certain tasks, such as Gaussian and motion deblurring, as well as when using latent diffusion models. A second direction for improvement lies in refining the approximation of $p_k(\mathbf{y}|\cdot)$ which we believe could lead to overall algorithmic improvements. Specifically, devising an approximation $\hat{p}_k^\theta(\mathbf{y}|\cdot)$ such that $\nabla \log \hat{p}_k^\theta(\mathbf{y}|\cdot)$ does not require a vector-Jacobian product could significantly reduce the algorithm's runtime. Lastly, we see potential in using the two-stage approximation introduced in our warm-start strategy (see Algorithm 3) at every diffusion step. Although this technique is promising, it currently leads to instabilities as $k$ decreases. Understanding and resolving these instabilities is also of key interest.

**Acknowledgements.** The work is supported by the Swedish Research Council, project 2024-05680. The work of Eric Moulines has been partly funded by the European Union (ERC-2022-SYG-OCEAN-101071601). Views and opinions expressed are however those of the authors only and do not necessarily reflect those of the European Union or the European Research Council Executive Agency. Neither the European Union nor the granting authority can be held responsible for them. We would like to thank IHU-LIRYC for the computing power made available to us. The work of Y.J. and B.M. has been supported by Technology Innovation Institute (TII), project Fed2Learn.

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

# A    BACKGROUND ON DENOISING DIFFUSION MODELS

## A.1    DENOISING DIFFUSION PROBABILISTIC MODELS

In this section, we provide further background on DDMs based on the DDPM framework (Ho et al., 2020; Dhariwal & Nichol, 2021; Song et al., 2021a). We rely on definitions provided in the main text.

DDPMs define generative models for $q$ relying only on parametric approximations $(\mathbf{m}_{0|t}^{\theta})_{t=1}^{T}$ of the denoisers $(\mathbf{m}_{0|t})_{t=1}^{T}$. These approximate denoisers are usually defined through the parameterization

$$\mathbf{m}_{0|t}^{\theta}(\mathbf{x}_t) = (\mathbf{x}_t - \sqrt{1 - \alpha_t}\boldsymbol{\epsilon}_t^{\theta}(\mathbf{x}_t))/\sqrt{\alpha_t} \tag{A.1}$$

and trained by minimizing the denoising loss

$$\sum_{t=1}^{T} w_t \mathbb{E}\left[\|\boldsymbol{\epsilon}_t - \boldsymbol{\epsilon}_t^{\theta}(\sqrt{\alpha_t}X_0 + \sqrt{1 - \alpha_t}\boldsymbol{\epsilon}_t)\|^2\right], \tag{A.2}$$

w.r.t. the neural network parameter $\theta$, where $(\boldsymbol{\epsilon}_t)_{t=1}^{T}$ are i.i.d. standard normal vectors, $X_0 \sim q$, and $(w_t)_{t=1}^{T}$ are some nonnegative weights. Having trained the denoisers, the generative model for $q$ is defined as follows. Let $(t_k)_{k=0}^{n}$ be an increasing sequence of time steps in $[\![0, T]\!]$ with $t_0 = 0$. We assume that $t_n$ is large enough so that $q_{t_n}(\mathbf{x}_{t_n}) = \int q(\mathbf{x}_0)q_{t_n|0}(\mathbf{x}_{t_n}|\mathbf{x}_0)\,\mathrm{d}\mathbf{x}_0$ is approximately the density of a multivariate standard normal distribution. For convenience, we assign the index $k$ to any quantity depending on $t_k$; e.g., we denote $q_{t_k}$ by $q_k$. Consider the backward decomposition

$$q_{0:n}(\mathbf{x}_{0:n}) = q(\mathbf{x}_0)\prod_{k=0}^{n-1} q_{k+1|k}(\mathbf{x}_{k+1}|\mathbf{x}_k) = q_n(\mathbf{x}_n)\prod_{k=0}^{n-1} q_{k|k+1}(\mathbf{x}_k|\mathbf{x}_{k+1})$$

of the forward process initialized at $q$, where $q_{k|k+1}(\mathbf{x}_k|\mathbf{x}_{k+1}) \propto q_k(\mathbf{x}_k)q_{k+1|k}(\mathbf{x}_{k+1}|\mathbf{x}_k)$. Next, for $(j, \ell, k) \in [\![0, n]\!]^3$ such that $j < \ell < k$, define

$$m_{\ell|j,k}(\mathbf{x}_j, \mathbf{x}_k) := \frac{\sqrt{\alpha_\ell/\alpha_j}(1 - \alpha_k/\alpha_\ell)}{1 - \alpha_k/\alpha_j}\mathbf{x}_j + \frac{\sqrt{\alpha_k/\alpha_\ell}(1 - \alpha_\ell/\alpha_j)}{1 - \alpha_k/\alpha_j}\mathbf{x}_k, \tag{A.3}$$

$$v_{\ell|j,k} := \frac{(1 - \alpha_\ell/\alpha_j)(1 - \alpha_k/\alpha_\ell)}{1 - \alpha_k/\alpha_j}. \tag{A.4}$$

Then the bridge kernel is

$$q_{\ell|j,k}(\mathbf{x}_\ell|\mathbf{x}_j, \mathbf{x}_k) = q_{\ell|j}(\mathbf{x}_\ell|\mathbf{x}_j)q_{k|\ell}(\mathbf{x}_k|\mathbf{x}_\ell)\big/q_{k|j}(\mathbf{x}_k|\mathbf{x}_j)$$
$$= \mathcal{N}(\mathbf{x}_\ell; m_{\ell|j,k}(\mathbf{x}_j, \mathbf{x}_k), v_{\ell|j,k}\mathbf{I}_d). \tag{A.5}$$

Using the bridge kernel, $q_{0:n}$ is approximated using the variational approximation

$$p_{0:n}^{\theta}(\mathbf{x}_{0:n}) = p_n^{\theta}(\mathbf{x}_n)\prod_{k=0}^{n-1} p_{k|k+1}^{\theta}(\mathbf{x}_k|\mathbf{x}_{k+1}),$$

where $p_{0|1}^{\theta}(\mathbf{x}_0|\mathbf{x}_1) = \mathcal{N}(\mathbf{x}_0; \mathbf{m}_{0|1}^{\theta}(\mathbf{x}_1), v_{0|1}\mathbf{I}_d)$, $v_{0|1}$ being a tunable parameter, and

$$p_{k|k+1}^{\theta}(\mathbf{x}_k|\mathbf{x}_{k+1}) := q_{k|0,k+1}(\mathbf{x}_k|\mathbf{m}_{0|k+1}^{\theta}(\mathbf{x}_{k+1}), \mathbf{x}_{k+1}) \quad k \in [\![1, n-1]\!]. \tag{A.6}$$

When $n = T$, the denoising objective (A.2) corresponds to the KL divergence $\mathsf{KL}(q_{0:n} \,\|\, p_{0:n}^{\theta})$ for a specific choice of weights $(w_t)_t$. In practice, a DDPM is trained using the objective (A.2) with large $T$ but at inference $n$ is usually much smaller.

## A.2    LATENT DIFFUSION MODELS

Latent diffusion models (LDM) (Rombach et al., 2022) define a DDM in a latent space. Let $\mathcal{E} : \mathbb{R}^d \to \mathbb{R}^p$ be an encoder function and $\mathcal{D} : \mathbb{R}^p \to \mathbb{R}^d$ a decoder function. We assume that these functions satisfy $\mathcal{D}_\sharp\mathcal{E}_\sharp q \approx q$, where, for instance, $\mathcal{E}_\sharp q$ denotes the law of the random variable $\mathcal{E}(X)$ where $X \sim q$. A LDM approximating $q$ is given by $\mathcal{D}_\sharp p_0^{\theta}$, where $p_0^{\theta}$ is a diffusion model trained on

samples from $\mathcal{E}_\sharp q$. Finally, when solving inverse problems with LDMs, we assume that the target distribution is instead

$$\pi(\mathbf{x}) \propto p(\mathbf{y}|\mathbf{x})\mathcal{D}_\sharp\mathcal{E}_\sharp q(\mathbf{x}).$$

Let $X \sim \pi$, $D \sim \mathcal{D}_\sharp\mathcal{E}_\sharp q$, $E \sim \mathcal{E}_\sharp q$, and $Z \sim \bar{\pi}$, where $\bar{\pi}(\mathbf{z}) \propto p(\mathbf{y}|\mathcal{D}_\sharp(\mathbf{z}))\mathcal{E}_\sharp q(\mathbf{z})$. For any bounded function $f$ on $\mathbb{R}^d$, we have, following the definition of $\pi$, that

$$\mathbb{E}[f(X)] = \frac{\mathbb{E}[p(\mathbf{y}|D)f(D)]}{\mathbb{E}[p(\mathbf{y}|D)]} = \frac{\mathbb{E}[p(\mathbf{y}|\mathcal{D}(E))f(\mathcal{D}(E))]}{\mathbb{E}[p(\mathbf{y}|\mathcal{D}(E))]} = \mathbb{E}[f(\mathcal{D}(Z))].$$

Hence $\mathsf{Law}(X) = \mathsf{Law}(\mathcal{D}(Z))$. As a result, to sample approximately from $\pi$, we first sample approximately from $\bar{\pi}$ using any diffusion posterior sampling algorithm with pre-trained DDM for $\mathcal{E}_\sharp q$, then decode the obtained samples using $\mathcal{D}$.

## A.3 MIDPOINT DECOMPOSITION

Before we proceed with the midpoint decomposition, we first recall that under the joint distribution obtained by initializing the posterior $\pi$ with the forward process (see (3.1)), it holds that for all $i < j$,

$$\pi_i(\mathbf{x}_i)q_{j|i}(\mathbf{x}_j|\mathbf{x}_i) = \pi_j(\mathbf{x}_j)\pi_{i|j}(\mathbf{x}_i|\mathbf{x}_j), \tag{A.7}$$

where $\pi_{i|j}(\mathbf{x}_i|\mathbf{x}_j) := \pi_i(\mathbf{x}_i)q_{j|i}(\mathbf{x}_j|\mathbf{x}_i)/\pi_j(\mathbf{x}_j)$ and integrates to one.

*Proof of Lemma 3.1.* Let $(\ell, k) \in [\![0, n]\!]$ be such that $\ell < k$. Applying repeatedly the definition of the bridge kernel (A.5) and the identity (A.7),

$$\begin{aligned}
q_{k|\ell,k+1}(\mathbf{x}_k|\mathbf{x}_\ell, \mathbf{x}_{k+1}) &= \frac{q_{k|\ell}(\mathbf{x}_k|\mathbf{x}_\ell)q_{k+1|k}(\mathbf{x}_{k+1}|\mathbf{x}_k)}{q_{k+1|\ell}(\mathbf{x}_{k+1}|\mathbf{x}_\ell)} \\
&= \frac{\pi_\ell(\mathbf{x}_\ell)q_{k|\ell}(\mathbf{x}_k|\mathbf{x}_\ell)q_{k+1|k}(\mathbf{x}_{k+1}|\mathbf{x}_k)}{\pi_\ell(\mathbf{x}_\ell)q_{k+1|\ell}(\mathbf{x}_{k+1}|\mathbf{x}_\ell)} \\
&= \frac{\pi_{\ell|k}(\mathbf{x}_\ell|\mathbf{x}_k)\pi_{k|k+1}(\mathbf{x}_k|\mathbf{x}_{k+1})\pi_{k+1}(\mathbf{x}_{k+1})}{\pi_{\ell|k+1}(\mathbf{x}_\ell|\mathbf{x}_{k+1})\pi_{k+1}(\mathbf{x}_{k+1})} \\
&= \frac{\pi_{\ell|k}(\mathbf{x}_\ell|\mathbf{x}_k)\pi_{k|k+1}(\mathbf{x}_k|\mathbf{x}_{k+1})}{\pi_{\ell|k+1}(\mathbf{x}_\ell|\mathbf{x}_{k+1})}.
\end{aligned}$$

It then follows that

$$\begin{aligned}
\pi_{k|k+1}(\mathbf{x}_k|\mathbf{x}_{k+1}) &= \int \pi_{\ell|k}(\mathbf{x}_\ell|\mathbf{x}_k)\pi_{k|k+1}(\mathbf{x}_k|\mathbf{x}_{k+1}) \, \mathrm{d}\mathbf{x}_\ell \\
&= \int q_{k|\ell,k+1}(\mathbf{x}_k|\mathbf{x}_\ell, \mathbf{x}_{k+1})\pi_{\ell|k+1}(\mathbf{x}_\ell|\mathbf{x}_{k+1}) \, \mathrm{d}\mathbf{x}_\ell.
\end{aligned}$$

$\square$

# B THE GAUSSIAN CASE

## B.1 DERIVATION

In this section we derive the recursions verified by the first and second moments of the marginal distribution $\hat{\pi}_0^\ell$ of the surrogate model (3.6) in the simplified setting of Example 3.2. We recall that in this specific example we assume that $q = \mathrm{N}(\boldsymbol{m}, \boldsymbol{\Sigma})$ where $(\boldsymbol{m}, \boldsymbol{\Sigma}) \in \mathbb{R}^d \times \mathcal{S}_d^{++}$ and $p(\mathbf{y}|\cdot) : \mathbf{x} \mapsto \mathcal{N}(\mathbf{y}; \mathbf{A}\mathbf{x}, \sigma_\mathbf{y}^2\mathbf{I}_{d_\mathbf{y}})$.

**Denoiser and DDPM transitions.** Since we are dealing with a Gaussian prior, the denoiser $\mathbf{m}_{0|k}$ can be computed in closed form for any $k \in [\![1, n]\!]$. Using Bishop (2006, Eqn. 2.116), we have that

$$\begin{aligned}
q_{0|k}(\mathbf{x}_0|\mathbf{x}_k) &\propto q(\mathbf{x}_0)q_{k|0}(\mathbf{x}_k|\mathbf{x}_0) \\
&= \mathcal{N}\left(\mathbf{x}_0; \boldsymbol{\Sigma}_{0|k}\left((\sqrt{\alpha_k}/v_k)\mathbf{x}_k + \boldsymbol{\Sigma}^{-1}\boldsymbol{m}\right), \boldsymbol{\Sigma}_{0|k}\right),
\end{aligned}$$

where $\boldsymbol{\Sigma}_{0|k} := ((\alpha_k/v_k)\mathbf{I} + \boldsymbol{\Sigma}^{-1})^{-1}$. Hence,

$$\mathbf{m}_{0|k}(\mathbf{x}_k) = \boldsymbol{\Sigma}_{0|k}\big((\sqrt{\alpha_k}/v_k)\mathbf{x}_k + \boldsymbol{\Sigma}^{-1}\boldsymbol{m}\big),$$

and we assume in the remainder of this section that $\mathbf{m}_{0|k}^\theta = \mathbf{m}_{0|k}$. From the expression of $q_{0|k}(\cdot|\mathbf{x}_k)$ we can immediately derive the more general backward transitions $q_{\ell|k}(\cdot|\mathbf{x}_k)$ for $\ell \in [\![1, k-1]\!]$ by noting that

$$q_{\ell|k}(\mathbf{x}_\ell|\mathbf{x}_k) = \int q_{\ell|0,k}(\mathbf{x}_\ell|\mathbf{x}_0, \mathbf{x}_k) q_{0|k}(\mathbf{x}_0|\mathbf{x}_k)\,\mathrm{d}\mathbf{x}_0\,.$$

From this, (A.3), (A.4), and the law of total expectation and covariance, it follows that $q_{\ell|k}(\cdot|\mathbf{x}_k) = \mathrm{N}(\boldsymbol{m}_{\ell|k}(\mathbf{x}_k), \boldsymbol{\Sigma}_{\ell|k}(\mathbf{x}_k))$, where

$$\boldsymbol{m}_{\ell|k}(\mathbf{x}_k) = m_{\ell|0,k}(\mathbf{m}_{0|k}(\mathbf{x}_k), \mathbf{x}_k)\,, \quad \boldsymbol{\Sigma}_{\ell|k} = \frac{\alpha_\ell(1 - \alpha_k/\alpha_\ell)^2}{(1 - \alpha_k)^2}\boldsymbol{\Sigma}_{0|k} + v_{\ell|0,k}\mathbf{I}_d\,.$$

On the other hand, the DDPM transitions are

$$p_{\ell|k}^\theta(\cdot|\mathbf{x}_k) = q_{\ell|0,k}(\cdot|\mathbf{m}_{0|k}(\mathbf{x}_k), \mathbf{x}_k) = \mathrm{N}(m_{\ell|0,k}(\mathbf{m}_{0|k}(\mathbf{x}_k), \mathbf{x}_k), v_{\ell|0,k}\mathbf{I}_d), \tag{B.1}$$

which shows that in this case, the true transitions and approximate ones differ only by their covariance.

**Moments recursion.** For $k \in [\![0, n]\!]$, we let $\hat{\pi}_k^\ell$ denote the $\mathbf{x}_k$ marginal of the surrogate model (3.6). We remind the reader that $\hat{\pi}_n^\ell = \mathrm{N}(0_d, \mathbf{I}_d)$. The marginals satisfy the recursion

$$\hat{\pi}_k^\ell(\mathbf{x}_k) = \int \hat{\pi}_{k|k+1}^\ell(\mathbf{x}_k|\mathbf{x}_{k+1})\hat{\pi}_{k+1}^\ell(\mathbf{x}_{k+1})\,\mathrm{d}\mathbf{x}_{k+1}\,, \quad k \in [\![0, n-1]\!]\,.$$

Since $\hat{p}_k^\theta(\mathbf{y}|\mathbf{x}_k) = \mathcal{N}(\mathbf{y}; \mathbf{A}\mathbf{m}_{0|k}^\theta(\mathbf{x}_k), \sigma_{\mathbf{y}}^2\mathbf{I}_{d_y})$ and $\mathbf{m}_{0|k}^\theta$ is linear in $\mathbf{x}_k$, it is easily seen that $\hat{\pi}_{\ell_k|k+1}^\theta(\cdot|\mathbf{x}_{k+1})$ is the density of a Gaussian distribution. Consequently, by definition (3.5) and the definition (A.5) of the bridge kernel, this is also the case for $\hat{\pi}_{k|k+1}^\ell(\cdot|\mathbf{x}_{k+1})$. Now assume that

$$\hat{\pi}_{k+1}^\ell(\mathbf{x}_{k+1}) = \mathcal{N}(\mathbf{x}_{k+1}; \hat{\boldsymbol{\mu}}_{k+1}^\ell, \hat{\boldsymbol{\Sigma}}_{k+1}^\ell)\,, \tag{B.2}$$

$$\hat{\pi}_{k|k+1}^\ell(\mathbf{x}_k|\mathbf{x}_{k+1}) = \mathcal{N}(\mathbf{x}_k; \mathbf{M}_{k|k+1}^\ell\mathbf{x}_{k+1} + \boldsymbol{c}_{k|k+1}^\ell, \hat{\boldsymbol{\Sigma}}_{k|k+1}^\ell)\,, \tag{B.3}$$

where $\mathbf{M}_{k|k+1}^\ell \in \mathbb{R}^{d\times d}$, $\hat{\boldsymbol{\Sigma}}_{k|k+1}^\ell \in \mathcal{S}_d^{++}$, and $\boldsymbol{c}_{k|k+1}^\ell \in \mathbb{R}^d$. Using the definition (A.5) of the bridge kernel we find that $\hat{\pi}_k^\ell = \mathrm{N}(\hat{\boldsymbol{\mu}}_k, \hat{\boldsymbol{\Sigma}}_k)$, where

$$\hat{\boldsymbol{\mu}}_k^\ell = \mathbf{M}_{k|k+1}^\ell\hat{\boldsymbol{\mu}}_{k+1}^\ell + \boldsymbol{c}_{k|k+1}^\ell\,,$$

$$\hat{\boldsymbol{\Sigma}}_k^\ell = \mathbf{M}_{k|k+1}^\ell\hat{\boldsymbol{\Sigma}}_{k+1}^\ell\mathbf{M}_{k|k+1}^{\mathsf{T}} + \hat{\boldsymbol{\Sigma}}_{k|k+1}^\ell\,.$$

Iterating these updates until reaching $k = 0$, starting from the initialization $\hat{\boldsymbol{\mu}}_n^\ell = 0_d$ and $\hat{\boldsymbol{\Sigma}}_n^\ell = \mathbf{I}_d$, yields the desired moments of the surrogate posterior $\pi_0^\ell$. It now remains to show that the backward transition $\hat{\pi}_{k|k+1}^\ell(\cdot|\mathbf{x}_{k+1})$ writes in the form (B.3) and identify $\mathbf{M}_{k|k+1}^\ell$ and $\boldsymbol{c}_{k|k+1}^\ell$.

First, we write the approximate likelihood in the form

$$\hat{p}_k^\theta(\mathbf{y}|\mathbf{x}_k) = \mathcal{N}(\mathbf{y}; \hat{\mathbf{A}}_k\mathbf{x}_k + \boldsymbol{b}_k, \sigma_{\mathbf{y}}^2\mathbf{I}_{d_y}),$$

where

$$\hat{\mathbf{A}}_k = (\sqrt{\alpha_k}/v_k)\mathbf{A}\boldsymbol{\Sigma}_{0|k}\,, \quad \boldsymbol{b}_k = \mathbf{A}\boldsymbol{\Sigma}_{0|k}\boldsymbol{\Sigma}^{-1}\boldsymbol{m}\,.$$

We also denote by $\boldsymbol{m}_{\ell|k}^\theta(\mathbf{x}_k)$ the mean of the Gaussian distribution with density given by the DDPM transition $p_{\ell|k}^\theta(\cdot|\mathbf{x}_k)$ in (B.1). We have that

$$\boldsymbol{m}_{\ell_k|k+1}^\theta(\mathbf{x}_{k+1}) = \mathbf{H}_{\ell_k|k+1}\mathbf{x}_{k+1} + \boldsymbol{h}_{\ell_k|k+1}\,,$$

where

$$\mathbf{H}_{\ell_k|k+1} := \frac{\alpha_{\ell_k}(1 - \alpha_{k+1}/\alpha_{\ell_k})}{v_{\ell_k}v_{k+1}}\boldsymbol{\Sigma}_{0|k} + \frac{\sqrt{\alpha_{k+1}}(1 - \alpha_{\ell_k})}{v_{k+1}}\mathbf{I}_d\,, \tag{B.4}$$

$$\boldsymbol{h}_{\ell_k|k+1} := \frac{\sqrt{\alpha_{\ell_k}}(1 - \alpha_{k+1}/\alpha_{\ell_k})}{v_{k+1}}\boldsymbol{\Sigma}_{0|k}\boldsymbol{\Sigma}^{-1}\boldsymbol{m}\,. \tag{B.5}$$

Then, applying (Bishop, 2006, Eqn. 2.116), we get

$$\hat{\pi}^{\theta}_{\ell_k|k+1}(\mathbf{x}_{\ell_k}|\mathbf{x}_{k+1}) = \mathcal{N}(\mathbf{x}_{\ell_k}; \widetilde{\mathbf{M}}^{\theta}_{\ell_k|k+1}\mathbf{x}_{k+1} + \tilde{\boldsymbol{c}}^{\theta}_{\ell_k|k+1}, \boldsymbol{\Gamma}_{\ell_k|k+1}),$$

where

$$\boldsymbol{\Gamma}_{\ell_k|k+1} := \left(v^{-1}_{\ell_k|0,k+1}\mathbf{I} + \sigma^{-2}_{\mathbf{y}}\hat{\mathbf{A}}^{\top}_{\ell_k}\hat{\mathbf{A}}_{\ell_k}\right)^{-1},$$

$$\widetilde{\mathbf{M}}^{\theta}_{\ell_k|k+1} := v^{-1}_{\ell_k|0,k+1}\boldsymbol{\Gamma}_{\ell_k|k+1}\mathbf{H}_{\ell_k|k+1},$$

$$\tilde{\boldsymbol{c}}^{\theta}_{\ell_k|k+1} := \boldsymbol{\Gamma}_{\ell_k|k+1}\left[\sigma^{-2}_{\mathbf{y}}\hat{\mathbf{A}}^{\top}_{\ell_k}(\mathbf{y} - \boldsymbol{b}_{\ell_k}) + v^{-1}_{\ell_k|0,k+1}\boldsymbol{h}_{\ell_k|k+1}\right].$$

Finally, following (3.5) we find that

$$\mathbf{M}^{\ell}_{k|k+1} = \frac{\sqrt{\alpha_k/\alpha_{\ell_k}}(1 - \alpha_{k+1}/\alpha_k)}{1 - \alpha_{k+1}/\alpha_{\ell_k}}\widetilde{\mathbf{M}}^{\theta}_{\ell_k|k+1} + \frac{\sqrt{\alpha_{k+1}/\alpha_k}(1 - \alpha_k/\alpha_{\ell_k})}{1 - \alpha_{k+1}/\alpha_{\ell_k}}\mathbf{I}_d,$$

$$\boldsymbol{c}^{\ell}_{k|k+1} = \frac{\sqrt{\alpha_k/\alpha_{\ell_k}}(1 - \alpha_{k+1}/\alpha_k)}{1 - \alpha_{k+1}/\alpha_{\ell_k}}\tilde{\boldsymbol{c}}^{\theta}_{\ell_k|k+1}.$$

## B.2 EXPERIMENTAL SETUP

In the experiment described in Example 3.2, we set $d = 100$ and generate $500$ instances of inverse problems $(\mathbf{A}, \boldsymbol{m}, \boldsymbol{\Sigma})$. For each instance, we compute the Wasserstein-2 distance between the resulting posterior distribution $\pi$ and $\hat{\pi}^{\ell(\eta)}_0$.

**Prior.** The mean of the prior is sampled from a standard Gaussian distribution. To generate the covariance $\boldsymbol{\Sigma} \in \mathcal{S}^{++}_d$, we first draw a matrix $\mathbf{G} \in \mathbb{R}^{d \times d}$ with i.i.d. entries sampled from $\mathrm{N}(0, 1)$. Then, for a better conditioning of $\boldsymbol{\Sigma}$, we normalize the columns of $\mathbf{G}$ and set $\boldsymbol{\Sigma} = \bar{\lambda}^2\mathbf{I} + \mathbf{G}\mathbf{G}^{\top}$, where $\bar{\lambda}^2$ is the mean of the squared singular values of $\mathbf{G}$.

**Likelihood.** To sample an ill-posed problem, we generate a rank-deficient matrix $\mathbf{A} \in \mathbb{R}^{d_{\mathbf{y}} \times d}$ with $d_{\mathbf{y}} \leq d$. We sample uniformly $d_{\mathbf{y}}$ from the interval $[\![d/10, d]\!]$ and draw the entries of $\mathbf{A}$ i.i.d. from $\mathrm{N}(0, 1)$. Regarding $\sigma_{\mathbf{y}}$, we sample it uniformly from the interval $[0.1, 0.5]$.

Finally, the resulting posterior is also Gaussian (Bishop, 2006, Eqn. 2.116)

$$\pi(\mathbf{x}) \propto p(\mathbf{y}|\mathbf{x})p(\mathbf{x}) \propto \mathcal{N}(\mathbf{x}; \boldsymbol{m}_{\mathbf{y}}, \boldsymbol{\Sigma}_{\mathbf{y}}),$$

where $\boldsymbol{\Sigma}_{\mathbf{y}} = (\boldsymbol{\Sigma}^{-1} + (1/\sigma^2_{\mathbf{y}})\mathbf{A}^{\top}\mathbf{A})^{-1}$ and $\boldsymbol{m}_{\mathbf{y}} = \boldsymbol{\Sigma}_{\mathbf{y}}\left((1/\sigma^2_{\mathbf{y}})\mathbf{A}^{\top}y + \boldsymbol{\Sigma}^{-1}\boldsymbol{m}\right)$. The Wasserstein-2 distance between the true posterior and $\hat{\pi}^{\ell(\eta)}_0$ is thus (Olkin & Pukelsheim, 1982)

$$W_2(\pi, \hat{\pi}^{\ell}_0)^2 = \|\boldsymbol{m}_{\mathbf{y}} - \hat{\boldsymbol{\mu}}^{\ell}_0\|^2 + \mathrm{tr}\left(\boldsymbol{\Sigma}_{\mathbf{y}} + \hat{\boldsymbol{\Sigma}}^{\ell}_0 - 2(\boldsymbol{\Sigma}^{\frac{1}{2}}_{\mathbf{y}}\hat{\boldsymbol{\Sigma}}^{\ell}_0\boldsymbol{\Sigma}^{\frac{1}{2}}_{\mathbf{y}})^{\frac{1}{2}}\right).$$

## C MORE DETAILS ON MGPS

### C.1 DETAILS ON THE LOSS

First, by the Data Processing inequality (Van Erven & Harremos, 2014, Example 2),

$$\mathsf{KL}(\lambda^{\boldsymbol{\varphi}}_{k|k+1}(\cdot|\mathbf{x}_{k+1}) \| \hat{\pi}^{\ell}_{k|k+1}(\cdot|\mathbf{x}_{k+1}))$$

$$\leq \int \log \frac{q_{k|\ell_k,k+1}(\mathbf{x}_k|\mathbf{x}_{\ell_k}, \mathbf{x}_{k+1})\lambda^{\boldsymbol{\varphi}}_{\ell_k|k+1}(\mathbf{x}_{\ell_k})}{q_{k|\ell_k,k+1}(\mathbf{x}_k|\mathbf{x}_{\ell_k}, \mathbf{x}_{k+1})\hat{\pi}^{\theta}_{\ell_k|k+1}(\mathbf{x}_{\ell_k}|\mathbf{x}_{k+1})}q_{k|\ell_k,k+1}(\mathbf{x}_k|\mathbf{x}_{\ell_k}, \mathbf{x}_{k+1})\lambda^{\boldsymbol{\varphi}}_{\ell_k|k+1}(\mathbf{x}_{\ell_k})\,\mathrm{d}\mathbf{x}_k\mathrm{d}\mathbf{x}_{\ell_k}$$

$$= \int \log \frac{\lambda^{\boldsymbol{\varphi}}_{\ell_k|k+1}(\mathbf{x}_{\ell_k})}{\hat{\pi}^{\theta}_{\ell_k|k+1}(\mathbf{x}_{\ell_k}|\mathbf{x}_{k+1})}q_{k|\ell_k,k+1}(\mathbf{x}_k|\mathbf{x}_{\ell_k}, \mathbf{x}_{k+1})\lambda^{\boldsymbol{\varphi}}_{\ell_k|k+1}(\mathbf{x}_{\ell_k})\,\mathrm{d}\mathbf{x}_k\mathrm{d}\mathbf{x}_{\ell_k}$$

$$= \mathsf{KL}(\lambda^{\boldsymbol{\varphi}}_{\ell_k|k+1} \| \hat{\pi}^{\theta}_{\ell_k|k+1}(\cdot|\mathbf{x}_{k+1}))$$

The gradient of $\boldsymbol{\varphi} \mapsto \mathsf{KL}(\lambda^{\boldsymbol{\varphi}}_{\ell_k|k+1} \| \hat{\pi}^{\theta}_{\ell_k|k+1}(\cdot|\mathbf{x}_{k+1}))$ for a given $\mathbf{x}_{k+1}$ writes

$$\nabla_{\boldsymbol{\varphi}}\mathcal{L}_k(\boldsymbol{\varphi}; X_{k+1})$$

$$= -\mathbb{E}\left[\nabla_{\boldsymbol{\varphi}}\log\hat{p}^{\theta}_{\ell_k}(\mathbf{y}|\hat{\boldsymbol{\mu}}_{\ell_k} + \mathrm{diag}(\mathrm{e}^{\hat{\boldsymbol{\rho}}_{\ell_k}})Z)\right] + \nabla_{\boldsymbol{\varphi}}\mathsf{KL}(\lambda^{\boldsymbol{\varphi}}_{\ell_k|k+1} \| p^{\theta}_{\ell_k|k+1}(\cdot|X_{k+1})) \quad \text{(C.1)}$$

and the second term can be expressed in a closed form since both distributions are Gaussians, *i.e.*,

$$
\nabla_{\boldsymbol{\varphi}} \mathsf{KL}(\lambda_{\ell_k|k+1}^{\boldsymbol{\varphi}} \parallel p_{\ell_k|k+1}^{\theta}(\cdot|\mathbf{x}_{k+1}))
$$

$$
= \nabla_{\boldsymbol{\varphi}} \left[ -\sum_{j=1}^{d} \hat{\boldsymbol{\rho}}_{\ell_k,j} + \frac{\|\hat{\boldsymbol{\mu}}_{\ell_k} - m_{\ell_k|0,k+1}(\mathbf{m}_{0|k+1}^{\theta}(\mathbf{x}_{k+1}), \mathbf{x}_{k+1})\|^2 + \sum_{j=1}^{d} \frac{\mathrm{e}^{2\hat{\boldsymbol{\rho}}_{\ell_k,j}}}{v_{\ell_k|0,k+1}}}{2 v_{\ell_k|0,k+1}} \right] .
$$

## C.2 WARM START

In this section we describe the warm-start approach discussed in the main paper. The complete algorithm with the warm-start procedure is given in Algorithm 3. The original version presented in Algorithm 1 relies on first sampling approximately $X_{\ell_k}$, given $X_{k+1}$, from the surrogate transition $\hat{\pi}_{\ell_k|k+1}^{\theta}(\cdot|X_{k+1})$ and then sampling $X_k$ from $q_{k|\ell_k,k+1}(\cdot|X_{\ell_k}, X_{k+1})$. In our warm-start approach, which we apply only during the first iterations of the algorithm (see the hyperparameter settings in Table 6), we draw inspiration from the decomposition

$$
\pi_{k|k+1}(\mathbf{x}_k|\mathbf{x}_{k+1}) = \int q_{k|1,k+1}(\mathbf{x}_k|\mathbf{x}_1, \mathbf{x}_{k+1}) \pi_{1|\ell_k}(\mathbf{x}_1|\mathbf{x}_{\ell_k}) \pi_{\ell_k|k+1}(\mathbf{x}_{\ell_k}|\mathbf{x}_{k+1}) \, \mathrm{d}\mathbf{x}_1 \mathrm{d}\mathbf{x}_{\ell_k} . \quad \text{(C.2)}
$$

It suggests introducing a second intermediary step that involves sampling approximately from the transition $X_1 \sim \pi_{1|\ell_k}(\cdot|X_{\ell_k})$. $X_k$ is then sampled from the bridge $q_{k|1,k+1}(\cdot|X_1, X_{k+1})$.

In order to draw approximate samples from $\pi_{1|\ell_k}(\cdot|X_{\ell_k})$ we leverage again a Gaussian variational approximation $\lambda_{1|\ell_k}^{\psi} := \mathrm{N}(\tilde{\boldsymbol{\mu}}_1, \mathrm{diag}(\mathrm{e}^{2\tilde{\boldsymbol{\rho}}_1}))$, which we fit by minimizing a proxy of the KL divergence between $\lambda_{1|\ell_k}^{\psi}$ and $\pi_{1|\ell_k}(\cdot|X_{\ell_k})$ of which the gradient w.r.t. $\psi := (\tilde{\boldsymbol{\mu}}_1, \tilde{\boldsymbol{\rho}}_1)$ writes

$$
\nabla_{\psi} \mathcal{L}_{1|\ell_k}^{\mathrm{ws}}(\psi; X_{\ell_k})_{|\psi_0} := -\nabla_{\psi} \left[ \sum_{j=1}^{d} \tilde{\boldsymbol{\rho}}_{1,j} - \frac{\|X_{\ell_k} - (\alpha_{\ell_k}/\alpha_1)^{1/2} \tilde{\boldsymbol{\mu}}_1\|^2 + (\alpha_{\ell_k}/\alpha_1) \sum_{j=1}^{d} \mathrm{e}^{\tilde{\boldsymbol{\rho}}_{1,j}^2}}{2(1 - \alpha_{\ell_k}/\alpha_1)} \right]_{|\psi_0}
$$

$$
- \mathbb{E} \left[ \nabla_{\psi}(\tilde{\boldsymbol{\mu}}_1 + \mathrm{diag}(\mathrm{e}^{\tilde{\boldsymbol{\rho}}_1}) Z)_{|\psi_0}^{\mathsf{T}} (\nabla_x \log p(\mathbf{y}|\cdot) + \mathbf{s}_1^{\theta})(\tilde{\boldsymbol{\mu}}_1 + \mathrm{diag}(\mathrm{e}^{\tilde{\boldsymbol{\rho}}_1}) Z_{|\psi_0}) \right] . \quad \text{(C.3)}
$$

This expression corresponds to the gradient w.r.t. $\psi$ of $\mathsf{KL}(\lambda_{1|\ell_k}^{\psi} \parallel \pi_{1|\ell_k}(\cdot|X_{\ell_k}))$ combined with the score approximation $\mathbf{s}_1^{\theta}$ of $\nabla_x \log q_1$ and the mild likelihood approximation $\nabla_x \log p_1(\mathbf{y}|\cdot)$ of $\nabla_x \log p(\mathbf{y}|\cdot)$. Indeed, we have that

$$
\mathsf{KL}(\lambda_{1|\ell_k}^{\psi} \parallel \pi_{1|\ell_k}(\cdot|X_{\ell_k})) = -\mathbb{E}_{\lambda_{1|\ell_k}^{\psi}} \left[ \log p_1(\mathbf{y}|X_1) + \log q_1(X_1) \right]
$$

$$
+ \int \log \frac{\lambda_{1|\ell_k}^{\psi}(\mathbf{x}_1)}{q_{\ell_k|1}(X_{\ell_k}|\mathbf{x}_1)} \lambda_{1|\ell_k}^{\psi}(\mathbf{x}_1) \, \mathrm{d}\mathbf{x}_1 .
$$

The second term can be computed in a closed form and its gradient corresponds to the first term in (C.3). As for the first term, under standard differentiability assumptions and applying the raparameterization trick, we find that

$$
\nabla_{\psi} \mathbb{E}_{\lambda_{1|\ell_k}^{\boldsymbol{\varphi}}} \left[ \log p_1(\mathbf{y}|X_1) + \log q_1(X_1) \right]_{|\psi_0}
$$

$$
= \nabla_{\psi} \mathbb{E} \left[ \left( \log p_1(\mathbf{y}|\cdot) + \log q_1 \right)(\tilde{\boldsymbol{\mu}}_1 + \mathrm{diag}(\mathrm{e}^{\tilde{\boldsymbol{\rho}}_1}) Z) \right]_{|\psi_0}
$$

$$
= \mathbb{E} \left[ \nabla_{\psi}(\tilde{\boldsymbol{\mu}}_1 + \mathrm{diag}(\mathrm{e}^{\tilde{\boldsymbol{\rho}}_1}) Z)_{|\psi_0}^{\mathsf{T}} (\nabla_x \log p_1(\mathbf{y}|\cdot) + \nabla_x \log q_1)(\tilde{\boldsymbol{\mu}}_1 + \mathrm{diag}(\mathrm{e}^{\tilde{\boldsymbol{\rho}}_1}) Z_{|\psi_0}) \right] .
$$

Plugging the previous approximations yields the second term in (C.3). The warm-start procedure is summarized in Algorithm 2. We also use a single sample Monte Carlo estimate to perform the optimization. Finally, at step $k$ and given $\hat{X}_{\ell_k}$, the initial parameter $\psi_k$ of the variational approximation is chosen so that

$$
\lambda_{1|\ell_k}^{\psi_k}(\mathbf{x}_1) = q_{1|0,\ell_k}(\mathbf{x}_1|\mathbf{m}_{0|\ell_k}^{\theta}(\hat{X}_{\ell_k}), \hat{X}_{\ell_k}) .
$$

---

**Algorithm 2** WarmStart

1: **Input:** step $k$, samples $(\hat{X}_{\ell_k}, X_{k+1})$, gradient steps $M$
2: $\tilde{\boldsymbol{\mu}}_1 \leftarrow m_{1|0,\ell_k}(\mathbf{m}^\theta_{0|\ell_k}(\hat{X}_{\ell_k}), \hat{X}_{\ell_k}), \quad \tilde{\boldsymbol{\rho}}_1 \leftarrow \frac{1}{2}\log v_{1|0,\ell_k}$
3: **for** $j = 1$ **to** $M$ **do**
4: $\quad Z \sim \mathrm{N}(0_d, \mathbf{I}_d)$
5: $\quad (\tilde{\boldsymbol{\mu}}_1, \tilde{\boldsymbol{\rho}}_1) \leftarrow \mathsf{OptimizerStep}(\nabla_\psi \widetilde{\mathcal{L}}^{\mathrm{ws}}_{1|\ell_k}(\cdot, Z; X_{\ell_k}); \tilde{\boldsymbol{\mu}}_1, \tilde{\boldsymbol{\rho}}_1)$
6: **end for**
7: $\hat{X}_1 \leftarrow \hat{\boldsymbol{\mu}}_1 + \mathrm{diag}(\mathrm{e}^{\hat{\boldsymbol{\rho}}_1})Z_1$ where $Z_1 \sim \mathrm{N}(0_d, \mathbf{I}_d)$
8: $X_k \leftarrow m_{k|1,k+1}(\hat{X}_1, X_{k+1}) + v^{1/2}_{k|1,k+1}Z_k$
9: **Output:** $X_k, \hat{X}_1$

---

**Algorithm 3** MGPS with warm start strategy

1: **Input:** $(\ell_k)^n_{k=1}$ with $\ell_n = n$, $\ell_1 = 1$, gradient steps $(M_k)^{n-1}_{k=1}$, warm start threshold $w$.
2: $X_n \sim \mathcal{N}(0_d, \mathbf{I}_d)$, $\hat{X}_n \leftarrow X_n$
3: $\hat{X}_0 \leftarrow \mathbf{m}^\theta_{0|n}(\hat{X}_n)$
4: **for** $k = n - 1$ **to** $1$ **do**
5: $\quad \hat{\boldsymbol{\mu}}_{\ell_k} \leftarrow m_{\ell_k|0,k+1}(\hat{X}_0, X_{k+1}), \quad \hat{\boldsymbol{\rho}}_{\ell_k} \leftarrow \frac{1}{2}\log v_{\ell_k|0,k+1}$
6: $\quad$ **for** $j = 1$ **to** $M_k$ **do**
7: $\qquad Z \sim \mathrm{N}(0_d, \mathbf{I}_d)$
8: $\qquad (\hat{\boldsymbol{\mu}}_{\ell_k}, \hat{\boldsymbol{\rho}}_{\ell_k}) \leftarrow \mathsf{OptimizerStep}(\nabla_{\boldsymbol{\varphi}} \widetilde{\mathcal{L}}_{\ell_k}(\cdot, Z; X_{k+1}); \hat{\boldsymbol{\mu}}_{\ell_k}, \hat{\boldsymbol{\rho}}_{\ell_k})$
9: $\quad$ **end for**
10: $\quad Z_{\ell_k}, Z_k \overset{\mathrm{i.i.d.}}{\sim} \mathrm{N}(0_d, \mathbf{I}_d)$
11: $\quad \hat{X}_{\ell_k} \leftarrow \hat{\boldsymbol{\mu}}_{\ell_k} + \mathrm{diag}(\mathrm{e}^{\hat{\boldsymbol{\rho}}_{\ell_k}})Z_{\ell_k}$
12: $\quad$ **if** $k \geq w$ **then**
13: $\qquad (X_k, \hat{X}_1) \leftarrow \mathsf{WarmStart}(k, \hat{X}_{\ell_k}, X_{k+1}, M_k)$
14: $\qquad \hat{X}_0 \leftarrow \hat{X}_1$
15: $\quad$ **else**
16: $\qquad X_k \leftarrow m_{k|\ell_k,k+1}(\hat{X}_{\ell_k}, X_{k+1}) + v^{1/2}_{k|\ell_k,k+1}Z_k$
17: $\qquad \hat{X}_0 \leftarrow \mathbf{m}^\theta_{0|\ell_k}(X_{\ell_k})$
18: $\quad$ **end if**
19: **end for**
20: **Output:** $\hat{X}_0$

---

After having sampled $\tilde{X}_1 \sim \lambda^{\psi^*_k}_{1|\ell_k}$, we use it to first sample $X_k \sim q_{k|1,k+1}(\cdot | \tilde{X}_1, X_{k+1})$ and then initialize the next variational approximation $\lambda^{\boldsymbol{\varphi}}_{\ell_{k-1}|k}$. The initial parameter $\boldsymbol{\varphi}_{k-1}$ is set so that

$$\lambda^{\boldsymbol{\varphi}_{k-1}}_{\ell_{k-1}|k}(\mathbf{x}_{\ell_{k-1}}) = q_{\ell_{k-1}|0,k}(\mathbf{x}_{\ell_{k-1}} | \tilde{X}_1, X_k),$$

see line 5 in Algorithm 3.

### C.3 DIFFERENCE WITH DPS

In this section, we detail the differences between MGPS and the DPS algorithm proposed in Chung et al. (2023). While DPS cannot be seen as an instantition of our methodology, we highlight the main differences by deriving a specific case of MGPS that closely resembles DPS.

We assume that $\ell_k = k$ for all $k \in [\![1, n-1]\!]$ and that $p(\mathbf{y}|\mathbf{x}) = \mathcal{N}(\mathbf{y}; \mathcal{A}(\mathbf{x}), \sigma^2_{\mathbf{y}}\mathbf{I}_{d_{\mathbf{y}}})$ where $\mathcal{A} : \mathbb{R}^d \to \mathbb{R}^{d_{\mathbf{y}}}$. We consider the same optimization procedure as in Algorithm 1 but we: (i) optimize only the mean parameter $\hat{\boldsymbol{\mu}}_k$ and fix the covariance of the variational approximation to $v_{k|k+1}\mathbf{I}_d$, (ii) perform a single stochastic optimization step.

Following that, the initialization strategy (3.9) boils down to setting

$$\lambda^{\boldsymbol{\varphi}_k}_{k|k+1}(\mathbf{x}_k|\mathbf{x}_{k+1}) = p^\theta_{k|k+1}(\mathbf{x}_k|\mathbf{x}_{k+1}) \tag{C.4}$$

which means that the inital mean parameter is $\hat{\boldsymbol{\mu}}_k^0 := m_{k|0,k+1}(\mathbf{m}_{0|k+1}^\theta(\mathbf{x}_{k+1}), \mathbf{x}_{k+1})$. With this initialization $\boldsymbol{\varphi}_k \mapsto \mathsf{KL}(\lambda_{k|k+1}^{\boldsymbol{\varphi}_k}(\cdot|\mathbf{x}_{k+1}) \| p_{k|k+1}^\theta(\cdot|\mathbf{x}_{k+1}))$ has its global optimum attained at $\boldsymbol{\varphi}_k = \hat{\boldsymbol{\mu}}_k^0$. Thus, the gradient (C.1) writes $\nabla_{\boldsymbol{\varphi}_k}\mathcal{L}_k(\boldsymbol{\varphi}_k; \mathbf{x}_{k+1})_{|\hat{\boldsymbol{\mu}}_k^0} = -\mathbb{E}\big[\nabla_{\boldsymbol{\varphi}_k}\log\hat{p}_k^\theta(\mathbf{y}|\hat{\boldsymbol{\mu}}_k + \sqrt{v_{k|k+1}}Z)_{|\hat{\boldsymbol{\mu}}_k^0}\big]$. Next, updating the parameters of the variational approximation using a single stochastic gradient descent step with the gradient estimate $-\nabla_{\boldsymbol{\varphi}_k}\log\hat{p}_k^\theta(\mathbf{y}|\hat{\boldsymbol{\mu}}_k + \sqrt{v_{k|k+1}}Z_k)_{|\hat{\boldsymbol{\mu}}_k^0}$, where $Z_k \sim \mathrm{N}(0_d, \mathbf{I}_d)$ and step-size

$$\gamma_k = \frac{\zeta\sigma_{\mathbf{y}}^2}{\|\mathbf{y} - \mathcal{A}\big(\mathbf{m}_{0|k}^\theta(\hat{\boldsymbol{\mu}}_k^0 + \sqrt{v_{k|k+1}}Z_k)\big)\|}$$

yields the variational approximation

$$\lambda_{k|k+1}^{\boldsymbol{\varphi}_k^*}(\mathbf{x}_k|\mathbf{x}_{k+1}) = \mathcal{N}(\mathbf{x}_k; \hat{\boldsymbol{\mu}}_k^0 - \zeta\nabla_{\boldsymbol{\varphi}_k}\|\mathbf{y} - \mathcal{A}\big(\mathbf{m}_{0|k}^\theta(\hat{\boldsymbol{\mu}}_k + \sqrt{v_{k|k+1}}Z_k)\big)\|_{|\hat{\boldsymbol{\mu}}_k^0}, v_{k|k+1}\mathbf{I}_d)\,.$$

In order to simply the expression we have used the fact that

$$\frac{-\nabla_{\boldsymbol{\varphi}_k}\log\hat{p}_k^\theta(\mathbf{y}|\hat{\boldsymbol{\mu}}_k + \sqrt{v_{k|k+1}}Z_k)_{|\hat{\boldsymbol{\mu}}_k^0}}{\|\mathbf{y} - \mathcal{A}\big(\mathbf{m}_{0|k}^\theta(\hat{\boldsymbol{\mu}}_k^0 + \sqrt{v_{k|k+1}}Z_k)\big)\|} = \frac{1}{\sigma_{\mathbf{y}}^2}\nabla_{\boldsymbol{\varphi}_k}\|\mathbf{y} - \mathcal{A}\big(\mathbf{m}_{0|k}^\theta(\hat{\boldsymbol{\mu}}_k + \sqrt{v_{k|k+1}}Z_k)\big)\|_{|\hat{\boldsymbol{\mu}}_k^0}\,.$$

Therefore, given $X_{k+1}$, a draw from the variational approximation in MGPS is obtained following the update

$$X_k = \tilde{X}_k - \zeta\nabla_{\mathbf{x}_k}\|\mathbf{y} - \mathcal{A}(\mathbf{m}_{0|k}^\theta(\mathbf{x}_k))\|_{|\mathbf{x}_k = \tilde{X}_k'}\,, \quad (\tilde{X}_k, \tilde{X}_k') \overset{\text{i.i.d.}}{\sim} p_{k|k+1}^\theta(\cdot|X_{k+1})\,.$$

On the other hand, the DPS transition is given by

$$\lambda_{k|k+1}^{\mathsf{DPS}}(\mathbf{x}_k|\mathbf{x}_{k+1}) := \mathcal{N}(\mathbf{x}_k; \hat{\boldsymbol{\mu}}_k^0 - \zeta\nabla_{\mathbf{x}_{k+1}}\|\mathbf{y} - \mathcal{A}(\mathbf{m}_{0|k+1}^\theta(\mathbf{x}_{k+1}))\|, v_{k|k+1}\mathbf{I}_d)\,.$$

and, hence a sample $X_k^{\mathsf{DPS}}$ is drawn following

$$X_k^{\mathsf{DPS}} = \tilde{X}_k - \zeta\nabla_{\mathbf{x}_{k+1}}\|\mathbf{y} - \mathcal{A}(\mathbf{m}_{0|k+1}^\theta(\mathbf{x}_{k+1}))\|_{|\mathbf{x}_{k+1} = X_{k+1}}\,, \quad \tilde{X}_k \sim p_{k|k+1}^\theta(\cdot|X_{k+1})\,.$$

The difference between MGPS and DPS is in: (i) the diffusion step used for the denoiser ($k$ for MGPS, $k+1$ for DPS), (ii) the sample where the gradient is evaluated.

## D EXPERIMENTS

In this section we provide the implementation details on our algorithm as well as the algorithms we benchmark against. We use the the hyperparameters recommended by the authors and tune them on each dataset if they are not provided.

### D.1 IMPLEMENTATION DETAILS AND HYPERPARAMETERS FOR MGPS

We implement Algorithm 1 with $\ell_k = \lfloor k/2 \rfloor$ and use the Adam optimizer (Kingma & Ba, 2015) with a learning rate of 0.03 for optimization. The number of gradient steps is adjusted based on the complexity of the task: posterior sampling with the `ImageNet` DDM prior or `FFHQ` LDM prior is more challenging and therefore requires additional gradient steps. Detailed hyperparameters are provided in Table 6.

The warm-start strategy outlined in Appendix C.2 improved reconstruction plausibility and eliminated potential artifacts. A similar effect was observed when performing multiple gradient steps ($M = 20$) during the initial stages. For latent-space models, switching the intermediate step to $\ell_k = k$ for the second half of the diffusion process has been crucial and significantly enhanced reconstruction quality by mitigating the smoothing effect, which often removes important details. A similar strategy has been useful for the Gaussian deblurring and motion deblurring tasks on the `ImageNet` dataset.

### D.2 IMPLEMENTATION DETAILS OF COMPETITORS

**DPS.** We implemented Chung et al. (2023, Algorithm 1) and refer to Chung et al. (2023, App. D) for the values of its hyperparameters. After tuning, we adopt $\gamma = 0.2$ for JPEG 2% and $\gamma = 0.07$ for High Dynamic Range tasks.

Table 6: The hyperparameters used in MGPS for the considered datasets.

| | Warm start | Threshold ($w$) | Diffusion steps | $\ell_k$ | Learning rate | Gradient steps | | |
|---|---|---|---|---|---|---|---|---|
| **FFHQ** | ✓ | $\lfloor 3n/4 \rfloor$ | $n \in \{50, 100, 300\}$ | $\lfloor k/2 \rfloor$ | 0.03 | $M_k = \begin{cases} 20 & \text{if } k \geq n-5 \\ 20 & \text{if } k \mod 10 = 0 \\ 2 & \text{otherwise} \end{cases}$ | | |
| **FFHQ** (LDM) | ✓ | $\lfloor 3n/4 \rfloor$ | $n \in \{50, 100, 300\}$ | $\ell_k = \begin{cases} \lfloor k/2 \rfloor & \text{if } k > \lfloor n/2 \rfloor \\ k & \text{otherwise} \end{cases}$ | 0.03 | $M_k = \begin{cases} 20 & \text{if } k \geq n-5 \\ 10 & \text{if } k \mod 10 = 0 \\ 5 & \text{otherwise} \end{cases}$ | | |
| **ImageNet** | ✓ | $\lfloor 3n/4 \rfloor$ | $n \in \{50, 100, 300\}$ | $\lfloor k/2 \rfloor$ | 0.03 | $M_k = \begin{cases} 20 & \text{if } k \geq n-5 \\ 10 & \text{if } k \mod 20 = 0 \\ 2 & \text{otherwise} \end{cases}$ | | |
| **Gaussian Mixture** | ✗ | – | $n = 300$ | $\lfloor k/2 \rfloor$ | 0.1 | $M_k = \begin{cases} 20 & \text{if } k \geq n-5 \\ 20 & \text{if } k \mod 10 = 0 \\ 2 & \text{otherwise} \end{cases}$ | | |
| **PTB-XL (ECG)** | ✓ | $\lfloor 3n/4 \rfloor$ | $n \in \{50, 300\}$ | $\lfloor k/2 \rfloor$ | 0.03 | $M_k = \begin{cases} 20 & \text{if } k \geq n-5 \\ 5 & \text{if } k \mod 20 = 0 \\ 5 & \text{otherwise} \end{cases}$ | | |

**DiffPIR.** We implemented Zhu et al. (2023, Algorithm 1) to make it compatible with our existing code base. We used the hyperparameters recommended in the official, released version[2]. Unfortunately, we did not manage to make the algorithm converge for nonlinear problems. While the authors give some guidelines to handle such problems (Zhu et al., 2023, Eqn. (13)), examples are missing in the paper and the released code. Similarly, we do not run it on the motion deblur task as the FFT-based solution provided in (Zhu et al., 2023) is only valid for circular convolution, yet we opted for the experimental setup of Chung et al. (2023) which uses convolution with reflect padding.

**DDNM.** We adapted the implementation in the released code[3] to our code base. Since DDNM utilizes the pseudo-inverse of the degradation operator, we noticed that it is unstable for operators whose SVD are prone to numerical errors, such as Gaussian Blur with wide convolution kernel.

**RedDiff.** We used the implementation of RedDiff available in the released code[4]. On nonlinear problems, for which the pseudo-inverse of the observation is not available, we initialized the variational optimization with a sample from the standard Gaussian distribution.

**PGDM.** We relied on the implementation provided in RedDiff's code as some of its authors are co-authors of PGDM. The implementation features a subtle difference with Song et al. (2023a, Algorithm 1): in the last line of the algorithm, the guidance term $g$ is multiplied by $\sqrt{\alpha_t}$ but in the implementation it is multiplied by $\sqrt{\alpha_{t-1}\alpha_t}$. This modification stabilizes the algorithm on most tasks. For JPEG 2%, we found that it worsens the performance. In this case we simply multiply by $\sqrt{\alpha_t}$, as in the original algorithm.

**PSLD** We implemented the PSLD algorithm provided in Rout et al. (2024, Algorithm (2)) and used the hyperparameters provided by the authors in the publicly available implementation[5].

**ReSample** We used the original code provided by the authors[6] and modified it to expose several hyperparameters that were not directly accessible in the released version. Specifically, we exposed the tolerance $\varepsilon$ and the maximum number of iterations $N$ for solving the optimization problems related to hard data consistency, as well as the scaling factor for the variance of the stochastic resampling distribution $\gamma$. Our experiments revealed that the algorithm is highly sensitive to $\varepsilon$. We found that setting it equal to the noise level of the inverse problem gave the best reconstruction across tasks and noise levels. We set the maximum number of gradient iterations to $N = 200$ to make the algorithm less computationally prohibitive. Finally, we tuned $\gamma$ for each task but found it has less impact on the quality of the reconstruction compared to $\varepsilon$.

---

[2] https://github.com/yuanzhi-zhu/DiffPIR
[3] https://github.com/wyhuai/DDNM
[4] https://github.com/NVlabs/RED-diff
[5] https://github.com/LituRout/PSLD
[6] https://github.com/soominkwon/resample

### D.2.1 RUNTIME AND MEMORY

To get the runtime and GPU memory consumption of an algorithm on a dataset, we average these two metrics over both samples of the dataset and the considered tasks in Section 4.

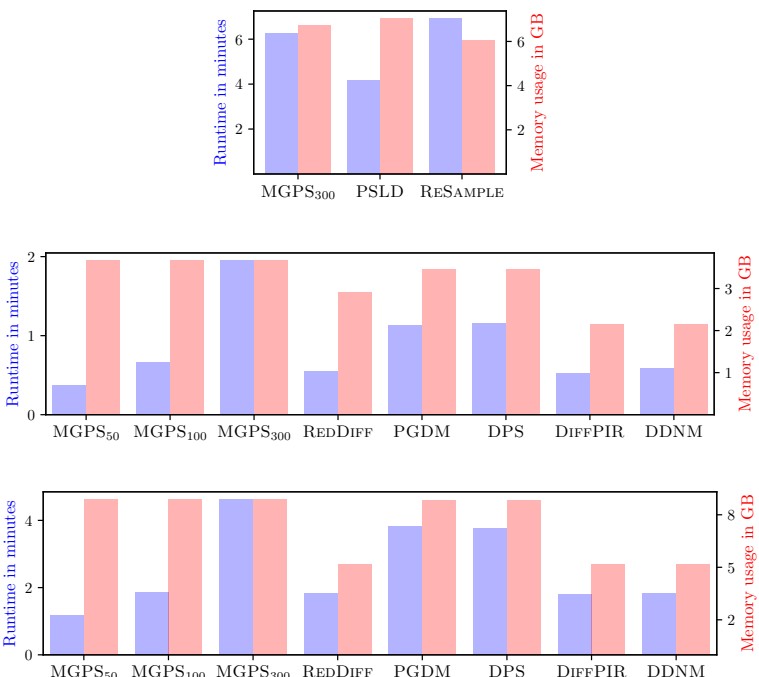

Figure 7: Runtime and memory requirement of the considered algorithms on datasets: `FFHQ` latent space (1st row), `FFHQ` pixel space (2nd row), and `ImageNet` (3rd row). The left axis displays the runtime in minutes, whereas the right axis the GPU memory requirement in Gigabytes (GB).

### D.3 GAUSSIAN MIXTURES

In this section, we elaborate more on the Gaussian mixture experiment in Section 4, where we consider a linear inverse problem with a Gaussian mixture as prior. Therefore, the likelihood is $p(\mathbf{y}|\cdot) : \mathbf{x} \mapsto \mathcal{N}(\mathbf{y}; \mathbf{A}\mathbf{x}, \sigma_{\mathbf{y}}^2 \mathbf{I}_{d_{\mathbf{y}}})$. Recall that a Gaussian mixture with $C \in \mathbb{N}$ components, whose weights, means, and covariances are $w_i > 0$, $\boldsymbol{m}_i \in \mathbb{R}^d$, and $\sigma_i^2 \mathbf{I}_d$, respectively, has the density

$$q(\mathbf{x}) = \sum_{i=1}^{C} w_i \mathcal{N}(\mathbf{x}; \boldsymbol{m}_i, \sigma_i^2 \mathbf{I}_d).$$

**Denoiser.** The denoiser is obtained via Tweedie's formula (Robbins, 1956),
$$\mathbf{m}_{0|k}(\mathbf{x}_k) = \left(\mathbf{x}_k + (1 - \alpha_k)\nabla \log q_k(\mathbf{x}_k)\right)/\sqrt{\alpha_k},$$
where the densities of the marginals $(q_k)_{k=1}^{n}$ are straightforward,

$$q_k(\mathbf{x}_k) = \int q(\mathbf{x}_0) q_{k|0}(\mathbf{x}_k|\mathbf{x}_0) \, \mathrm{d}\mathbf{x}_0 = \sum_{i=1}^{C} w_i \int \mathcal{N}(\mathbf{x}_0; \boldsymbol{m}_i, \sigma_i^2 \mathbf{I}_d) q_{k|0}(\mathbf{x}_k|\mathbf{x}_0) \, \mathrm{d}\mathbf{x}_0$$

$$= \sum_{i=1}^{C} w_i \mathcal{N}(\mathbf{x}_k; \sqrt{\alpha_k}\boldsymbol{m}_i, (\alpha_k \sigma_i^2 + v_k)\mathbf{I}_d).$$

**Posterior.** Furthermore, following Bishop (2006, Eqn. 2.116), the posterior can be shown to be a Gaussian mixture

$$\pi(\mathbf{x}) \propto g(\mathbf{x})q(\mathbf{x}) \propto \sum_{i=1}^{C} \bar{w}_i \mathcal{N}(\mathbf{x}; \bar{\boldsymbol{m}}_i, \bar{\boldsymbol{\Sigma}}_i),$$

with new weights, covariances, and means given by

$$\bar{w}_i = w_i \mathcal{N}(\mathbf{y}; \mathbf{A}\boldsymbol{m}_i, \sigma_{\mathbf{y}}^2 \mathbf{I}_{d_{\mathbf{y}}} + \sigma_i^2 \mathbf{A}\mathbf{A}^\top),$$

$$\bar{\boldsymbol{\Sigma}}_i = \left((1/\sigma_i^2)\mathbf{I}_d + (1/\sigma_{\mathbf{y}}^2)\mathbf{A}^\top \mathbf{A}\right)^{-1},$$

$$\bar{\boldsymbol{m}}_i = \bar{\boldsymbol{\Sigma}}_i\left((1/\sigma_{\mathbf{y}}^2)A^\top \mathbf{y} + (1/\sigma_i^2)\boldsymbol{m}_i\right),$$

where the weights are un-normalized.

**Experimental setup.** We consider two setups where $d \in \{20, 200\}$ and generate Gaussian mixtures of $C = 25$ components with means $(\boldsymbol{m}_i)_{i=1}^C = \{(8i, 8j, \dots, 8i, 8j) \in \mathbb{R}^d : (i, j) \in [\![-2, 2]\!]^2\}$, unit covariances, and weights $w_i$ drawn from a uniform distribution on $[0, 1]$ then normalized to sum to 1. The linear inverse problem $(\mathbf{y}, \mathbf{A})$ is generated by first drawing the matrix $\mathbf{A} \in \mathbb{R}^{1 \times d}$ with entries i.i.d. from $N(0, 1)$, then sampling $\mathbf{x}^*$ from the prior $q$ and finally computing $\mathbf{y} = \mathbf{A}\mathbf{x}^* + \sigma_{\mathbf{y}}\varepsilon$ with $\varepsilon \sim N(0, 1)$ to get the observation. The standard deviation of the inverse problem is fixed to $\sigma_{\mathbf{y}} = 0.05$ across all problem instances.

To assess the performance of each algorithm, we draw 2000 samples and compare against 2000 samples from the true posterior distribution using the Sliced Wasserstein (SW) distance by averaging over $10^4$ slices. In Table 1, we report the average SW and the 95% confidence interval over 100 replicates.

## D.4 EXTENDED IMAGE EXPERIMENTS

To strengthen our conclusions and calculate the FID, we reran MGPS and the two closest competitors on 1000 images for the following five tasks: half mask, JPEG, motion deblur, nonlinear deblur, and high dynamic range, using the three priors (FFHQ, ImageNet, FFHQ-LDM). The Table 7 shows that MGPS significantly outperforms the other competitors, including for the FID, with no significant change in other metrics compared to what we calculated on a smaller dataset.

We extend, in Tables 8 and 9, the results in Table 2 by including MGPS with $n = 50$ and $n = 100$. In the setting $n = 100$, the runtime of MGPS is twice lower than that of DPS, PGDM and comparable to that of DIFFPIR, DDNM and REDDIFF, see Figure 7. It is also outperforming them on most of the tasks, especially the nonlinear ones, as is seen in the table below. In the setting $n = 50$, MGPS has the lowest runtime among all the methods while maintaining competitive performance.

Finally, in Appendix D.6, we present sample reconstructions on the various tasks. For each algorithm, we generate five samples and select the four most visually appealing ones. Completely black or white images correspond to failure cases of DPS.

## D.5 ECG EXPERIMENTS

### D.5.1 IMPLEMENTATION DETAILS

**PTB-XL dataset** We use 12-lead ECGs at a sampling frequency of 100 Hz from the PTB-XL dataset (Wagner et al., 2020). For both training and generation, we do not use the augmented limb leads aVL, aVR and aVF as they can be obtained with the following relations: aVL=(I-III)/2, aVF=(II+III)/2 and aVR=-(I+II)/2. This leads to inputs of shape $T \times 9$ instead of $T \times 12$ (where $T$ is the length of the signal). For evaluation metrics, we reconstruct the augmented limb leads. For the MB task, following Alcaraz & Strodthoff (2022), we use 2.56-second ECG random crops, hence the inputs are of shape $256 \times 9$. For the ML task, we use the full 10-second ECGs and pad them into a tensor of shape $1024 \times 9$. Table 11 summarizes the input shapes for models and algorithms for each task. Table 12 summarizes the distribution of train/val/test splits and the number of cardiac conditions (RBBB, LBBB, AF, SB) per split. Note that for posterior sampling, we only use the test set, and for both training and sampling we never use the cardiac condition.

**Diffusion model** Following Alcaraz & Strodthoff (2023; 2022), we used Structured State Space Diffusion (SSSD) models (Gu et al., 2022; Goel et al., 2022). SSSD generates a sequence $u(t)$ with the following differential equation denpending on the input seuqnece $\mathbf{x}(t)$ and a hidden state $h(t)$:

$$h'(t) = Ah(t) + Bx(t) \tag{D.1}$$

$$u(t) = Ch(t) + Dx(t), \tag{D.2}$$

Table 7: Mean and confidence intervals of LPIPS/PSNR/SSIM values and FID value for various linear and nonlinear imaging tasks on 1k images for the 3 priors. Best is in **bold**.

| Metric | FFHQ | | | ImageNet | | | FFHQ-LDM | |
|---|---|---|---|---|---|---|---|---|
| | MGPS | DDNM | DIFFPIR | MGPS | DDNM | DIFFPIR | MGPS | RESAMPLE |
| Half mask | | | | | | | | |
| FID | **27.0** | 38.6 | 45.2 | **40.0** | 50.0 | 57.0 | **49.5** | 66.6 |
| LPIPS | $\mathbf{0.19 \pm 0.00}$ | $0.23 \pm 0.00$ | $0.25 \pm 0.00$ | $\mathbf{0.30 \pm 0.00}$ | $0.38 \pm 0.01$ | $0.40 \pm 0.01$ | $\mathbf{0.26 \pm 0.00}$ | $0.30 \pm 0.00$ |
| PNSR | $15.9 \pm 0.2$ | $\mathbf{16.3 \pm 0.2}$ | $16.1 \pm 0.3$ | $15.0 \pm 0.1$ | $\mathbf{16.0 \pm 0.1}$ | $15.8 \pm 0.1$ | $\mathbf{15.6 \pm 0.1}$ | $15.7 \pm 0.1$ |
| SSIM | $0.70 \pm 0.00$ | $\mathbf{0.74 \pm 0.00}$ | $0.72 \pm 0.01$ | $0.63 \pm 0.00$ | $\mathbf{0.68 \pm 0.01}$ | $0.67 \pm 0.01$ | $\mathbf{0.69 \pm 0.00}$ | $0.67 \pm 0.00$ |
| | MGPS | DPS | DIFFPIR | MGPS | DPS | DIFFPIR | MGPS | RESAMPLE |
| Motion deblur | | | | | | | | |
| FID | **29.7** | 36.7 | 77.0 | **35.3** | 55.0 | 87.3 | **44.6** | 51.8 |
| LPIPS | $\mathbf{0.12 \pm 0.00}$ | $0.17 \pm 0.00$ | $0.22 \pm 0.00$ | $\mathbf{0.20 \pm 0.01}$ | $0.40 \pm 0.01$ | $0.39 \pm 0.01$ | $\mathbf{0.19 \pm 0.00}$ | $0.20 \pm 0.00$ |
| PNSR | $26.7 \pm 0.1$ | $24.1 \pm 0.1$ | $\mathbf{27.4 \pm 0.1}$ | $24.4 \pm 0.1$ | $21.4 \pm 0.1$ | $24.2 \pm 0.1$ | $26.4 \pm 0.1$ | $\mathbf{26.7 \pm 0.1}$ |
| SSIM | $\mathbf{0.77 \pm 0.00}$ | $0.70 \pm 0.01$ | $0.71 \pm 0.00$ | $\mathbf{0.67 \pm 0.01}$ | $0.55 \pm 0.01$ | $0.61 \pm 0.00$ | $\mathbf{0.76 \pm 0.00}$ | $0.72 \pm 0.00$ |
| JPEG (QF = 2) | | | | | | | | |
| FID | **31.6** | 87.6 | 109 | **61.4** | 128.8 | 92.8 | **45.0** | 65.3 |
| LPIPS | $\mathbf{0.15 \pm 0.00}$ | $0.37 \pm 0.00$ | $0.33 \pm 0.01$ | $\mathbf{0.40 \pm 0.01}$ | $0.60 \pm 0.01$ | $0.61 \pm 0.00$ | $\mathbf{0.21 \pm 0.00}$ | $0.26 \pm 0.01$ |
| PNSR | $\mathbf{25.2 \pm 0.1}$ | $19.0 \pm 0.2$ | $24.5 \pm 0.1$ | $\mathbf{22.2 \pm 0.1}$ | $16.7 \pm 0.1$ | $22.2 \pm 0.1$ | $24.6 \pm 0.1$ | $\mathbf{24.8 \pm 0.1}$ |
| SSIM | $\mathbf{0.73 \pm 0.01}$ | $0.55 \pm 0.02$ | $0.70 \pm 0.00$ | $\mathbf{0.60 \pm 0.01}$ | $0.41 \pm 0.02$ | $0.60 \pm 0.01$ | $\mathbf{0.71 \pm 0.00}$ | $0.66 \pm 0.01$ |
| Nonlinear deblur | | | | | | | | |
| FID | **50.8** | 164 | 88.4 | 113 | 272 | **112** | **69.2** | 71.5 |
| LPIPS | $\mathbf{0.23 \pm 0.01}$ | $0.51 \pm 0.02$ | $0.68 \pm 0.01$ | $\mathbf{0.43 \pm 0.01}$ | $0.83 \pm 0.01$ | $0.66 \pm 0.01$ | $\mathbf{0.26 \pm 0.01}$ | $0.32 \pm 0.01$ |
| PNSR | $\mathbf{24.3 \pm 0.2}$ | $16.2 \pm 0.5$ | $21.9 \pm 0.1$ | $22.2 \pm 0.2$ | $9.9 \pm 0.4$ | $20.7 \pm 0.2$ | $23.9 \pm 0.1$ | $\mathbf{24.2 \pm 0.1}$ |
| SSIM | $\mathbf{0.70 \pm 0.01}$ | $0.45 \pm 0.02$ | $0.42 \pm 0.01$ | $\mathbf{0.58 \pm 0.01}$ | $0.41 \pm 0.01$ | $0.241 \pm 0.01$ | $\mathbf{0.69 \pm 0.01}$ | $0.67 \pm 0.01$ |
| High dynamic range | | | | | | | | |
| FID | **20.9** | 153 | 47.5 | **20.2** | 316 | 35.7 | 44.2 | **38.7** |
| LPIPS | $\mathbf{0.08 \pm 0.01}$ | $0.40 \pm 0.04$ | $0.20 \pm 0.01$ | $\mathbf{0.11 \pm 0.01}$ | $0.83 \pm 0.02$ | $0.20 \pm 0.01$ | $0.14 \pm 0.00$ | $\mathbf{0.12 \pm 0.00}$ |
| PNSR | $\mathbf{27.0 \pm 0.1}$ | $18.7 \pm 0.2$ | $21.7 \pm 0.1$ | $26.3 \pm 0.2$ | $9.9 \pm 0.2$ | $21.9 \pm 0.1$ | $25.5 \pm 0.1$ | $\mathbf{26.0 \pm 0.1}$ |
| SSIM | $\mathbf{0.83 \pm 0.01}$ | $0.55 \pm 0.04$ | $0.72 \pm 0.01$ | $\mathbf{0.83 \pm 0.01}$ | $0.23 \pm 0.02$ | $0.71 \pm 0.01$ | $0.80 \pm 0.01$ | $\mathbf{0.83 \pm 0.01}$ |

where $A, B, C, D$ are transition matrices. We use the SSSD$^{SA}$ (Alcaraz & Strodthoff, 2022) (also denoted as Sashimi (Goel et al., 2022)) architecture publicly available[7]. We parametrize the matrix $A = \Lambda - pp^*$ where $p$ is a vector and $\Lambda$ is a diagonal matrix. This parametrization with facilitates the control of the spectrum of $A$ to enforce stability (negative real part of eigenvalues). We use a multi-scale architecture, with a stack of residual S4 blocks. The top tier processes the raw audio signal at its original sampling rate, while lower tiers process downsampled versions of the input signal. The outputs of the lower tiers are upsampled with up-pooling and down-pooling operations and combined with the input to the upper tier to provide a stronger conditioning signal. Hyper-parameters are described in table 13. The selected model is the model of the last epoch. Training time is 30 hours for 2.5 seconds ECGs, 42 hours for 10 seconds ECGs.

**Algorithm parameters** We use the same parameters as described in table 6 for MGPS and the implementation described in appendix D.2 for competitors. Since we had to run the experiments over 2k samples, we set the number of diffusion steps such that the runtime approximates 30 seconds for generating 10 samples. This leads to the parameters shown in table 14. For the ML experiment, we also tested MGPS with 300 diffusion steps, which resulted in a runtime of 5 minutes and 30 seconds per batch of 10 samples. The results improved, but we already outperform competitors with just 50 diffusion steps and a 30-second runtime.

**Evaluation metrics** For the MB task, for each observation, we generate 10 samples - instead of 100 as in in (Alcaraz & Strodthoff, 2022). We then compute the Mean Absolute Error (MAE) and Root Mean Squared Error (RMSE) between each generated sample and the ground-truth. The final score is the average of these errors over the 10 generated samples. We report the confidence intervals over the test set in table 4 and table 16.

For the ML task, we use a classifier trained to detect four cardiac conditions: Right Bundle Branch Block (RBBB), Left Bundle Branch Block (LBBB), Atrial Fibrillation (AF), and Sinus Bradycardia

---

[7] https://github.com/AI4HealthUOL/SSSD
[8] https://github.com/albertfgu/diffwave-sashimi/tree/master

Table 8: Mean LPIPS/PSNR/SSIM values for various linear and nonlinear imaging tasks on the FFHQ $256 \times 256$ dataset. Best is in **bold** and second best is underlined.

| Task | MGPS$_{50}$ | MGPS$_{100}$ | MGPS$_{300}$ | DPS | PGDM | DDNM | DIFFPIR | REDDIFF |
|---|---|---|---|---|---|---|---|---|
| | | | | LPIPS ↓ | | | | |
| SR (×4) | 0.13 | 0.10 | **0.09** | **0.09** | 0.33 | 0.14 | 0.13 | 0.36 |
| SR (×16) | 0.27 | 0.26 | 0.26 | **0.24** | 0.44 | 0.30 | 0.28 | 0.51 |
| Box inpainting | 0.16 | 0.12 | **0.10** | 0.19 | 0.17 | 0.12 | 0.18 | 0.19 |
| Half mask | 0.24 | 0.22 | **0.20** | 0.24 | 0.26 | 0.22 | 0.23 | 0.28 |
| Gaussian Deblur | 0.21 | 0.18 | 0.15 | 0.16 | 0.87 | 0.19 | **0.12** | 0.26 |
| Motion Deblur | 0.19 | 0.15 | **0.13** | 0.16 | – | – | – | 0.21 |
| | | | | | | | | |
| JPEG (QF = 2) | 0.20 | 0.17 | **0.16** | 0.39 | 1.10 | – | – | 0.32 |
| Phase retrieval | 0.20 | 0.14 | **0.11** | 0.46 | – | – | – | 0.25 |
| Nonlinear deblur | **0.23** | **0.23** | 0.23 | 0.52 | – | – | – | 0.66 |
| High dynamic range | 0.13 | 0.09 | **0.07** | 0.49 | – | – | – | 0.20 |
| | | | | PSNR ↑ | | | | |
| SR (×4) | 27.83 | 27.79 | 27.79 | 28.24 | 23.34 | **29.52** | 27.17 | 27.25 |
| SR (×16) | 20.45 | 20.34 | 20.22 | 20.67 | 17.65 | **22.43** | 20.75 | 21.91 |
| Box inpainting | 21.55 | 22.22 | **22.68** | 18.39 | 21.13 | 22.35 | 21.96 | 21.79 |
| Half mask | 15.10 | 15.32 | 15.54 | 14.82 | 16.03 | 16.16 | 15.17 | **16.21** |
| Gaussian Deblur | 25.09 | 25.19 | 25.89 | 24.20 | 13.36 | 26.69 | 25.89 | **26.72** |
| Motion Deblur | 26.07 | 26.64 | 26.48 | 24.24 | – | – | – | **27.58** |
| | | | | | | | | |
| JPEG (QF = 2) | 25.00 | **25.23** | 24.94 | 18.50 | 12.76 | – | – | 24.42 |
| Phase retrieval | 24.20 | 26.60 | **27.25** | 14.87 | – | – | – | 24.85 |
| Nonlinear deblur | 24.16 | 24.21 | **24.37** | 15.89 | – | – | – | 21.97 |
| High dynamic range | 24.77 | 26.07 | **27.74** | 16.83 | – | – | – | 21.25 |
| | | | | SSIM ↑ | | | | |
| SR (×4) | 0.81 | 0.80 | 0.79 | 0.81 | 0.50 | **0.85** | 0.77 | 0.70 |
| SR (×16) | 0.58 | 0.57 | 0.55 | 0.59 | 0.38 | **0.67** | 0.60 | 0.62 |
| Box inpainting | 0.80 | 0.81 | 0.82 | 0.76 | 0.70 | **0.83** | 0.80 | 0.70 |
| Half mask | 0.69 | 0.70 | 0.71 | 0.66 | 0.56 | **0.73** | 0.67 | 0.65 |
| Gaussian Deblur | 0.71 | 0.73 | 0.75 | 0.69 | 0.14 | **0.77** | 0.73 | 0.76 |
| Motion Deblur | 0.77 | **0.78** | 0.77 | 0.71 | – | – | – | 0.71 |
| | | | | | | | | |
| JPEG (QF = 2) | **0.75** | 0.74 | 0.73 | 0.51 | 12.76 | – | – | 0.71 |
| Phase retrieval | 0.74 | 0.78 | **0.79** | 0.43 | – | – | – | 0.61 |
| Nonlinear deblur | **0.70** | **0.70** | 0.69 | 0.46 | – | – | – | 0.42 |
| High dynamic range | 0.79 | 0.81 | **0.86** | 0.48 | – | – | – | 0.71 |

(SB) on the PTB-XL dataset. We follow the XRESNET1D50 described in (Strodthoff et al., 2020) with hyper-parameters reported in table 15. We apply the classifier to the ground-truth ECG of the test set and to the samples generated from lead I. As in the MB task, for each observation, 10 samples are generated. The output of the classifier is averaged over these 10 samples. For each cardiac condition, we compute balanced accuracy to account for class imbalance (see table 12). The classification threshold is selected using PTB-XL validation-set.

### D.5.2 ADDITIONAL RESULTS

**Discussion** In table 4, we demonstrated that most posterior sampling algorithms outperform the trained diffusion model for the missing block reconstruction task. This result is particularly interesting as it suggests that training a diffusion model (which takes several days) is not necessary for this task. However, when Alcaraz & Strodthoff (2023) trained the model on the reconstruction of random missing blocks (RMB), where 50% of each lead is independently removed, the model outperformed all posterior sampling algorithms on the MB task. We report in table 16 the results of the top two algorithms, as well as the model trained on the MB task and the RMB task. The significant improvement between RMB and MB can be seen as an enhancement due to data augmentation.

**Generated samples**

### D.6 SAMPLE IMAGES

While it may appear that some of the methods underperform on some tasks/images compared to the original publications, for instance Figure 14 and Figure 16, they still produce competitive

Table 9: Mean LPIPS/PSNR/SSIM values for various linear and nonlinear imaging tasks on the IMAGENET $256 \times 256$ dataset. Best is in **bold** and second best is underlined.

| Task | MGPS$_{50}$ | MGPS$_{100}$ | MGPS$_{300}$ | DPS | PGDM | DDNM | DiffPir | RedDiff |
|------|------|------|------|------|------|------|------|------|
| | | | | LPIPS ↓ | | | | |
| SR (×4) | 0.36 | 0.33 | **0.30** | 0.41 | 0.78 | 0.34 | 0.36 | 0.56 |
| SR (×16) | 0.58 | 0.55 | 0.53 | **0.50** | 0.60 | 0.70 | 0.63 | 0.83 |
| Box inpainting | 0.31 | 0.26 | **0.22** | 0.34 | 0.29 | 0.28 | 0.28 | 0.36 |
| Half mask | 0.39 | 0.34 | **0.29** | 0.44 | 0.38 | 0.38 | 0.35 | 0.44 |
| Gaussian Deblur | 0.35 | **0.29** | 0.32 | 0.35 | 1.00 | 0.45 | **0.29** | 0.52 |
| Motion Deblur | 0.35 | 0.25 | **0.22** | 0.39 | – | – | – | 0.40 |
| JPEG (QF = 2) | 0.50 | 0.46 | **0.42** | 0.63 | 1.31 | – | – | 0.51 |
| Phase retrieval | 0.54 | 0.52 | **0.47** | 0.62 | – | – | – | 0.60 |
| Nonlinear deblur | 0.49 | 0.47 | **0.44** | 0.88 | – | – | – | 0.67 |
| High dynamic range | 0.22 | 0.15 | **0.10** | 0.85 | – | – | – | 0.21 |
| | | | | PSNR ↑ | | | | |
| SR (×4) | 24.68 | 24.70 | 24.77 | 23.52 | 15.67 | **25.55** | 24.26 | 24.24 |
| SR (×16) | 18.56 | 18.42 | 18.04 | 18.22 | 15.80 | **20.43** | 19.37 | 19.95 |
| Box inpainting | 17.52 | 17.95 | 18.25 | 14.34 | 17.35 | **20.08** | 19.77 | 18.90 |
| Half mask | 14.98 | 15.14 | 15.60 | 14.65 | 14.36 | **17.06** | 15.79 | 16.96 |
| Gaussian Deblur | 22.56 | 21.96 | 20.10 | 21.20 | 9.93 | **23.29** | 22.10 | 23.27 |
| Motion Deblur | 23.91 | **25.03** | 24.50 | 21.59 | – | – | – | 24.43 |
| JPEG (QF = 2) | 21.96 | 22.17 | **22.44** | 16.11 | 5.29 | – | – | 22.15 |
| Phase retrieval | 16.36 | 16.94 | **18.10** | 14.40 | – | – | – | 15.78 |
| Nonlinear deblur | 21.89 | 22.23 | **22.36** | 8.49 | – | – | – | 20.76 |
| High dynamic range | 23.93 | 25.64 | **27.04** | 9.32 | – | – | – | 22.88 |
| | | | | SSIM ↑ | | | | |
| SR (×4) | 0.65 | 0.66 | 0.66 | 0.60 | 0.23 | **0.70** | 0.63 | 0.61 |
| SR (×16) | 0.41 | 0.38 | 0.35 | 0.40 | 0.24 | **0.50** | 0.46 | 0.47 |
| Box inpainting | 0.71 | 0.74 | 0.76 | 0.70 | 0.62 | **0.78** | 0.74 | 0.67 |
| Half mask | 0.61 | 0.63 | 0.65 | 0.55 | 0.53 | **0.69** | 0.63 | 0.62 |
| Gaussian Deblur | 0.56 | 0.53 | 0.44 | 0.51 | 0.07 | **0.60** | 0.51 | 0.57 |
| Motion Deblur | 0.64 | **0.69** | 0.65 | 0.55 | – | – | – | 0.60 |
| JPEG (QF = 2) | 0.59 | **0.60** | **0.60** | 0.39 | 0.01 | – | – | 0.59 |
| Phase retrieval | 0.37 | 0.40 | **0.43** | 0.29 | – | – | – | 0.26 |
| Nonlinear deblur | 0.57 | **0.58** | **0.58** | 0.24 | – | – | – | 0.41 |
| High dynamic range | 0.75 | 0.81 | **0.84** | 0.25 | – | – | – | 0.73 |

Table 10: Mean LPIPS/PSNR/SSIM values for various linear and nonlinear imaging tasks on FFHQ $256 \times 256$ dataset with LDM prior. Best is in **bold** and second best is underlined.

| Task | MGPS | ReSample | PSLD | MGPS | ReSample | PSLD | MGPS | ReSample | PSLD |
|------|------|------|------|------|------|------|------|------|------|
| | LPIPS ↓ | | | PSNR ↑ | | | SSIM ↑ | | |
| SR (×4) | **0.11** | 0.20 | 0.22 | **28.46** | 26.08 | 25.53 | **0.83** | 0.69 | 0.70 |
| SR (×16) | **0.30** | 0.36 | 0.35 | 20.64 | 21.09 | **21.42** | 0.57 | 0.56 | **0.63** |
| Box inpainting | **0.16** | 0.22 | 0.26 | **22.94** | 18.80 | 20.39 | **0.79** | 0.75 | 0.66 |
| Half mask | **0.25** | 0.30 | 0.31 | **15.11** | 14.59 | 14.75 | **0.69** | 0.67 | 0.61 |
| Gaussian Deblur | 0.16 | **0.15** | 0.35 | **27.57** | 27.44 | 19.95 | **0.79** | 0.75 | 0.47 |
| Motion Deblur | **0.18** | 0.19 | 0.41 | 26.49 | **26.85** | 18.14 | **0.77** | 0.72 | 0.39 |
| JPEG (QF = 2) | **0.20** | 0.26 | – | **24.75** | 24.33 | – | **0.72** | 0.67 | – |
| Phase retrieval | **0.34** | 0.41 | – | **22.21** | 19.05 | – | **0.62** | 0.47 | – |
| Nonlinear deblur | **0.26** | 0.30 | – | 23.79 | **24.44** | – | **0.70** | 0.68 | – |
| High dynamic range | **0.15** | **0.15** | – | 25.17 | **25.42** | – | 0.79 | **0.81** | – |

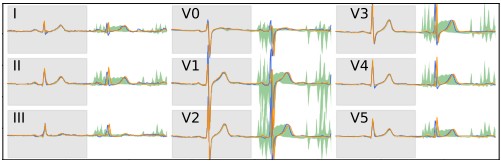

Figure 8: Missing block imputation with MGPS on 2.56s 12-lead ECG. Ground-truth in blue, 10%–90% quantile range in green, random sample in orange.

Table 11: ECG size and input-shape per task.

| Task | ECG size (seconds) | Total leads | Used leads | Input shape |
|------|-----|-----|-----|-----|
| MB | 2,56 | 12 | I, II, III, V1–6 | $256 \times 9$ |
| ML | 10 | 12 | I, II, III, V1–6 | $1024 \times 9$ |

Table 12: PTB-XL dataset description.

| Split | All | RBBB | LBBB | AF | SB |
|------|-----|-----|-----|-----|-----|
| Train | 17,403 | 432 | 428 | 1211 | 503 |
| Val | 2,183 | 55 | 54 | 151 | 64 |
| Test | 2,203 | 54 | 54 | 152 | 64 |

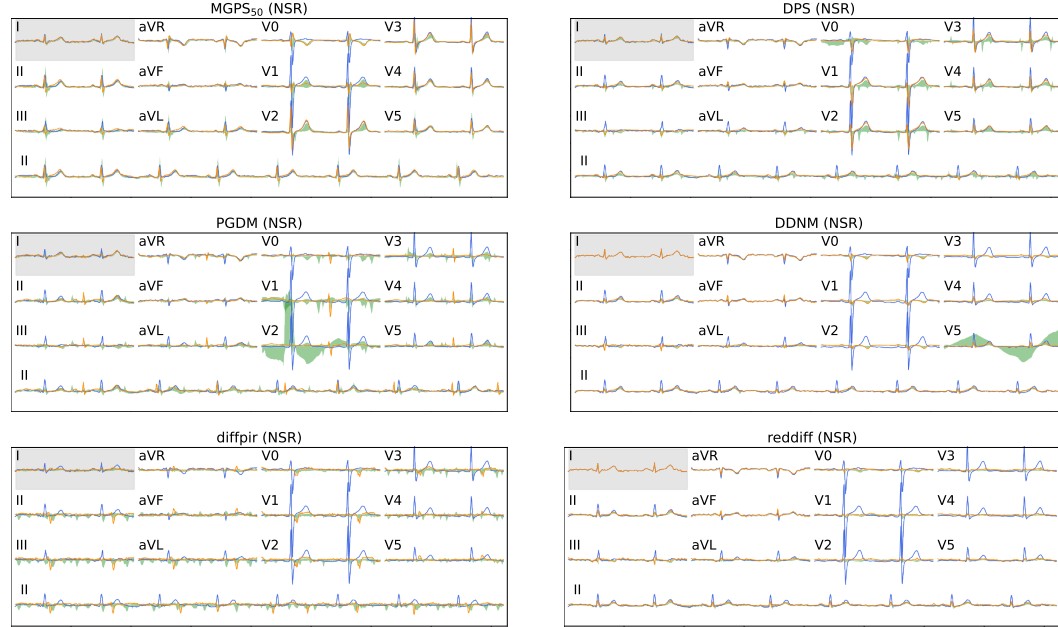

Figure 9: Missing lead reconstruction from lead I on 10s 12-lead Normal Sinus Rythm (NSR) ECGs. Ground-truth in blue, 10%–90% quantile range in green, random sample in orange.

reconstructions on others; see for example Figure 15, Figure 18, and Figure 23. Similar patterns are also displayed in Zhang et al. (2023, Figure 10) and Liu et al. (2023, Figure 9). With that being said, we highlight that these discrepancies appear more frequently on the ImageNet dataset than the FFHQ one. This can be explained by the fact that `ImageNet` is notoriously challenging due to its diversity, encompassing 1000 classes. This also seems to happen on one of the most difficult tasks, namely the half mask one.

Table 13: ECG diffusion generative model hyper-parameters.

| Hyper-parameter | Value |
| --- | --- |
| Residual layers | 4 |
| Pooling factor | [1, 2, 2] |
| Feature expansion | 2 |
| Diffusion embedding dim. 1 | 128 |
| Diffusion embedding dim. 2 | 512 |
| Diffusion embedding dim. 3 | 512 |
| Diffusion steps | 1000 |
| Optimizer | Adam |
| Number of iterations | 150k |
| Loss function | MSE |
| Learning rate | 0.002 |
| Batch size | 128 |
| Number of parameters | 16 millions |

Table 14: Number of diffusion steps used in posterior sampling algorithms for ECG tasks.

| $\text{MGPS}_{50}$ | $\text{MGPS}_{300}$ | DPS | PGDM | DDNM | DIFFPIR | REDDIFF |
| --- | --- | --- | --- | --- | --- | --- |
| 50 | 300 | 400 | 200 | 200 | 500 | 400 |

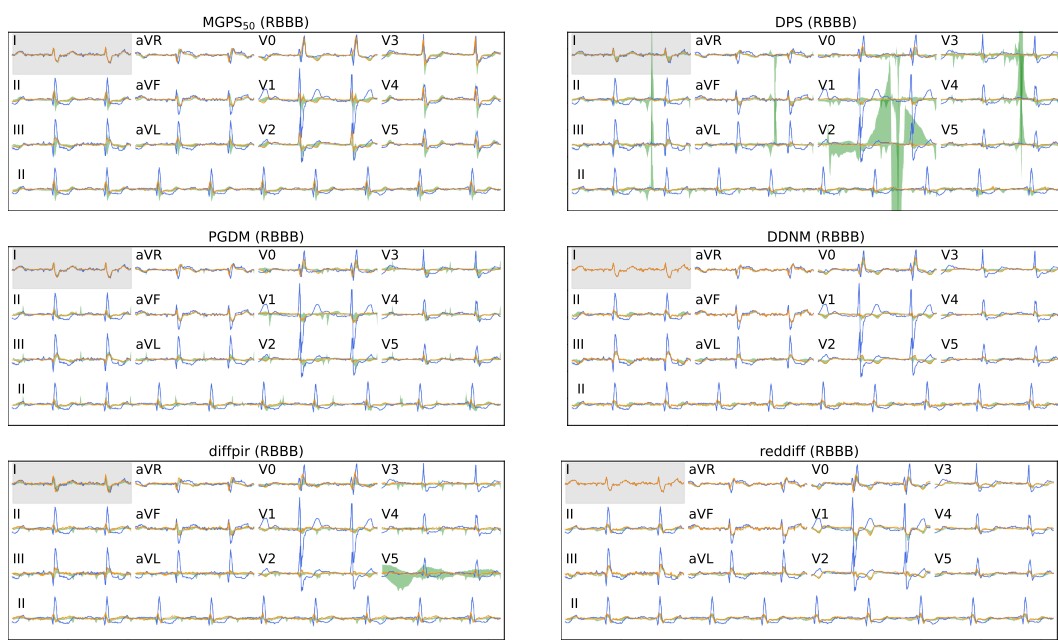

Figure 10: Missing lead reconstruction from lead I on 10s 12-lead Right Bundle Branch Block (RBBB) ECGs. Ground-truth in blue, 10%–90% quantile range in green, random sample in orange.

Table 15: XRESNET1D50 downstream classifier hyper-parameters.

| Hyper-parameter | Value |
| --- | --- |
| Blocks | 4 |
| Layers per block | [3, 4, 6, 3] |
| Expansion | 4 |
| Stride | 1 |
| Optimizer | Adam |
| Learning rate | 0.001 |
| Batch size | 0.001 |
| Epochs | 100 |

Table 16: MAE and RMSE for missing block task on the PTB-XL dataset.

| Metric | MGPS | DDNM | TRAINEDDIFF-MB | TRAINEDDIFF-RMB |
|--------|------|------|----------------|-----------------|
| MAE | $0.111 \pm 2\mathrm{e}{-3}$ | $0.103 \pm 2\mathrm{e}{-3}$ | $0.116 \pm 2\mathrm{e}{-3}$ | $\mathbf{0.0879 \pm 2\mathrm{e}{-3}}$ |
| RMSE | $0.225 \pm 4\mathrm{e}{-3}$ | $0.224 \pm 4\mathrm{e}{-3}$ | $0.266 \pm 3\mathrm{e}{-3}$ | $\mathbf{0.217 \pm 6\mathrm{e}{-3}}$ |

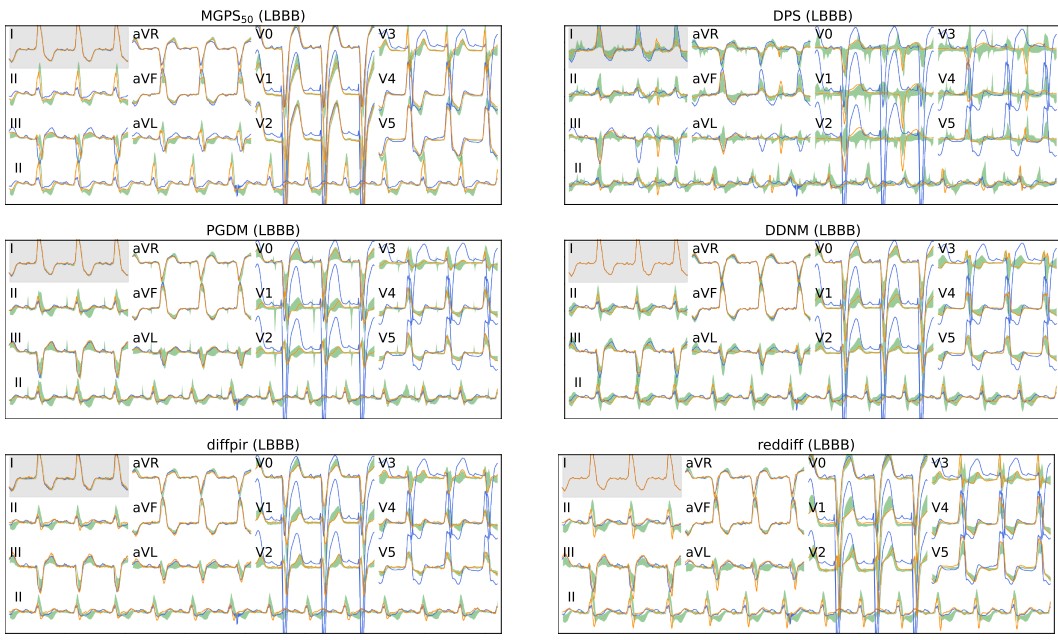

Figure 11: Missing lead reconstruction from lead I on 10s 12-lead Left Bundle Branch Block (LBBB) ECGs. Ground-truth in blue, 10%–90% quantile range in green, random sample in orange.

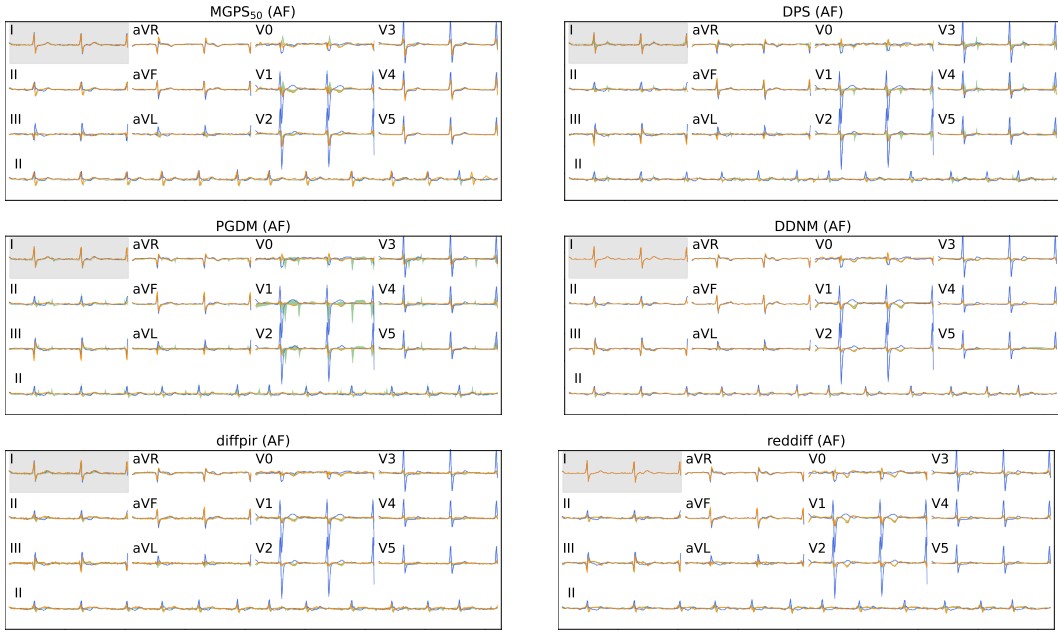

Figure 12: Missing lead reconstruction from lead I on 10s 12-lead Atrial Fibrillation (AF) ECGs. Ground-truth in blue, 10%–90% quantile range in green, random sample in orange.

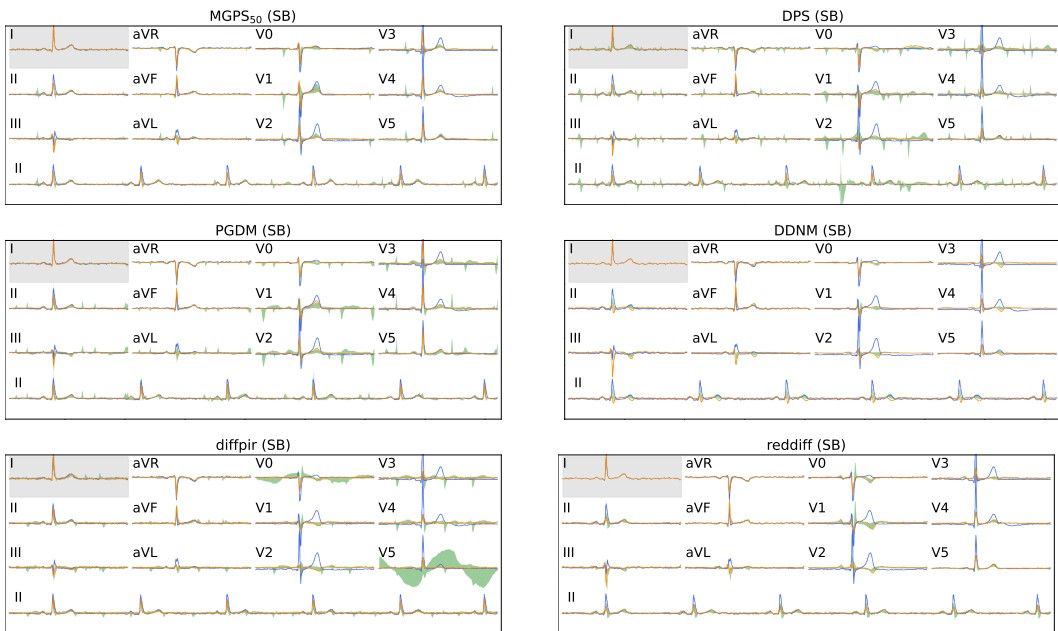

Figure 13: Missing lead reconstruction from lead I on 10s 12-lead Sinus Bradycardia (SB) ECGs. Ground-truth in blue, 10%–90% quantile range in green, random sample in orange.

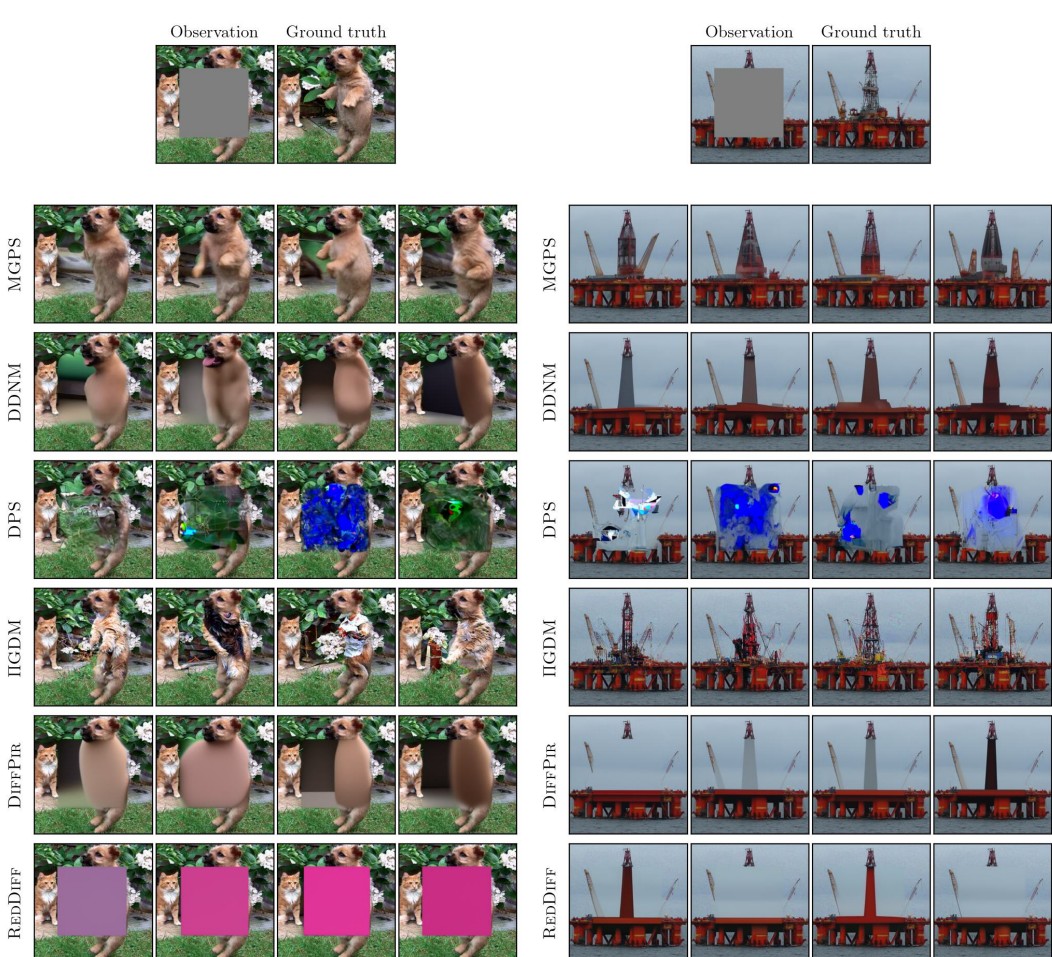

Figure 14: Sample reconstructions with box mask on `ImageNet` dataset.

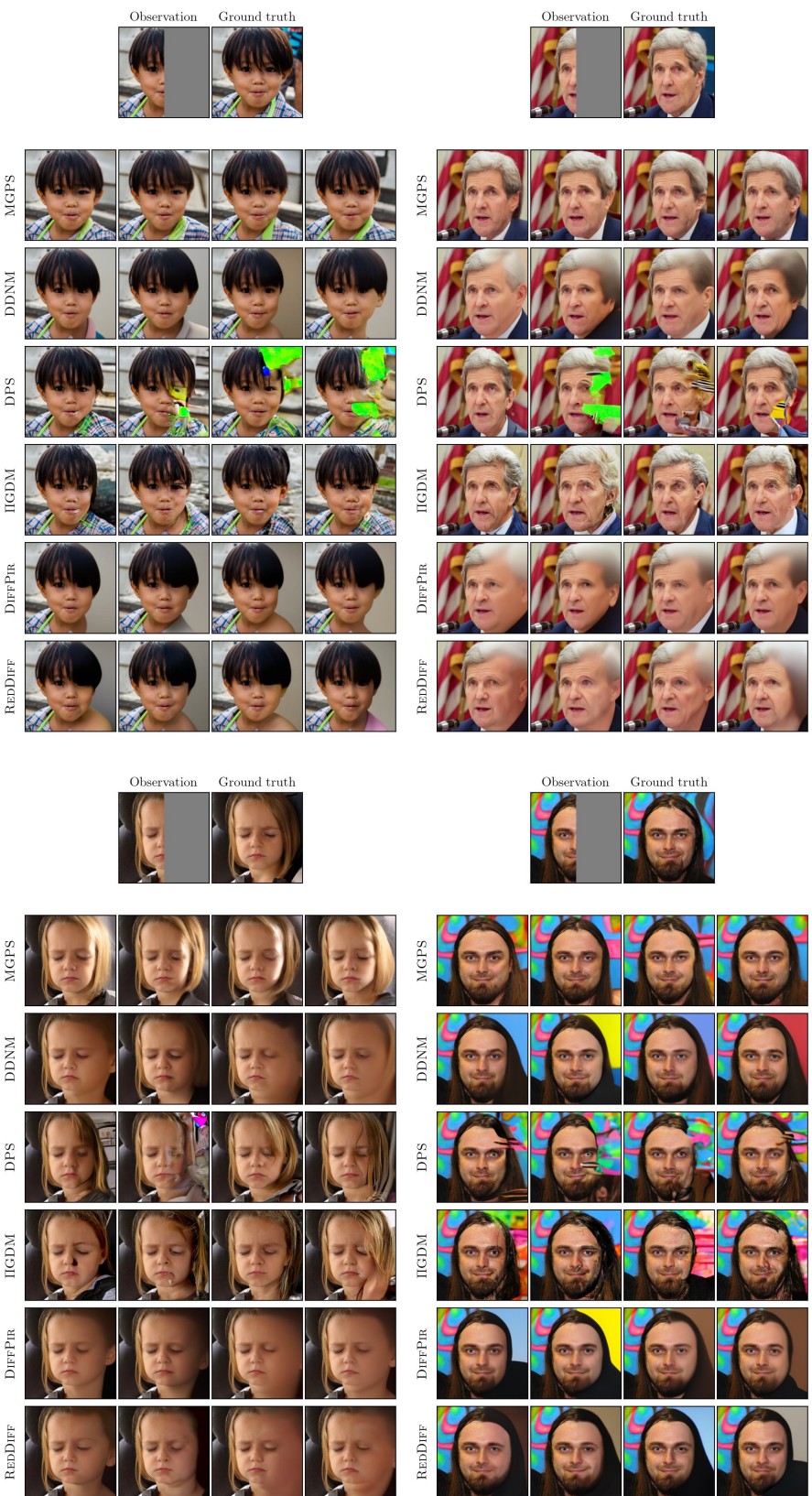

Figure 15: Sample reconstructions with half mask on FFHQ dataset.

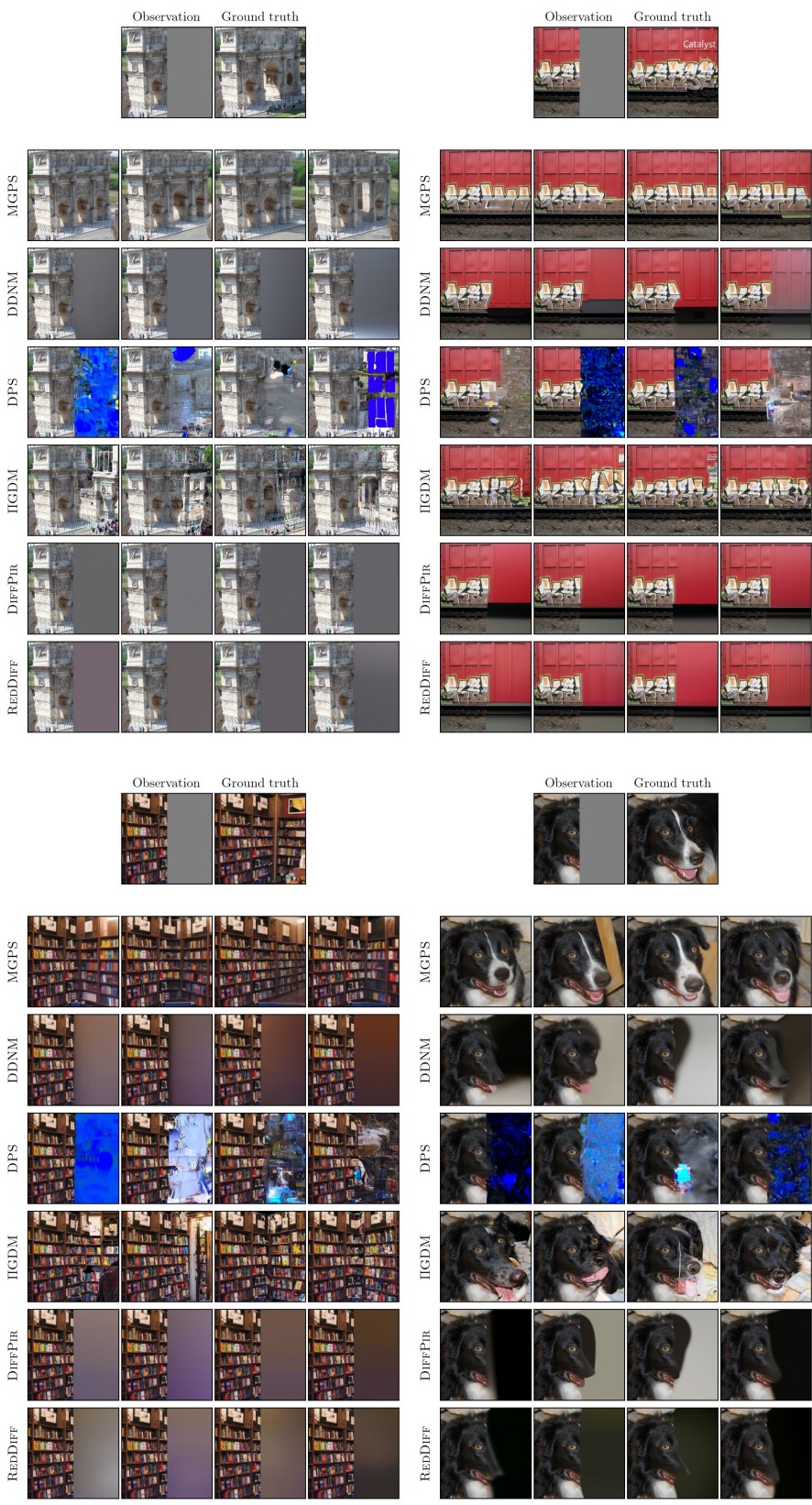

Figure 16: Sample reconstructions with half mask on ImageNet dataset.

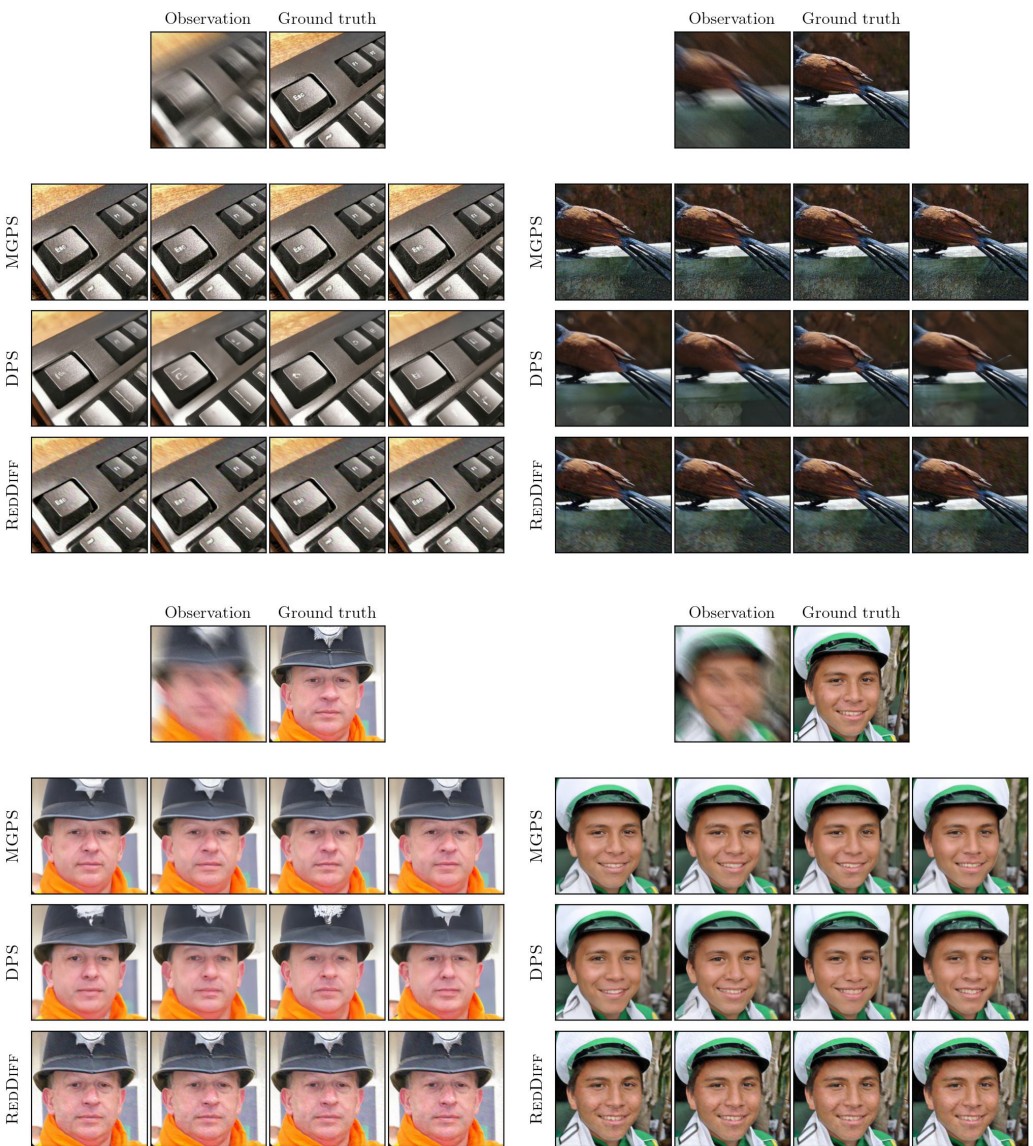

Figure 17: Sample reconstructions on motion deblurring task.

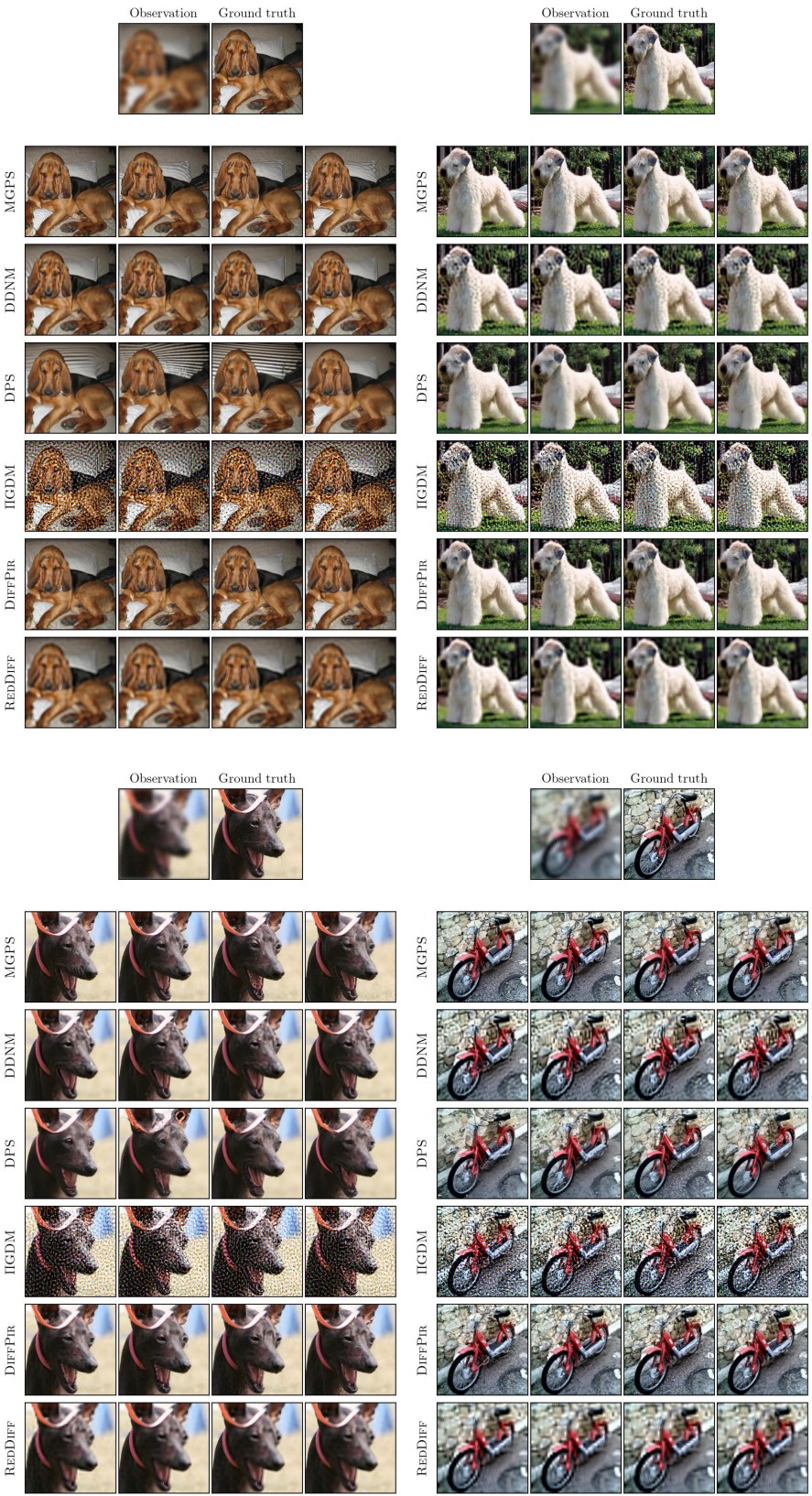

Figure 18: Sample reconstructions on Gaussian deblurring task.

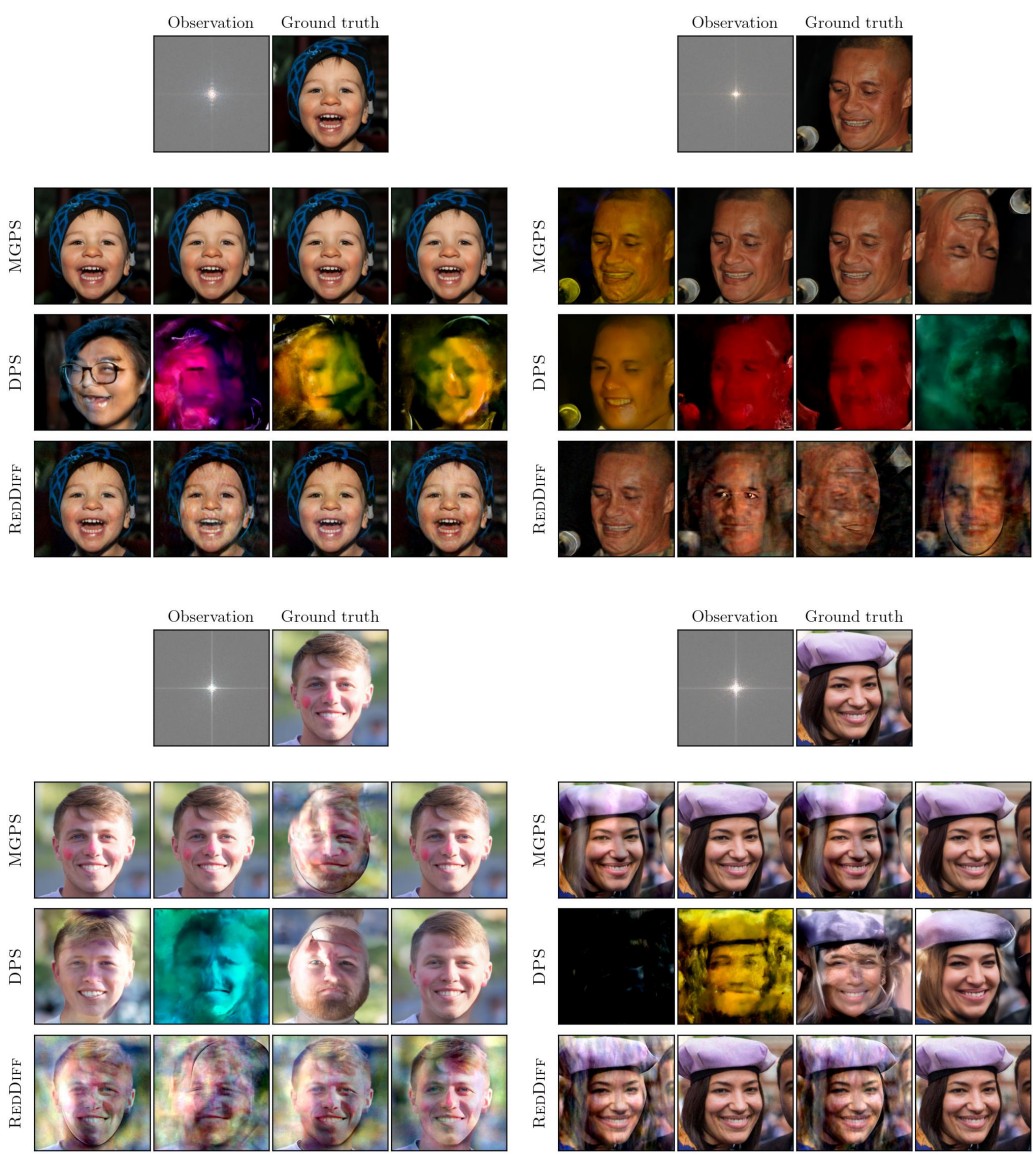

Figure 19: Sample reconstructions on phase retrieval task.

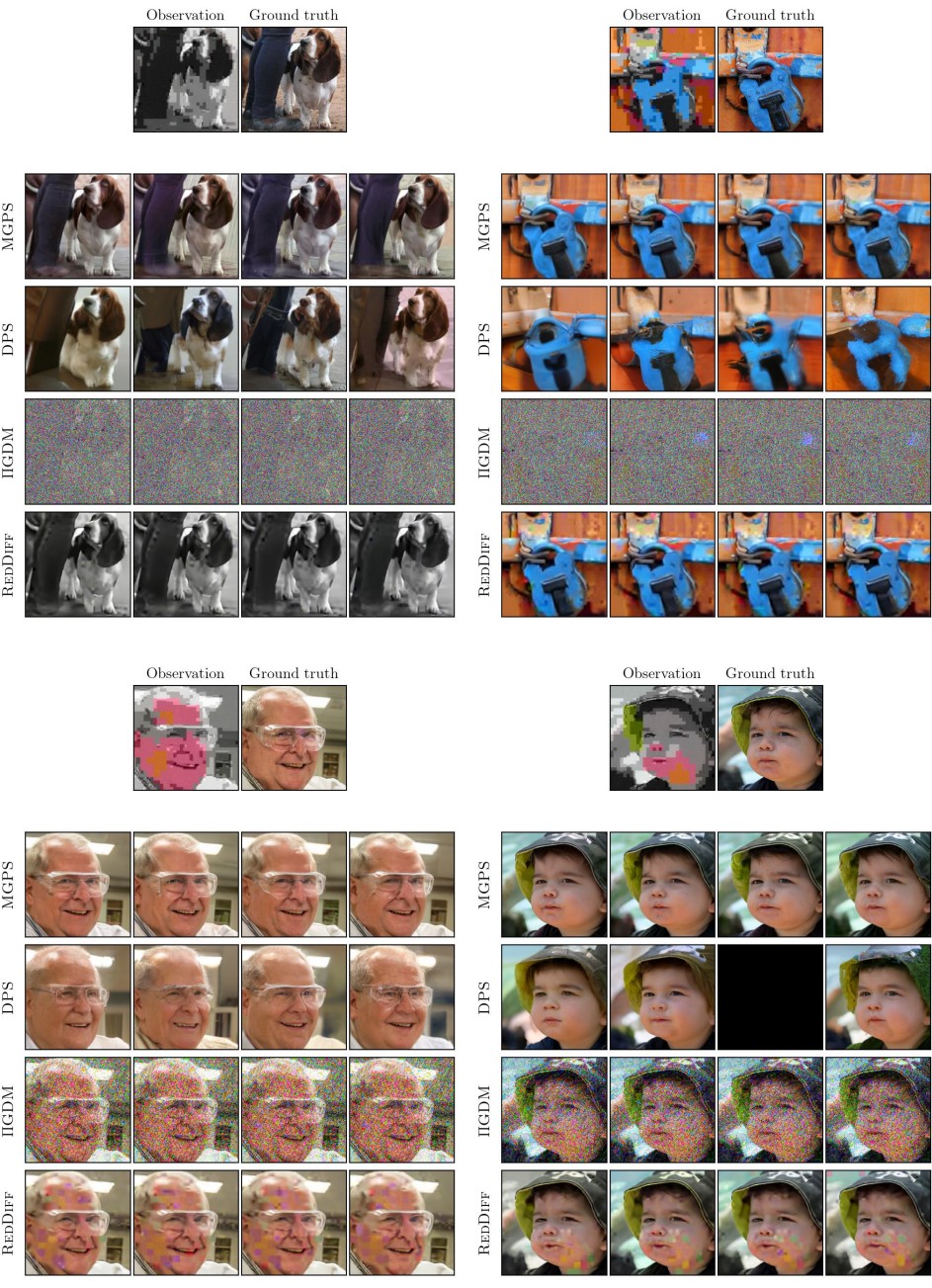

Figure 20: Sample reconstructions on JPEG 2 task.

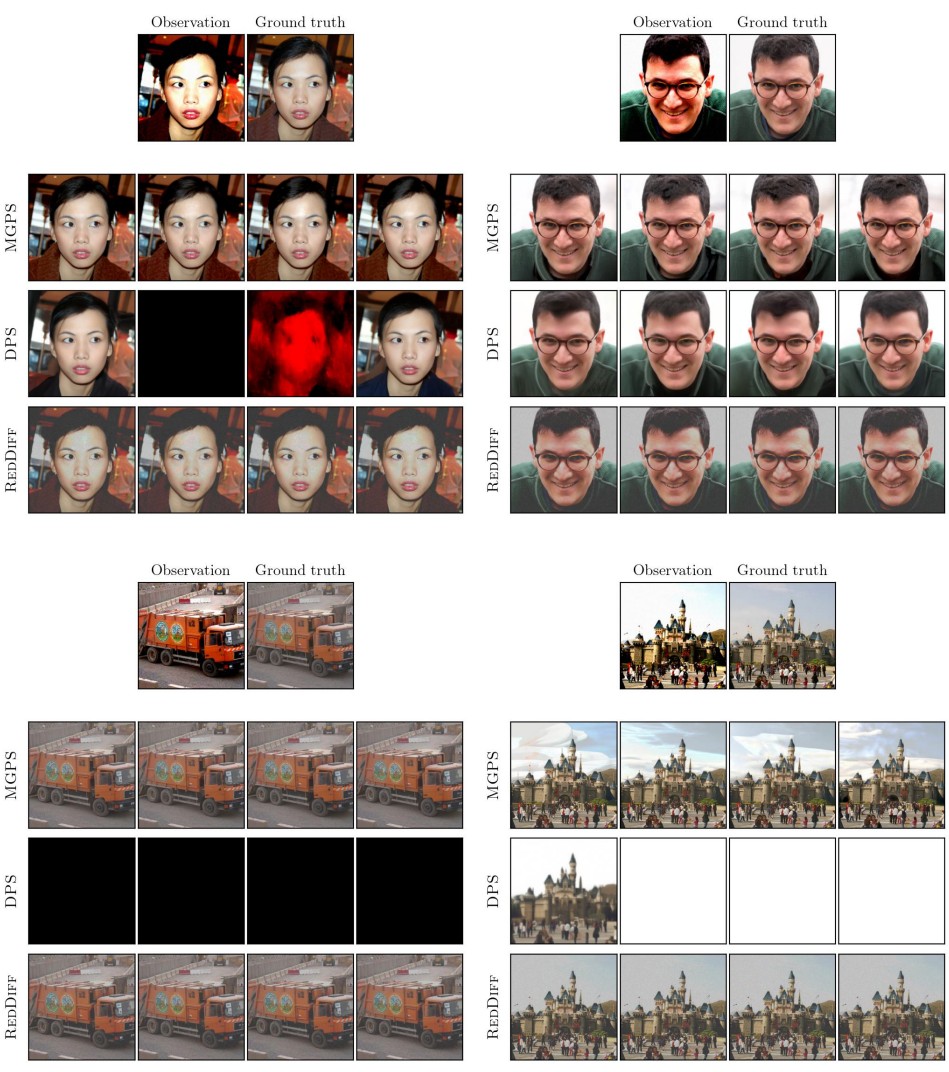

Figure 21: Sample reconstructions on high dynamic range task.

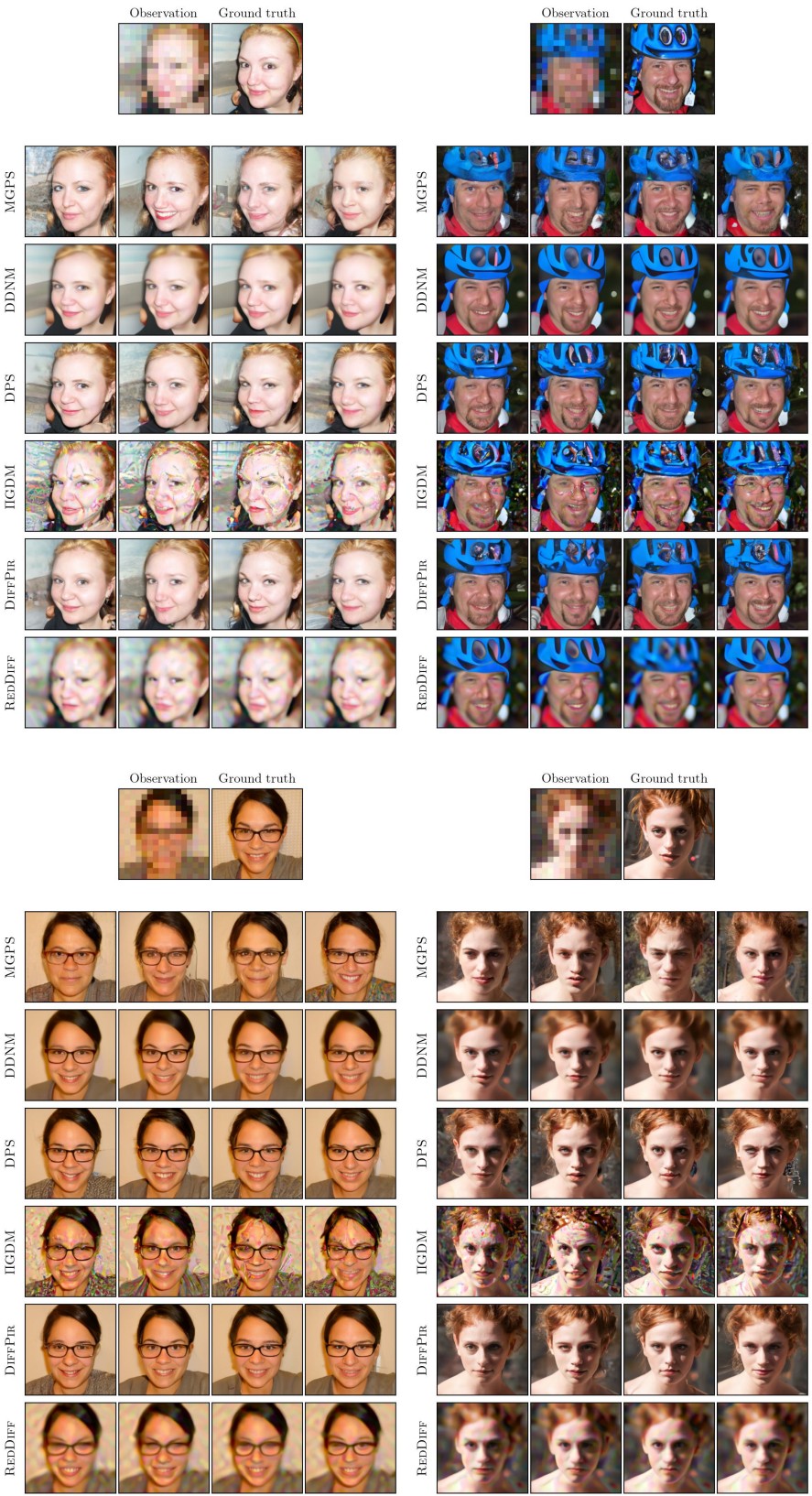

Figure 22: Sample reconstructions on SR (16×) task.

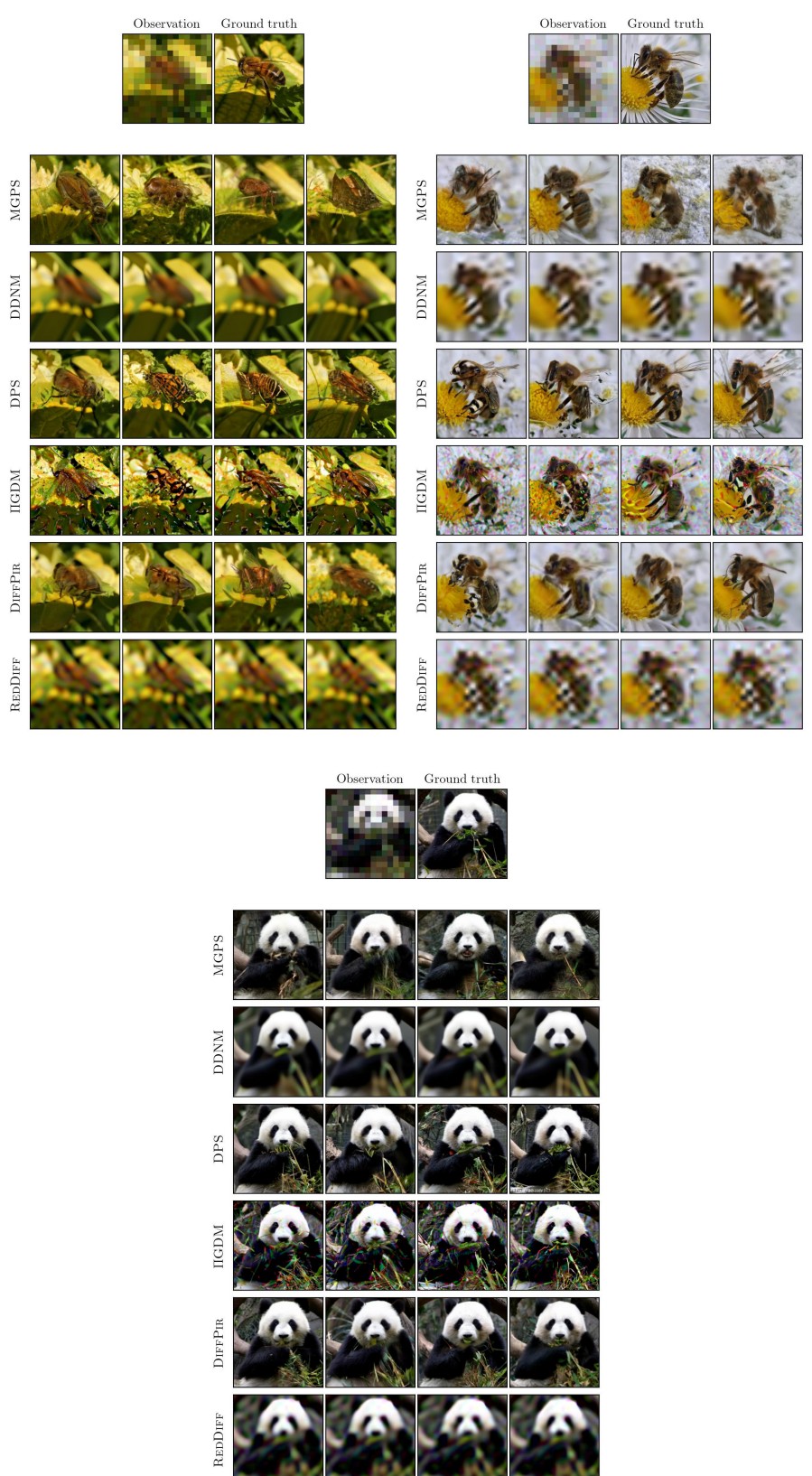

Figure 23: More sample reconstructions on SR (16×) task.

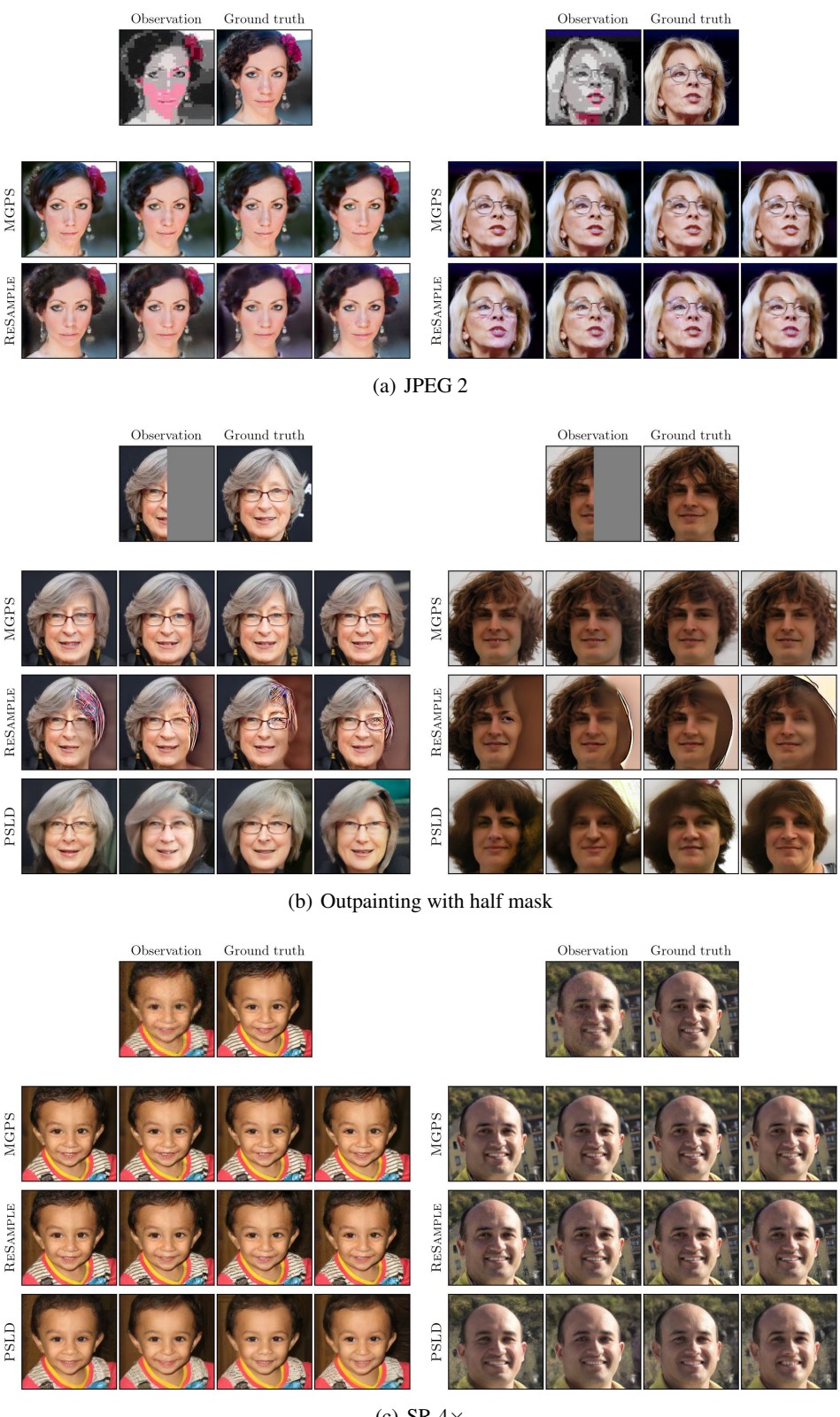

Figure 24: Sample reconstructions with latent diffusion models on FFHQ dataset.

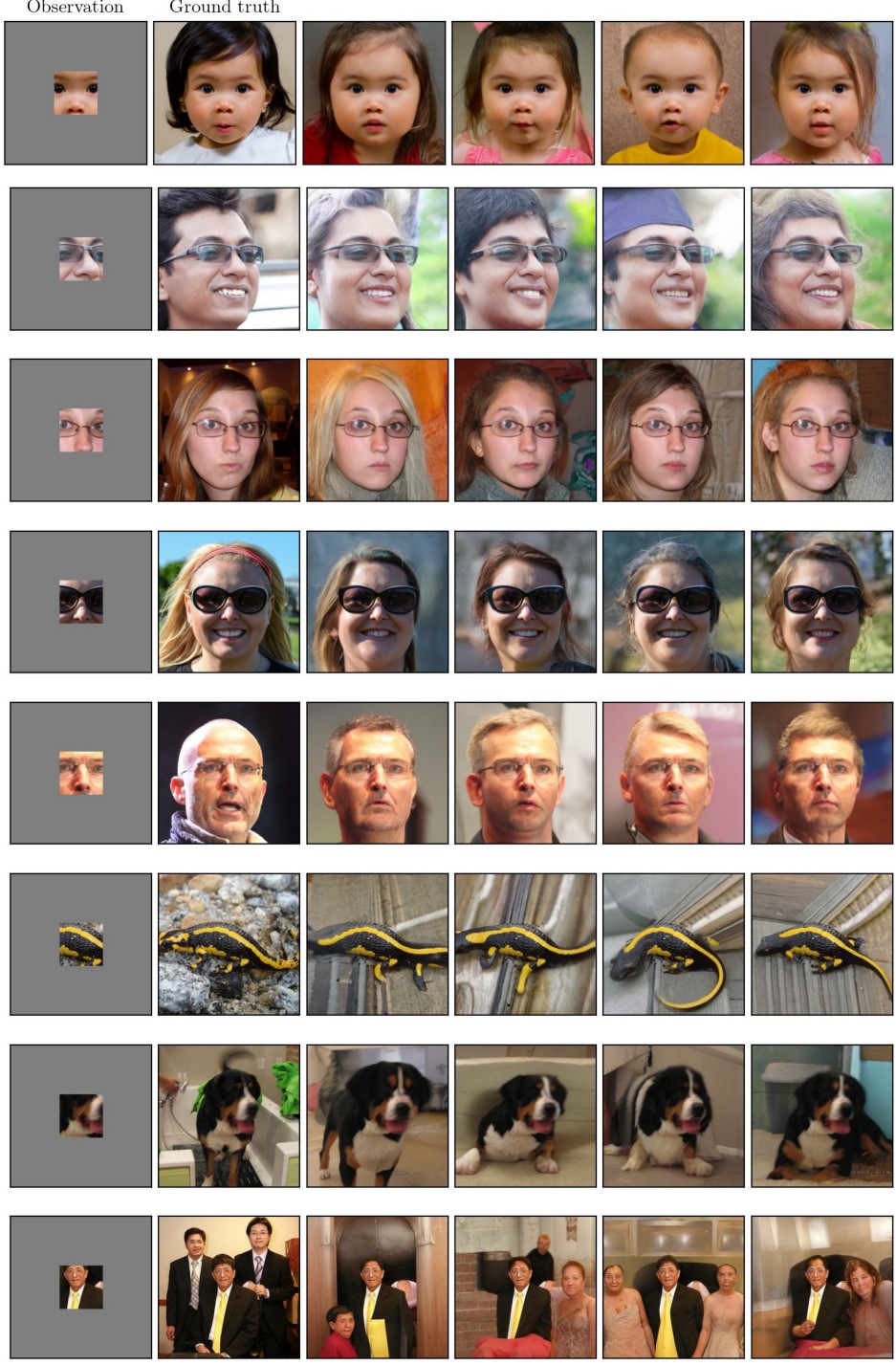

Figure 25: More sample reconstructions with MGPS on the expand task.

Observation    Ground truth

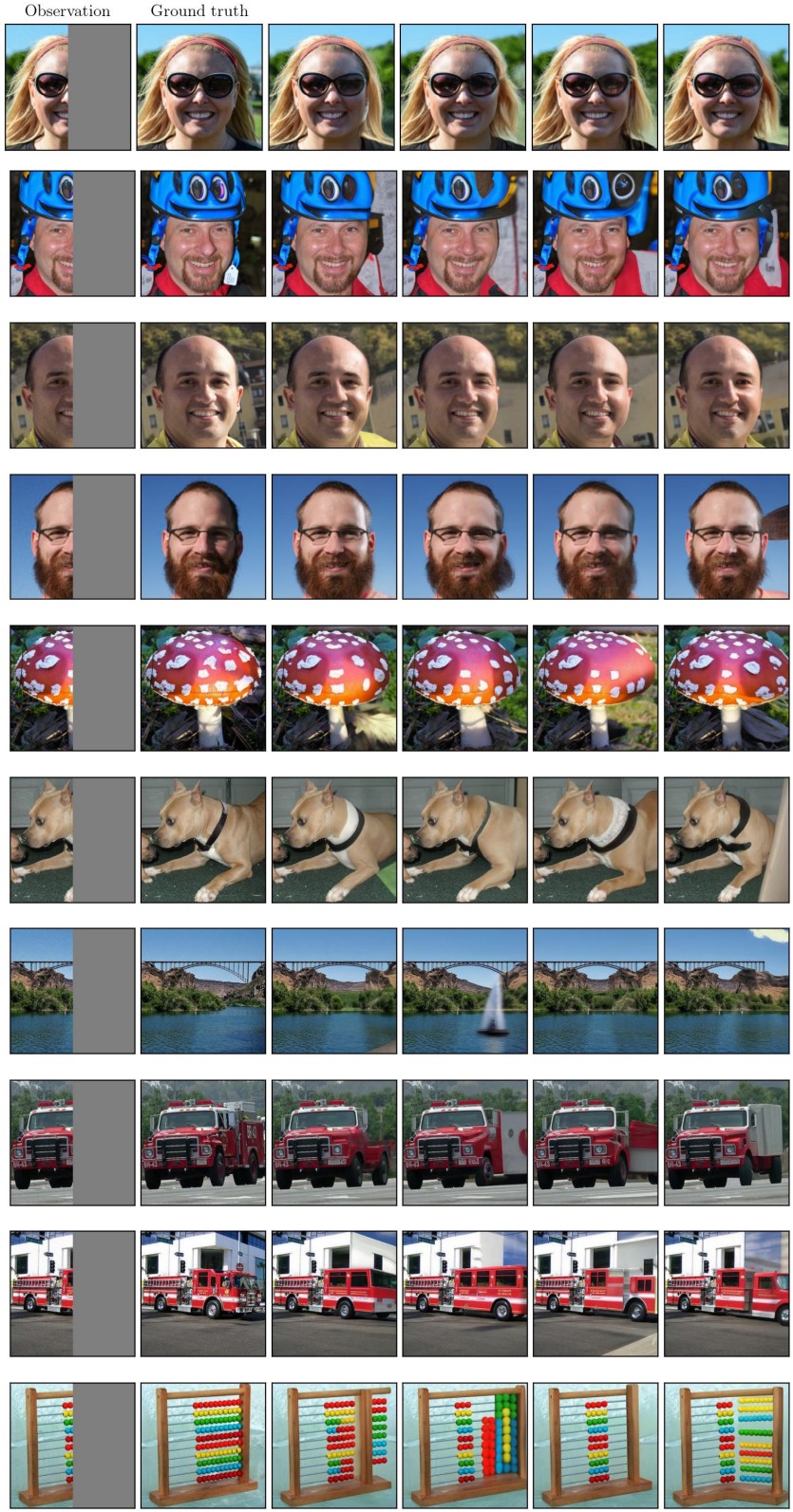

Figure 26: More sample reconstructions with MGPS and half mask.

Observation     Ground truth

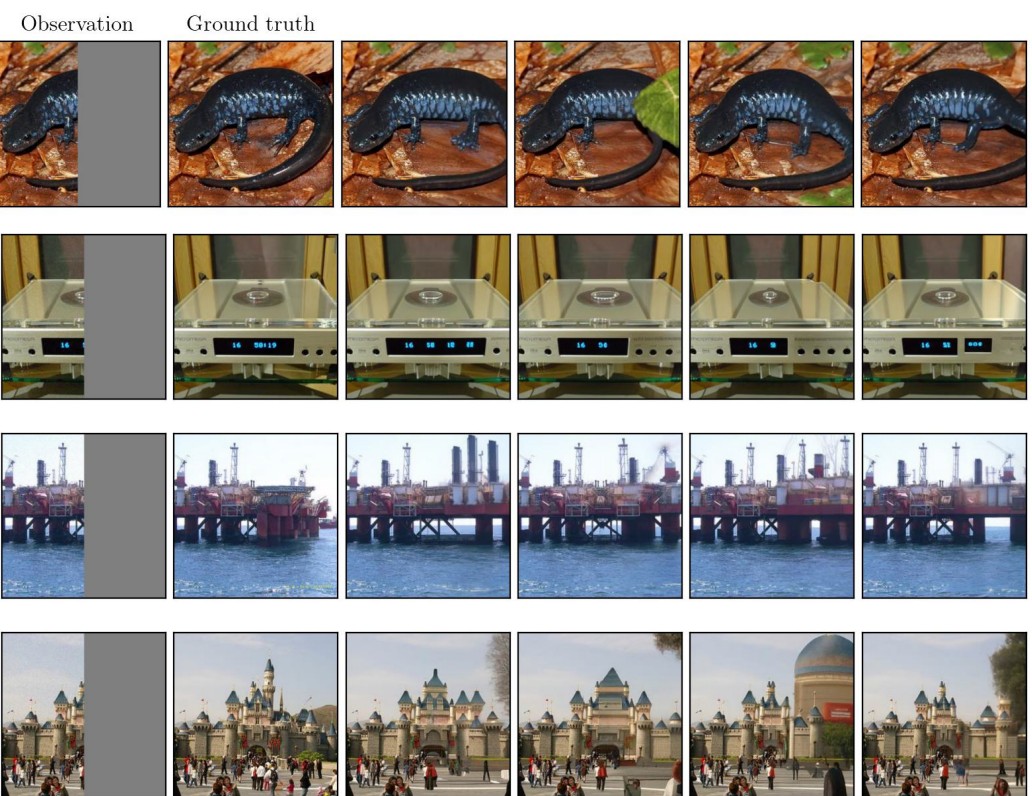

Figure 27: More sample reconstructions with MGPS and half mask.

