# OpenReview forum: "Variational Diffusion Posterior Sampling with Midpoint Guidance"
_ICLR.cc/2025/Conference — ICLR 2025 Oral_

### Official Review · Reviewer_HCev · 2024-10-21

**Soundness:** 4
**Presentation:** 3
**Contribution:** 4
**Rating:** 8
**Confidence:** 4

**Summary:**

The paper proposes MGPS, a variational inference approach to diffusion model-based inverse problem solving (DIS) in the standard guiding reverse diffusion setup. Two clever ideas are used:

1. Instead of using the DPS approx after denoising $k + 1 \rightarrow k$, denoise it up to $0 < \ell_k < k$, and use the posterior mean of $x_{\ell_k}$ instead of the posterior mean of $x_k$ to compute the DPS gradient. This way, you can control the trade-off arising in the approximation error for the denoising kernel and the likelihood computation.

2. For every reverse diffusion timestep $k$, use $j$ iterations of stochastic optimization for fitting a variational distribution for the reverse posterior kernel. The authors propose to use a Gaussian kernel but also optimize over the diagonal elements of covariance. This is different from the previous works where typically, an isotropic variance is used, probably with the exception of [1].

Overall, the paper is well-written, has a clear theory, and has great results. The reviewer especially appreciates the efforts that the authors made to make the experiments as fair as possible, carefully addressing the details that are often ignored. I do have some questions and concerns on the practical implementation of the method, and in some places, how to derive it. Nevertheless, I think the paper should be a clear accept.


**References**

[1] Peng, Xinyu, et al. "Improving Diffusion Models for Inverse Problems Using Optimal Posterior Covariance." ICML 2024.

**Strengths:**

1. The paper is well-written and relatively straightforward to follow.

2. When an approximation is used, the authors do a good job of explaining the rationale, by either showing it theoretically or experimentally.

3. MGPS is a good balance between a theoretically grounded solution and a practical solution, not requiring too much computation.

4. The results of image restoration tasks are clearly SOTA.

5. Experiments are complete. Numerical experiments on toy data, image restoration experiments, and ECG completion all indicate the superiority of the method.

**Weaknesses:**

1. Some parts of the derivation are unclear. The authors propose a variational distribution $\lambda^\varphi$ for $\hat{\pi}^\theta$ in (3.6), which, to my understanding, is already tractable. Since $\hat{g}^\theta$ is Gaussian from DPS, and $p_{\ell_k}^\theta$ is also Gaussian, then isn't $\hat{\pi}^\theta$ already a Gaussian? Why would one need an additional variational distribution to approximate this?

2. Adding on 1, I don't quite understand why including the diagonal terms of the covariance for additional optimization would induce better fitting, when $\hat{\pi}^\theta$ would have istropic covariance. Both these points maybe from my misunderstanding, but it should be clarified.

3. Balancing the denoising and likelihood approximation errors is interesting. Is it safe to say that the mainstream previous works that use something similar to (2.5) be considered as simply taking $\ell_k = k$? Or is it not directly comparable? I believe the authors could spend some more effort on clarifying the difference between the existing methods. There is a related works section, but the connections seem a bit vague. An additional appendix section could be nice.

4. The upper bound of the variational inference problem is defined and used, but it is not explained how this is derived.

**Questions:**

1. In Tab. 2, only the LPIPS values are reported. I understand that this is probably to save space, but I would recommend also including other standard metrics such as PSNR, SSIM, and FID.

2. MGPS is based on DDPM sampling. Would there be any way to incorporate faster solvers into this?

I am willing to further raise the score after clarification.

---

> ### Author Response · Authors · 2024-11-21
>
> Dear Reviewer HCev, we would like to thank you for taking the time to review our paper. We are happy that you found our paper interesting. Below we address your main concerns.
>
> **(W1)** While it is true that $p_{\ell_k | k+1} ^\theta(\cdot | x_{k+1})$  is Gaussian, we emphasize that even in the case of linear inverse problems, the DPS approximation yields  $\hat{g}_{\ell_k} ^\theta(x _{\ell})$  $=\mathcal{N}(y; A \hat{m}^\theta _{0|\ell_k}(x _{\ell_k}), \sigma^2 _y I)$ (or $p^\theta _{\ell_k}(y|\cdot)$ with the new notation). The mean of this Gaussian distribution is **non-linear** in $x _{\ell_k}$ which means that $\hat\pi^\theta _{\ell_k|k+1}(\cdot |x _{k+1})$ is not necessarily a Gaussian distribution. It would be the case only if the mean in the potential $\hat{g}^\theta _{\ell_k}(x _{\ell_k})$ were linear in $x _{\ell_k}$. Finally, further non-linearities in the mean of the potential arise when dealing with non-linear inverse problems as considered in the paper. Thus, the Gaussian variational approximation, or any sort of approximation, is necessary.
>
> **(W2)** This is a fair point. In our methodology we try to approximate, at each step, the conditional distribution $\pi_{\ell_k | k+1}(\cdot | x_{k+1})$ with $\ell_k$ much smaller than $k$. This distribution can be unimodal or multi-modal with anisotropic structure in the modes, as opposed to the transition $\pi_{k|k+1}(\cdot | x_{k+1})$ which is well-approximated when the covariance of the Gaussian approximation is isotropic. Let us give an example. Assume that the target distribution is $\pi = \mathrm{N}(\mu, \Sigma)$. We can compute exactly the covariance $Cov(X_\ell| X_{k+1})$ of the backward transition $\pi_{\ell|k+1}(\cdot | x_{k+1})$ for any $\ell \in [0:k]$. It is given by
> $$
> Cov(X_\ell| X_{k+1}) = (\alpha_\ell - \alpha_{k+1}) \bigg[(1 - \alpha_{k+1}) \Sigma^{-1} + \mathrm{I}\bigg]^{-1} + \frac{(1 - \alpha_\ell)(1 - \alpha_{k+1} / \alpha_\ell)}{1 - \alpha_{k+1}} \mathbf{I}
> $$
> It is seen that when $\ell$ is much smaller than $k$ the covariance can be very different from an isotropic one. To account for this, we allow more flexibility in our variational approximation by optimizing the diagonal terms and, in practice, we found that this improves the performance. Of course, optimizing over the full covariance matrix would be better, however, this is not feasible in practice because of the large  size of the matrix in high dimension setups.
>
> **(W3)** As far as we are concerned, the only method we know of that is close to our methodology when $\ell_k = k$ is the work in [1] that we mention in the related works section and explain how it is related. As for the other methods such as DPS, they do not exactly interpret as the case $\ell_k = k$ and more approximations are required. DPS cannot be framed as a special case of our algorithm but we can still obtain a special case that closely matches it. This is now thoroughly explained in Appendix C.3.
>
> **(W4)** We now explain how the upper bound is obtained in Appendix C.1.
>
> Regarding your questions:
>
> **(Q1)** We now include an extended table including the PSNR and SSIM values in Tables 7, 8 in the appendix, as well as in the new table 9 for latent diffusion.  Please note that initially we did not include the PSNR and SSIM as these provide ambiguous results that poorly reflect the actual performance of the algorithms. As an example, DDNM has the best PSNR and SSIM on the Half mask task (see tables 7 and 8), but it does not provide accurate reconstruction on ImageNet and FFHQ as seen in  figures 15 and 16. See also the reconstructions of DDNM in the figures 9 to 14 the paper [1]. The LPIPS on the other hand aligns well with the qualitative results that we display. As for the FID, it requires running all the algorithms and tasks on a large number of images (1000 at least) and this is prohibitively expensive to do in our case given the number of tasks/algorithms we consider on FFHQ/latent FFHQ, ImageNet as well as the ECG dataset for which we have trained our own diffusion model.
>
> **(Q2)** In order to extend our methodology to faster solvers, we require the equivalent of the identity (3.4) in the main paper. This is for example feasible for DDIM-type samples and we will add the derivation in the final version of the paper. As for other fast solvers such as EDM [2] it is not yet clear how they can be used within our methodology and we leave this for future research.
> Finally, please note that the other reviewers have requested some changes in the notations as well as in the presentation. We have implemented these modifications by making some important changes to section 3 so as to explain the method more intuitively. We have also added a figure to illustrate the idea behind our method.

---

> > ### Author Response · Authors · 2024-11-21
> >
> > [1] Zhang, Guanhua, Jiabao Ji, Yang Zhang, Mo Yu, Tommi S. Jaakkola, and Shiyu Chang. "Towards coherent image inpainting using denoising diffusion implicit models." (2023).
> > [2] Karras, Tero, Miika Aittala, Timo Aila, and Samuli Laine. "Elucidating the design space of diffusion-based generative models." Advances in neural information processing systems 35 (2022): 26565-26577.

---

> > > ### Author Response · Authors · 2024-11-25
> > >
> > > Dear reviewer HCev,
> > >
> > > We would like to thank you again for your valuable comments, which have greatly helped us improve the paper.
> > >
> > > We are pleased to inform you that we have had the opportunity to rerun MGPS on 1000 images for the five tasks, along with the two closest competitors for each task, using the three priors (ImageNet, FFHQ, FFHQ latent). This allowed us to calculate the FID as you suggested in your comments, and we can confirm that MGPS still outperforms the competitors on the FID. The results are presented in our global response "Finalized Experiments", in the comment section.
> > >
> > > As the discussion period is coming to an end, could you please let us know if all your concerns have been addressed? We would be happy to respond to any remaining questions or remarks if necessary.

---

> > > > ### Comment · Reviewer_HCev · 2024-11-27
> > > >
> > > > I would like to thank the authors for a thorough and complete response. All my concerns have been resolved. I believe the paper is a valuable add to the community. I will raise my score to 8.

---

### Official Review · Reviewer_eMe4 · 2024-10-28

**Soundness:** 4
**Presentation:** 3
**Contribution:** 3
**Rating:** 8
**Confidence:** 4

**Summary:**

The authors propose 'midpoint guidance posterior sampling,' where the reverse diffusion process is decomposed into two steps:

1. Denoising the current diffusion measurements into a 'midpoint' state.
2. Renoising the midpoint measurements to obtain the next diffusion iterate.

The denoising approach consists of approximating a Gaussian variational approximation in conjunction with the DPS guidance proposed by Chung et al to sample an estimated image. The variational approximation is learned at each diffusion timestep.

The renoising stage appears to be the typical DDIM/DDPM computation of the next diffusion iterate.

This approach of midpoint guidance achieves very strong empirical performance on a wide variety of problems/datasets and has strong theoretical foundations.

**Strengths:**

S1) The approach is well-motivated and has strong theoretical backing, with a novel theoretical result included in Appendix A.3.

S2) The authors evaluate many different problems on multiple datasets, achieving relatively strong performance across the board.

S3) The paper is well-written and reasonably easy to follow (although I have some gripes with notation, outlined in the 'Weaknesses' section below.

S4) Many great visuals, especially in the appendices.

S5) The authors provide very detailed and explicit implementation details for their competitors. I think that this is of vital importance and appreciate the efforts made by the authors to share these details.

**Weaknesses:**

At a high level, I quite like this work. However, as described below, I feel that the experimental results are incomplete.

W1) Even though the paper is well-written, and all mathematical elements check out (at least to me), the notation makes all of the math in Section 3. In particular, the differentiation between scalar and vector quantities is not sufficient. I would suggest that the authors make vector quantities boldface (e.g., $\boldsymbol{x}$). This would make things much easier to read.

W2) While I appreciate the robust slate of experiments, using test sets of size 50 (at least for FFHQ, ImageNet) is not sufficient for two reasons:
1. I would argue that the standard test set size for any diffusion inverse solver is 1k images. This seems to be the accepted number, and large enough to truly understand model performance. A test of size 50 is just not sufficient for acceptance to a conference like ICLR. I feel that the results would be more convincing if there were fewer experiments with a larger test set. Note that I am explicitly speaking on the image datasets, where samples are plentiful. There are plenty of problems where there is not enough data to have a test set comprised of 1k samples and that is fine. To summarize: I think that fewer experiments are not a bad thing if it means more robust testing.
2. Only using 50 test samples does not enable reliable computation of FID (a metric that is noticeably absent from the paper). FID is a standard metric when evaluating diffusion inverse solvers. With it absent, the experimental results are incomplete.

W3) I also think that the experimental results are incomplete due to the lack of a pixel-space quality metric. The authors argue against PSNR/SSIM in Section 4, but I disagree. I think that LPIPS is a great choice, but that PSNR is still a necessity when testing because similarity in the pixel-space matters too. If the samples don't respect the measurements, that is a problem and LPIPS may not catch it since it is a feature-based metric.

To summarize W2 and W3, I think that the authors need to reconsider their evaluation of problems which use the image datasets. I would suggest:
1. 1k test samples.
2. Computing PSNR and FID in addition to LPIPS.

These changes are critical to fully understanding model performance. Without this sufficient experimental evaluation, I cannot recommend this paper for acceptance.

**Questions:**

See weaknesses.

---

> ### Author Response · Authors · 2024-11-21
>
> Dear Reviewer eMe4, we would like to thank you for your comments and positive feedback regarding our paper.. Below we address your concerns:
>
> **(W1)** We welcome your suggestion for using boldface for the vectors. We have now implemented this in the updated version of the paper that we have uploaded. We would also like to mention that following the suggestions by the other reviewers, we have also changed some of the notations to improve the readability and revised significantly Section 3 of the paper in order to make the presentation more intuitively accessible. In addition, we have added a Figure to illustrate the intuition.
>
> **(W2)** We understand your concern about the number of test samples. In our original submission, we focus on illustrating the wide applicability of our method to different image reconstruction tasks, considering 10 different tasks. However,  given our limited computational resources, we had to limit the number of  test samples to 50.
> Indeed, for instance, running a single task for 3 algorithms on 1k images would have required 600 GPU hours. Additionally, running fewer tasks would not highlight the strength of MGPS, which is a general algorithm that can be applied to different problems and modalities.
> For the time being we have sought a compromise and we have increased the sample size to 300 on a subset of the experiments, including 95% confidence intervals. The new results align with our initial findings; see the general comment above. Note that we have run these experiments on what we consider to be the most difficult tasks but we are currently running experiments reaching 1000 samples for the final version of the paper. Given the confidence intervals, we expect to see no significant difference with what we currently have.
> Finally, we would like to insist that our set of experiments is designed (i) to show that our method provides a good approximation of the posterior distribution, and (ii) to demonstrate its wide applicability. Regarding the first point, we compare with the **exact** posterior in the Gaussian mixture example. Regarding the second one, we have considered 10 various tasks for the image experiments, compared against 7 competitors, and used both pixel space and latent space diffusion adding up to 3 different priors. Then, we have also considered the ECG data experiment, for which we have trained our own generative model, to show that our method works on different modalities.
> We ensured  to run the experiments for all the competitors and refrained from using the values given in the papers and so this effectively limited the number of images we initially considered for evaluation, due to our limited computational budget.
>
> **(W3)** We also include the PSNR/SSIM and have updated the tables in the paper; see Table 7 and Table 8. Regarding these metrics, we would like to insist that our method performs well on the three metrics at the same time. On the other hand, while DDNM achieves the best PSNR, it does not perform well in LPIPS. In fact, higher PSNR does not translate to good reconstructions as evidenced by Figure 13 and Figure 15. Similarly, other authors have observed the same reconstructions for DDNM; see for example the figures 9 to 14 in [1].
> Finally, we did not include the FID since we need a much larger number of samples to get a robust estimate. Still, since we are now in the process of increasing the sample size up to a regime where accurate estimation of FID is possible, we will make sure to include it in the final version of the paper once the experiments are finished running.
> We believe that by considering different types and extensive experiments, especially a toy experiment in which the posterior is available in closed form, we have provided strong evidence in favor of our method.
>
>  [1] Zhang, Guanhua, Jiabao Ji, Yang Zhang, Mo Yu, Tommi S. Jaakkola, and Shiyu Chang. "Towards coherent image inpainting using denoising diffusion implicit models." (2023).

---

> > ### Comment · Reviewer_eMe4 · 2024-11-21
> >
> > Thank you to the authors for addressing my concerns. With respect to the mentioned compute issue, my thinking is that the paper shouldn't be submitted until sufficient evaluation has been completed; this rebuttal period is not the time to be doing that.
> >
> > Even so, I feel that the authors have partially addressed my concerns and have promised to fully address my concerns by the final version of their paper. With this in mind, along with the other reviews and author replies, I raise my scores for presentation to 3 and soundness to 4. I also raise my overall score to 6. I will also note that if all of my concerns had been addressed in this rebuttal period (not just promised) I would have raised my overall score to an 8.

---

> ### Author Response · Authors · 2024-11-23
>
> Dear Reviewer,
>
> Thank you for your prompt response and for raising your score. We understand your concern about seeing some experiments with 1000 images, especially for computing the FID, as "fewer experiments are not a bad thing if it means more robust testing." To address this, we have selected two tasks that we consider the most challenging and have tested our algorithm against the closest competitors on 1000 image on the three priors: half mask and JPEG. Below, we provide the four metrics (including the FID), and it is evident that MGPS outperforms all reported competitors on the FID by a noticeable margin.
> We hope this addresses your concerns. Please let us know if there are any other tasks you would like us to evaluate before the end of the rebuttal period to address all of your concerns and ensure all promises are fulfilled.
>
>
> **HALF MASK**
>
> | Dataset | Metric | MGPS | DDNM | DIFFPIR |
> |----|-----|-----|----|----|
> | FFHQ  | **FID** | **27.01** | 38.62 | 45.20  |
> | FFHQ  | LPIPS |**0.19**+/-0.003| 0.23+/-0.004 |0.25+/-0.004  |
> | FFHQ  | PSNR | 15.86+/-0.12 | **16.27**+/-0.13 |16.14+/-0.14  |
> | FFHQ  | SSIM | 0.70+/-0.003| **0.74**+/-0.004 |0.72+/-0.005  |
> | ImageNet  | **FID** | **40.09** | 50.02 |56.99  |
> | ImageNet   | LPIPS | **0.30**+/-0.004 | 0.38+/-0.005 |0.40+/-0.005  |
> | ImageNet | PSNR |15.01+/-0.10 | **16.02**+/-0.12 |15.75+/-0.12  |
> | ImageNet   | SSIM | 0.63+/-0.004| **0.68**+/-0.005 | 0.67+/-0.005  |
>
> **JPEG2**
>
> | Dataset | Metric | MGPS | DPS | Reddiff |
> |----|-----|-----|----|----|
> | FFHQ  | **FID** | **31.60** | 87.58 |108.54  |
> | FFHQ  | LPIPS | **0.15**+/-0.003 | 0.37+/-0.02 | 0.33+/-0.005  |
> | FFHQ  | PSNR | **25.20**+/-0.09 | 18.96+/-0.19 | 24.47+/-0.08  |
> | FFHQ  | SSIM | **0.73**+/-0.005 | 0.55+/-0.02 |0.70+/-0.004 |
> | ImageNet  | **FID** | **61.35** | 128.77 |92.84  |
> | ImageNet  | LPIPS |**0.40**+/-0.009 | 0.60+/-0.014 |0.47+/-0.008  |
> | ImageNet  | PSNR | 22.15+/-0.10 | 16.66+/-0.14 |**22.19**+/-0.09  |
> | ImageNet  | SSIM | **0.60**+/-0.008 | 0.41+/-0.015 | **0.60**+/-0.007 |
>
> **Latent diffusion**
> | Task | Metric | MGPS | Resample |
> |----|----|----|----|
> | Half mask  | **FID** | **49.45** | 66.55  |
> | Half mask | LPIPS | **0.26**+/-0.004 | 0.30+/-0.003  |
> | Half mask  | PSNR | **15.56**+/-0.13 | 14.73+/-0.09  |
> | Half mask  | SSIM | **0.69**+/-0.004 | 0.67+/-0.003 |
> | JPEG2  | **FID** | **45.07** | 65.30  |
> | JPEG2  | LPIPS | **0.21**+/-0.004  | 0.26+/-0.005  |
> | JPEG2  | PSNR | 24.64+/-0.08 | **24.77**+/-0.09 |
> | JPEG2  | SSIM | **0.71**+/-0.004 | 0.65+/-0.005 |

---

> > ### Author Response · Authors · 2024-11-25
> >
> > Dear Reviewer eMe4,
> >
> > We would like to thank you for considering our experiments during this rebuttal and for increasing our score once again. We are pleased to inform you that we have had the opportunity to finalize the experiments on 1000 images for the 3 other tasks: motion blur, nonlinear blur, and high dynamic range, using the three priors (ImageNet, FFHQ, FFHQ latent). We report that MGPS still outperforms the other competitors on the FID (see Main Comment "Finalized Experiments" above).
> >
> > We appreciate your feedback, which has significantly helped to improve the robustness of our study. We believe that all your concerns have now been addressed, but we would be happy to respond to any remaining questions or remarks before the end of the rebuttal period.

---

### Official Review · Reviewer_bHyn · 2024-11-03

**Soundness:** 4
**Presentation:** 3
**Contribution:** 3
**Rating:** 8
**Confidence:** 5

**Summary:**

The paper proposes a novel midpoint guidance strategy for posterior sampling of diffusion models. The main idea is to first move to a mid-point state l_k that is a function of k for guidance, and then noise back to obtain X_k unconditionally. This seems to solve issues associated with other SOTA approaches that perform the guidance based on variations of Tweedie's formula. Extensive experiments are provided, along with comparison to SOTA methods. Improvements are shown in almost all inverse problems, and notably for nonlinear inverse problems.

I think the paper has good merit, but its exposition is overly complicated and notation does not match the rest of the literature. These dampened my enthusiasm, but I'm happy to reevaluate after the authors respond.

**Strengths:**

- The idea of midpoint guidance is novel, and intuitively appealing to solve the issues associated with guidance issues especially at the early stages of the diffusion process.
- Very thorough evaluation.
- Results are good, showing improvement over SOTA methods, including both DDMs and LDMs (that are commonly used in these applications).
- Multiple nonlinear inverse problems are studied, an area where other posterior sampling methods have issues. A good level of improvement is shown in these applications.
- Performance also evaluated across NFEs.
- Good sample variety is shown.

**Weaknesses:**

- Unfortunately, the exposition is overly complicated. The idea can be explained much more clearly, but also partly due to non-standard notation, the gist does not come across easily. Section 3 would benefit from a substantial rewrite that changes the notation (please see below), and highlights the main ideas, and potentially even including a figure to show the midpoint guidance idea.
- Similarly, the notation does not match rest of the literature on posterior diffusion sampling for inverse problems. For instance, the posterior is denoted by \pi(x), which is written as a marginal, though this should depend on the observations y. Similarly g(.) in (2.1) should be conditional on y, but this is not done either. This propagates throughout the paper, and makes it hard to appreciate the contributions. Please use standard notation to match other existing works.
- The 50 randomly selected evaluation points for FFHQ and ImageNet is a bit unconvincing. There are standard validation sets that are publicly available for both databases, and these are commonly used in other papers (e.g. DPS, PGDM). Please report your results over a larger set (& also explain how random selection was done).
- Unclear why only LPIPS is provided. PSNR/SSIM must be provided as well. I understand improvement may not be uniform for those metrics, but this should be up to the reader to figure out.
- DPS is typically run with NFE = 1000, but it was used for N = 300 in this paper for comparison. This may further close the LPIPS gap.

The following points are not really weaknesses, but I wanted to note them:
- The ECG application comes out of nowhere. Without enough motivation and knowing the difficulty of the task, the contribution here is a bit hard to appreciate.
- There are also two additional references that may be of interest:
a) arXiv:2402.02149, which is similar to Boys et al in spirit
b) arXiv:2407.11288 (ECCV 2024), which learns w_k in (2.5) and also uses a diagonal approximation to circumvent vector-Jacobian calculations

**Questions:**

Following up on the weaknesses:
- Why was a different notation used compared to existing works on posterior sampling for inverse problems?
- Why was only 50 randomly selected evaluation points used for FFHQ and ImageNet? How was this random selection done?
- Why was only LPIPS used for evaluation, and other standard metrics like PSNR/SSIM were not reported?
- Why was DPS run for 300 steps instead of the more standard 1000?

---

> ### Author Response · Authors · 2024-11-21
>
> Dear Reviewer bHyn, we would like to thank you for taking the time to review our paper. We are happy you found that our paper has good merit. Below we address your concerns.
>
> **(W1-2)** Thank you for this feedback. We have implemented all your suggestions in the revised version paper which is now updated in openreview. Notably we have:
> modified Section 3 so that the idea is explained more intuitively. Now we start the paragraph “Midpoint decomposition” with a straight-to-the-point explanation of what our method tries to achieve. The explanation is accompanied with a visual explanation in Figure 1. We thank the reviewer for this good point.
> changed the notations. The $g$ function has been replaced by a likelihood function $p(y|\cdot)$. Reviewer eMe4 has also suggested using boldface for the vectors and we believe that it also improves the readability. You have also suggested adding the $y$ conditioning to $\pi$, which is a sensible idea. Still, if we implement this modification we would also need to add it to the conditional distributions to be consistent in the notations, i.e. $\pi_{i|j}(x_i | x_j)$->$\pi_{i|j}(x_i | x_j, y)$ and such. As we believe that this will lead to notational overload we decided not to implement it.
> Please check out the modifications in the paper and let us know if you think we should make any further adjustments.
>
> **(W3-4)** We have prioritized running a more  diverse set of experiments (10 tasks on images, 2 tasks on ECG) on the 7 competitors at the cost of using a smaller number of sample images, due to computational constraints. In the updated version of the paper we provide extended results on the most difficult tasks and on a larger set of images; more precisely we have drawn without replacement 300 images from both the ImageNet and FFHQ validation datasets. Finally, following your suggestion we now also include PSNR and SSIM. These new results are consistent with the initial findings of the paper. Please see the main comment. Our method performs well on the three metrics at the same time. While competitors such as DDNM or RedDiff may have in some cases a larger PSNR or SSIM, they still exhibit a large LPIPS, resulting in subpar reconstructions, as evidenced in the appendix Figures 13, 14 and 15 for example.
>
> **(W5)**  In all the experiments we use DPS and PGDM with **n=1000** steps. This can be checked in the configuration files of the code we have provided in the supplementary material, namely files ``dps.yaml`` and ``pgdm.yaml`` in  ``configs/experiments/sampler/`` folder. The sentence in the experiment Section was ambiguous and is now fixed. We meant that the runtime of our algorithm with $n=300$ steps is larger than the runtime of DPS and PGDM.
>
> Thank you for the references, we were aware of only one of these works.
>
> **Note on ECG:**
>
> Thank you for your comment. We understand that the introduction of the ECG application may seem sudden. To clarify, the aim of this experiment is to explore the application of posterior sampling algorithms beyond classical image-related tasks to demonstrate their generalizability and societal impact.
>
> **Medical Context:** Cardiovascular diseases account for approximately 1/3 of global deaths. Better detection could improve management of these conditions. Wearable devices like SmartWatches have a good potential for improving diagnosing cardiovascular diseases, as patients may experience brief episodes of symptoms (e.g., paroxysmal Atrial Fibrillation) that may not be detected during a doctor's visit. Using these monitors for extended periods can help ensure a timely and accurate diagnosis. However, they only provide a partial view (lead I instead of 12 leads) of cardiac electrophysiology.
>
> **Challenge of Classification from Incomplete ECGs:** A recent study showed that the Apple Watch correctly identified atrial fibrillation (AF) in only 34 out of 90 episodes [1]. We propose completing ECGs to 12 leads using posterior sampling algorithms to address the issue of incomplete ECGs. To our knowledge, we are the first to use ECG completion with posterior sampling for detecting anomalies in incomplete ECGs.
> We hope this clarification helps to better understand the motivation behind our choice of the ECG application and the importance of our work for the posterior sampling community. We will add a comment on this in the final version of the paper. Please feel free to suggest any additional changes you deem necessary.
>
>
> [1] Dhruv R. Seshadri  and Barbara Bittel  and Dalton Browsky  and Penny Houghtaling  and Colin K. Drummond  and Milind Y. Desai  and A. Marc Gillinov. Accuracy of Apple Watch for Detection of Atrial Fibrillation, Circulation (2020), https://doi.org/10.1161/CIRCULATIONAHA.119.044126

---

> > ### Comment · Reviewer_bHyn · 2024-11-22
> >
> > The authors have addressed almost all my concerns, adding more experiments and explaining DPS NFE + ECG application. I will change my score to an 8.

---

> > > ### Author Response · Authors · 2024-11-25
> > >
> > > Dear Reviewer bHyn,
> > >
> > > We would like to thank you for responding to our rebuttal and for re-evaluating our work. Your comments have significantly helped us to improve the quality of our paper. We would be happy to respond to any remaining questions or remarks before the end of the rebuttal period.

---

### Official Review · Reviewer_U6Fn · 2024-11-05

**Soundness:** 3
**Presentation:** 3
**Contribution:** 3
**Rating:** 8
**Confidence:** 3

**Summary:**

This paper introduces a novel diffusion-based posterior sampling method called Midpoint Guidance Posterior Sampling (MGPS) to address Bayesian inverse problems. In cases where denoising diffusion models (DDMs) are used as priors, MGPS aims to approximate the posterior while balancing the complexity between guidance and prior transition terms. The method leverages an intermediate midpoint state to improve posterior approximation and incorporates a Gaussian variational approximation for additional flexibility. MGPS is validated on linear and nonlinear inverse problems across various domains, including image and ECG signal reconstruction, and outperforms several state-of-the-art methods.

**Strengths:**

1. The approach introduces a midpoint guidance mechanism that provides a novel trade-off for guidance and complexity of the learned transition term in diffusion models. This make it different from other diffusion-based posterior sampling methods.

2. The method is well-justified with mathematical rigor. The decomposition of the backward transition and the use of Gaussian variational approximations.

3. MGPS is extensively evaluated across both synthetic and real-world tasks, such as Gaussian mixture sampling, image super-resolution, and ECG imputation. Experimental results show significant improvements in reconstruction quality over baseline methods.

4. The paper is generally well-organized, providing clear problem definitions, methodological details, and experimental results. Detailed explanations and algorithmic steps (Algorithm 1) support reproducibility.

**Weaknesses:**

1. More exploration of $\eta$ and the midpoint sequence's impact on different task types and data complexities would clarify MGPS's adaptability across scenarios. The authors showed the effect of $\eta$ in the Gaussian toy example. It would be interesting to see $\eta$'s influence in other tasks.

2. The trade-off introduced by the midpoint state could be theoretically explored further. The authors mention the need for tuning this midpoint sequence but provide limited theoretical insights on why this works well. For instance, the bounds on the approximation errors may be further derived, or the balance between the prior and guidance terms could be analyzed with the midpoint sequence.

3. Concerns about the correct implementation of other compared methods remain for me, as those methods perform in Figures 13 and 15 worse than those in the original publications related to some image domain tasks. Or are those methods undertuned?

**Questions:**

1. Would there be an optimal mid-point schedule for $l_k$, which allows fewer diffusion model evaluations and comparable performance? Or better technique can be used to find the optimal schedule? Or can we suggest a criteria for evaluating the trade-off between performance and computational cost?

2. What is the key reason that makes mid-point guidance perform better? When the same prior is applied, what difference does the midpoint sequence make to sample algorithm as difference stages of the posterior distributions?

---

> ### Author Response · Authors · 2024-11-21
>
> Dear Reviewer U6Fn, we would like to thank you for taking the time to review our paper and for the positive feedback. Below we address your concerns.
>
> **(W1)** We agree with the reviewer that it is valuable to explore the impact of $\eta$ on other tasks and to obtain further theoretical results. The example of Gaussian prior with linear inverse problem is appealing as it ensures that all terms (guidance and denoising densities) remain Gaussian, enabling an explicit form of the posterior to be obtained as shown in Appendix B. However, this advantage does not extend to more general priors and nonlinear inverse problems, where analysis becomes more complex, even in relatively simple cases such as Gaussian mixtures. While working on the image experiments, specifically on the ImageNet dataset, we found empirically that the optimal values ($\eta \approx 0.5)$ we obtained in the Gaussian example also bring significant improvements for the image reconstructions, compared to setting $\eta \approx 1$ or $\eta \approx 0$. We are currently working on adding these empirical findings, and we will include the same in the appendix of the paper as soon as they are completed.
>
> **(W2)** A theoretical analysis of the algorithm would definitely be valuable. Deriving the dependence of the Wasserstein-2 distance on the choice of the sequence, which could then be optimized, is challenging even in the simple Gaussian case. In the more general case, deriving a bound on the KL divergence between the posterior and the marginal of the surrogate model that is informative wrt the choice of the midpoint sequence, first requires a proper study of the DPS approximation. Specifically, we must bound the discrepancy between the true potential $g_k$ (now $p_k(y|\cdot)$ with the new notation), and its approximation $g(m^\theta _{0|k}(x_k))$. Then, we would need to use this analysis to bound the one-step KL divergence between the true posterior transition and the surrogate transition. We have already taken some steps in this direction and believe that further technical and non-trivial work is required. Overall, we believe that such an analysis is a very good future direction of research and is out of the scope of the current paper, whose aim is to deliver a convincing proof of concept.
>
> **(W3)** We would like to assure the reviewer that we prioritized using the code and hyperparameters provided by the authors of the original papers and fine-tuned each method for each dataset as needed. In some instances, we have even contacted the authors to check that we are correctly using their code. Furthermore, we have included the code used for our experiments with our submission.
> First, while it may appear that some of the methods underperform on some tasks/images compared to the original publications, as you point for Figures 13 and 15, please note that they still produce competitive reconstructions on others; see for example Figure 15, 18, and 23 in the updated version of the paper.
> Second, we would like to stress that other papers report similar reconstructions on the same tasks. For example, see Figure 10,12 in [1], and Figure 9, 10 in [2], which display similar patterns.
> That being said, we would also like to stress that these discrepancies appear more frequently on the ImageNet dataset than the FFHQ one. This can be explained by the fact that  ImageNet is notoriously challenging due to its diversity, encompassing 1000 classes. This also seems to happen on one of the most difficult tasks, namely the half mask one.
> The reviewer is right in pointing out these disparities. In the updated version of the paper, which you can check out, we comment on the same; see Section D.6.

---

> > ### Author Response · Authors · 2024-11-21
> >
> > Regarding your questions:
> >
> > **(Q1)** Our choice of the sequence of midpoints is primarily based on a heuristic hinted at by the Gaussian example. Empirically, we have observed that using an adaptive sequence of midpoints leads to improved reconstructions, please refer to hyperparameters setups in Table 6 for FFHQ LDM. Therefore, we acknowledge that there likely exists an optimal sequence of midpoints that could yield better reconstructions with fewer diffusion steps. However, it is essential to be able to quantify the approximation error as a function of the sequence of midpoints beforehand in a way that is both numerically tractable and enables "sufficient flexibility" when manipulating the equations. As we highlight in the section "Limitations and Future Directions", developing methods to assess the impact of the sequence of midpoints is a promising direction for future research.
> >
> > **(Q2)** In our opinion, the main driving factor that makes the midpoint guidance competitive is the fact that at each step of the diffusion process we use the approximation of the guidance term at the step $\ell_k << k$, which has a smaller approximation error compared to the guidance term used in the DPS paper for example.
> > We have extensively experimented with our method and empirically noticed that when we use $\ell_k \approx k$ in our method, and on challenging image reconstruction tasks such as half mask, we are simply unable to reach the reconstruction quality (coherence and details) that we obtain with $\ell_k = \lfloor 0.5 k \rfloor$.
> >
> > [1] Zhang, Guanhua, Jiabao Ji, Yang Zhang, Mo Yu, Tommi S. Jaakkola, and Shiyu Chang. "Towards coherent image inpainting using denoising diffusion implicit models." (2023)
> > [2] Liu, Anji, Mathias Niepert, and Guy Van den Broeck. "Image Inpainting via Tractable Steering of Diffusion Models." arXiv preprint arXiv:2401.03349 (2023)

---

> > > ### Author Response · Authors · 2024-11-25
> > >
> > > Dear Reviewer U6Fn,
> > >
> > > We are grateful for your insightful review of our work. Your comments have been invaluable in helping us improve the quality and clarity of our paper. We have carefully considered each of your points and have made a thorough and comprehensive response to address your concerns (here, and in the Main Comment above).
> > >
> > > We would like to inquire if you have any further questions regarding our response. Your insights are valuable to us, and we greatly appreciate your attention and feedback!

---

> > > > ### Comment · Reviewer_U6Fn · 2024-11-25
> > > >
> > > > Thank you for the kind words and also for your answers and explanations, which addressed my initial concerns. I would like to retain my initial rating.

---

### Author Response · Authors · 2024-11-21
**General comment**

We thank the reviewers for taking the time to give their much-valued feedback on our work. We believe that the reviews and the discussion will contribute to improving the clarity of our work.  We also thank the reviewers for pointing out the innovative aspects, the robust theoretical foundation, and the strong empirical results of our approach on a variety of tasks and modalities. We have addressed the suggestions and questions from each reviewer individually in their dedicated rebuttal section and have specified the additions that we made in the revised version of our work attached. Specifically:
- We have improved the notation in the paper as well as the presentation of the methodology in Section 3. We now provide a more intuitive motivation for our approach and include a figure to illustrate it.
- We have extended the metrics tables in the appendix of the paper by adding PSNR and SSIM; see Table 7, 8 and 9.
- Below, we provide additional metrics that were computed on the basis of 300 sample images and on what we believe are the most challenging tasks. We include 95% confidence intervals indicating that our results are statistically significant. The tables in the final version of the paper will be updated accordingly with metrics computed on 1000 sample images.

---

> ### Author Response · Authors · 2024-11-21
> **General comment (continued)**
>
> ### Results on FFHQ
>
> **LPIPS**
>
> | Task | MGPS | DPS | PGDM | DDNM | DIFFPIR| REDDIFF |
> ---|---|---|---|---|---|---|
> Half mask          |**0.19** ± 0.01 |0.24 ± 0.01 |0.24 ± 0.01 |0.23 ± 0.01 |0.25 ± 0.01 |0.28 ± 0.01 |
> JPEG (QF=2)                     |**0.15** ± 0.01 |0.34 ± 0.03 |1.12 ± 0.01 |- |- |0.32 ± 0.01 |
> Motion Deblur               |**0.12** ± 0.01 |0.17 ± 0.01 |- |- |- |0.22 ± 0.01 |
> Nonlinear Deblur            |**0.23** ± 0.01 |0.51 ± 0.04 |- |- |- |0.68 ± 0.02 |
> | High Dynamic Range |  **0.07** ± 0.02    | 0.40 ± 0.06   | -    | -   | -   | 0.20 +\- 0.03  |
>
> **PSNR**
>
> | Task | MGPS | DPS | PGDM | DDNM | DIFFPIR| REDDIFF |
> ---|---|---|---|---|---|---|
> Half mask          |15.91 ± 0.28 |14.86 ± 0.26 |15.29 ± 0.28 |**16.38** ± 0.35 |16.04 ± 0.36 |15.68 ± 0.34 |
> JPEG (QF=2)                     |**25.23** ± 0.17 |19.56 ± 0.60 |12.57 ± 0.10 |- |- |24.53 ± 0.13 |
> Motion Deblur               |26.71 ± 0.21 |24.13 ± 0.21 |- |- |- |**27.48** ± 0.13 |
> Nonlinear Deblur            |**24.35** ± 0.31 |16.08 ± 0.87 |- |- |- |21.94 ± 0.25 |
> | High Dynamic Range |  **26.95** ± 0.20    | 18.71 ± 0.32   | -    | -   | -   |  21.69 ± 0.20  |
>
> **SSIM**
>
> | Task | MGPS | DPS | PGDM | DDNM | DIFFPIR| REDDIFF |
> ---|---|---|---|---|---|---|
> Half mask          |0.70 ± 0.01 |0.67 ± 0.01 |0.59 ± 0.01 |**0.74** ± 0.01 |0.72 ± 0.01 |0.63 ± 0.01 |
> JPEG (QF=2)                     |**0.73** ± 0.01 |0.56 ± 0.03 |0.10 ± 0.01 |- |- |0.71 ± 0.01 |
> Motion Deblur               |**0.77** ± 0.01 |0.70 ± 0.01 |- |- |- |0.71 ± 0.01 |
> Nonlinear Deblur            |**0.70** ± 0.01 |0.44 ± 0.03 |- |- |- |0.42 ± 0.01 |
> | High Dynamic Range |  **0.83** ± 0.04    | 0.55 ± 0.06   | -    | -   | -   |  0.72 ± 0.04 |
>
>
> ### Results on ImageNet
>
> **LPIPS**
>
> | Task | MGPS | DPS | PGDM | DDNM | DIFFPIR| REDDIFF |
> |----|-----|-----|----|----|-----|-----|
> | Half Mask  | **0.31** ± 0.03  | 0.40 ± 0.03  | 0.34 ± 0.03  | 0.38 ± 0.03  | 0.40 ± 0.03 | 0.46 ± 0.03 |
> | Motion Deblur     | **0.20** ± 0.03  | 0.40 ± 0.04  | -  | -  | -  | 0.39 ± 0.04 |
> | JPEG (QF=2)             |    **0.41** ± 0.05  | 0.60 ± 0.06   | -   | -  | -   | 0.49 ± 0.04   |
> | Nonlinear Deblur|  **0.43** ± 0.04    |0.82 ± 0.05 | -    | -   | -   | 0.66 ± 0.05 |
> | High Dynamic Range|  **0.10** ± 0.03    | 0.84 ± 0.05   | -    | -   | -   | 0.19 ± 0.04  |
>
> **PSNR**
>
> | Task | MGPS | DPS | PGDM | DDNM | DIFFPIR| REDDIFF |
> |----|-----|-----|----|----|-----|-----|
> | Half Mask  | 14.96 ± 0.18 | 12.15 ± 0.19   | 14.05 ± 0.19  | **15.97** ± 0.21|  15.64 ± 0.22  | 14.84 ± 0.20  |
> | Motion Deblur     | **24.27** ± 0.20 | 21.38 ± 0.21  | -   | -   | -   | 24.06 ± 0.19 |
> | JPEG (QF=2)             |    **22.08** ± 0.19    | 16.33 ± 0.27  | -  | -   | -  | 22.07 ±  0.18  |
> | Nonlinear Deblur| **22.13±0.22**    |10.13 ± 0.28   | -    | -   | -   |  20.57 ± 0.18  |
> | High Dynamic Range|  **26.31** ± 0.23   | 9.56 ± 0.26  | -   | -  | -  | 22.12 ± 0.23  |
>
> **SSIM**
>
> | Task | MGPS | DPS | PGDM | DDNM | DIFFPIR| REDDIFF |
> |----|-----|-----|----|----|-----|-----|
> | Half Mask  | 0.63 ± 0.03 | 0.58 ± 0.03  | 0.52 ± 0.02  | **0.68** ± 0.03  | 0.67 ± 0.03 | 0.59 ± 0.03 |
> | Motion Deblur     | **0.66** ± 0.04| 0.55 ± 0.05   | -   | -   | -  | 0.61 ± 0.03  |
> | JPEG (QF=2)     |  **0.60** ± 0.04  | 0.40 ± 0.06   | -   | -  | -    | 0.59 ± 0.04   |
> | Nonlinear Deblur|  **0.58** ± 0.04    | 0.25 ± 0.06   | -    | -   | -   |  0.41 ± 0.04  |
> | High Dynamic Range|  **0.83** ± 0.04  | 0.23 ± 0.06   | -    | -   | -   | 0.72 ± 0.04  |
>
>
> ### Result on FFHQ with LDM
>
> **LPIPS**
>
> | Task | MGPS | ReSample | PSLD |
> |----|-----|-----|----|
> | Half Mask  | **0,26** ± 0.01| 0.30 ± 0.03  | 0.32 ± 0.03  |
> | Motion Deblur     | **0.19** ± 0.01  | 0.20 ± 0.03  | 0.70 ± 0.03|
> | JPEG (QF=2)             |    **0.21** ± 0.03      |0.26±0.03   | -  |
> | Nonlinear Deblur|  **0.26** ± 0.03    |0.33±0.04   | -  |
> | High Dynamic Range|  0.14 ± 0.03    | **0.12** ± 0.03   | -    |
>
> **PSNR**
>
> | Task | MGPS | ReSample | PSLD |
> |----|-----|-----|----|
> | Half Mask  | **15.30** ± 0.33 | 14.89 ± 0.17  | 14.62±0.19  |
> | Motion Deblur     | 26.39 ± 0.21 | **26.73** ± 0.15  |17.71±0.12 |
> | JPEG (QF=2)             |    24.66±0.14      | **24.77** ± 0.15   | -  |
> | Nonlinear Deblur|  23.83±0.18    | **24.10** ± 0.19   | -  |
> | High Dynamic Range|  25.41±0.20     | **25.91** ±0.21   | -    |
>
> **SSIM**
>
> | Task | MGPS | ReSample | PSLD |
> |----|-----|-----|----|
> | Half Mask  | **0.69** ± 0.01  | 0.67 ± 0.02 | 0.60 ± 0.03  |
> | Motion Deblur     | **0.76** ± 0.01  | 0.72 ± 0.03 | 0.24 ± 0.02|
> | JPEG (QF=2)             |    **0.72** ± 0.03      |0.66 ± 0.03   | -  |
> | Nonlinear Deblur|  **0.70** +/ -0.03    |0.67 ± 0.03  | -  |
> | High Dynamic Range|  0.79 ± 0.03     | **0.83** ± 0.03   | -    |

---

> ### Author Response · Authors · 2024-11-25
> **Update with Finalized Experiments**
>
> We would like to thank the reviewers once again for their insightful comments.
>
> We have finalized the previously mentioned experiments. To strengthen our conclusions and calculate the **FID**, we reran MGPS and the two closest competitors on **1000 images** for the following five tasks: half mask, JPEG, motion deblur, nonlinear deblur, and high dynamic range, using the three priors (ImageNet, FFHQ, FFHQ latent). The tables presented below show that MGPS significantly outperforms the other competitors, including for the FID, with no significant change in other metrics compared to what we calculated on a smaller dataset.
>
> We believe we have addressed all the reviewers' concerns. However, we remain open to any additional feedback that could further improve the quality and robustness of our paper and would be happy to address any further questions before the end of the rebuttal period.

---

> > ### Author Response · Authors · 2024-11-25
> > **Update with Finalized Experiments (continued)**
> >
> > # Results on FFHQ
> > | Task | Metric | MGPS | DDNM | DIFFPIR |
> > |----|-----|-----|----|----|
> > | Half Mask | **FID** | **27.01** | 38.62 | 45.20  |
> > | Half Mask | LPIPS |0.19+/-0.003| 0.23+/-0.004 |0.25+/-0.003  |
> > | Half Mask | PSNR | 15.86+/-0.12 | 16.27+/-0.13 |16.14+/-0.14  |
> > | Half Mask | SSIM | 0.70+/-0.003| 0.74+/-0.004 |0.72+/-0.005  |
> >
> > | Task | Metric | MGPS | DPS | REDDIFF |
> > |----|-----|-----|----|----|
> > | Motion Deblur  | **FID** | **29.69** | 36.67 |77.01  |
> > | Motion Deblur  | LPIPS | 0.12+/-0.002 | 0.17+/-0.003 | 0.22+/-0.003  |
> > | Motion Deblur  | PSNR | 26.68+/-0.12 | 24.12+/-0.12 | 27.42+/-0.07 |
> > | Motion Deblur  | SSIM | 0.77+/-0.004 | 0.70+/-0.005 |0.71+/-0.002 |
> > | JPEG2  | **FID** | **31.60** | 87.58 |108.54  |
> > | JPEG2  | LPIPS | 0.15+/-0.003 | 0.37+/-0.02 | 0.33+/-0.005  |
> > | JPEG2  | PSNR | 25.20+/-0.09 | 18.96+/-0.19 | 24.47+/-0.08  |
> > | JPEG2  | SSIM | 0.73+/-0.005 | 0.55+/-0.02 |0.70+/-0.004 |
> > | Nonlinear Deblur  | **FID** | **50.81** | 163.60 |88.38  |
> > | Nonlinear Deblur  | LPIPS | 0.23+/-0.005 | 0.51+/-0.02 | 0.68+/-0.01  |
> > | Nonlinear Deblur  | PSNR | 24.28+/-0.16 | 16.19+/-0.47 | 21.89+/-0.13 |
> > | Nonlinear Deblur  | SSIM | 0.70+/-0.005 | 0.45+/-0.02 |0.42+/-0.006 |
> > | High Dynamic Range  | **FID** | **20.91** | 152.70 |47.50  |
> > | High Dynamic Range  | LPIPS | 0.08+/-0.01 | 0.40+/-0.04 | 0.20+/-0.01  |
> > | High Dynamic Range  | PSNR | 26.95+/-0.11 | 18.71+/-0.18 | 21.69+/-0.11  |
> > | High Dynamic Range  | SSIM | 0.83+/-0.01 | 0.55+/-0.04 |0.72+/-0.01 |
> >
> > # Results on ImageNet
> > | Task | Metric | MGPS | DDNM | DIFFPIR |
> > |----|-----|-----|----|----|
> > | Half Mask | **FID** | **40.09** | 50.02 |56.99  |
> > | Half Mask | LPIPS | 0.30+/-0.004 | 0.38+/-0.005 |0.40+/-0.005  |
> > | Half Mask | PSNR |15.01+/-0.10 | 16.02+/-0.12 |15.75+/-0.12  |
> > | Half Mask | SSIM | 0.63+/-0.004| 0.68+/-0.005 | 0.67+/-0.005  |
> >
> > | Task | Metric | MGPS | DPS | REDDIFF |
> > |----|-----|-----|----|----|
> > | Motion Deblur  | **FID** |**35.33** | 55.05 |87.29  |
> > | Motion Deblur  | LPIPS |0.20+/-0.005 | 0.40+/-0.009 |0.39+/-0.007  |
> > | Motion Deblur  | PSNR | 24.36+/-0.09 | 21.44+/-0.11 |24.17+/-0.08 |
> > | Motion Deblur  | SSIM | 0.67+/-0.007 | 0.55+/-0.01 | 0.61+/-0.004 |
> > | JPEG  | **FID** | **61.35** | 128.77 |92.84  |
> > | JPEG  | LPIPS |0.40+/-0.009 | 0.60+/-0.014 |0.47+/-0.008  |
> > | JPEG  | PSNR | 22.15+/-0.10 | 16.66+/-0.14 |22.19+/-0.09  |
> > | JPEG  | SSIM | 0.60+/-0.008 | 0.41+/-0.015 | 0.60+/-0.007 |
> > | High Dynamic Range  | **FID** |**20.20** | 315.27 |35.74  |
> > | High Dynamic Range  | LPIPS |0.11+/-0.005 | 0.83+/-0.015 |0.20+/-0.007 |
> > | High Dynamic Range  | PSNR | 26.27+/-0.15 | 9.90+/-0.18 | 21.91+/-0.14 |
> > | High Dynamic Range  | SSIM | 0.83+/-0.007 | 0.23+/-0.016 | 0.71+/-0.008 |
> >
> > # Results on FFHQ with LDM
> > | Task | Metric | MGPS | Resample |
> > |----|----|----|----|
> > | Half Mask  | **FID** | **49.45** | 66.55  |
> > | Half Mask  | LPIPS | 0.26+/-0.004 | 0.30+/-0.003  |
> > | Half Mask  | PSNR | 15.56+/-0.13 | 14.73+/-0.09  |
> > | Half Mask  | SSIM | 0.69+/-0.004 | 0.67+/-0.003 |
> > | Motion Deblur | **FID** | **44.58** | 51.77  |
> > | Motion Deblur | LPIPS | 0.19+/-0.004 | 0.20+/-0.004  |
> > | Motion Deblur | PSNR | 26.36+/-0.11 | 26.68+/-0.10 |
> > | Motion Deblur | SSIM | 0.76+/-0.004 | 0.72+/-0.004 |
> > | JPEG  | **FID** | **45.07** | 65.30  |
> > | JPEG  | LPIPS | 0.21+/-0.004 | 0.26+/-0.005  |
> > | JPEG  | PSNR | 24.64+/-0.08 | 24.77+/-0.09 |
> > | JPEG  | SSIM | 0.71+/-0.004 | 0.65+/-0.005 |
> > | Nonlinear Deblur | **FID**| **69.19** |71.51  |
> > | Nonlinear Deblur | LPIPS | 0.26+/-0.005 | 0.32+/-0.007  |
> > | Nonlinear Deblur | PSNR | 23.87+/-0.09 | 24.18+/-0.10 |
> > | Nonlinear Deblur | SSIM | 0.69+/-0.005 | 0.67+/-0.005 |
> > | High Dynamic Range  | **FID** | 44.15 | 38.71  |
> > | High Dynamic Range  | LPIPS | 0.14+/-0.003 | 0.12+/-0.004  |
> > | High Dynamic Range  | PSNR | 25.45+/-0.10 | 25.98+/-0.11 |
> > | High Dynamic Range  | SSIM | 0.80+/-0.006 | 0.83+/-0.005 |

---

### Meta-Review · Area_Chair_yGaU · 2024-12-18

**Metareview:**

The paper proposes a diffusion-based method for posterior sampling in diffusion models. The four reviewers all indicated that the paper is clearly above threshold for acceptance. They found the paper to be written well and present an idea that could be found widely useful. The empirical evaluation was sufficient to demonstrate its improvement over other SOTA models. During a detailed discussion phase, the authors provided additional information that caused three reviewers to improve their scores. It is important for this additional information to be incorporated into the final draft of the paper.

**Additional Comments On Reviewer Discussion:**

The authors provided much details during the discussion period and the reviewers were engaged in this, with three of them increasing their scores, in some cases significantly.

---

### Decision · Program_Chairs · 2025-01-22

Accept (Oral)